# Temporal integration of auxin information for the regulation of patterning

Carlos S Galvan-Ampudia[1†], Guillaume Cerutti[1†], Jonathan Legrand[1], Géraldine Brunoud[1], Raquel Martin-Arevalillo[1], Romain Azais[1], Vincent Bayle[1], Steven Moussu[1‡], Christian Wenzl[2], Yvon Jaillais[1], Jan U Lohmann[2], Christophe Godin[1]*, Teva Vernoux[1]*

[1]Laboratoire Reproduction et Développement des Plantes, Univ Lyon, ENS de Lyon, UCB Lyon 1, CNRS, INRAE, Inria, Lyon, France; [2]Department of Stem Cell Biology, Centre for Organismal Studies, Heidelberg University, Heidelberg, Germany

**Abstract** Positional information is essential for coordinating the development of multicellular organisms. In plants, positional information provided by the hormone auxin regulates rhythmic organ production at the shoot apex, but the spatio-temporal dynamics of auxin gradients is unknown. We used quantitative imaging to demonstrate that auxin carries high-definition graded information not only in space but also in time. We show that, during organogenesis, temporal patterns of auxin arise from rhythmic centrifugal waves of high auxin travelling through the tissue faster than growth. We further demonstrate that temporal integration of auxin concentration is required to trigger the auxin-dependent transcription associated with organogenesis. This provides a mechanism to temporally differentiate sites of organ initiation and exemplifies how spatio-temporal positional information can be used to create rhythmicity.

**\*For correspondence:**
christophe.godin@ens-lyon.fr (CG);
teva.vernoux@ens-lyon.fr (TV)

[†]These authors contributed equally to this work

**Present address:** [‡]Department of Plant Molecular Biology, University of Lausanne, Lausanne, Switzerland

**Competing interests:** The authors declare that no competing interests exist.

## Introduction

Specification of differentiation patterns in multicellular organisms is regulated by gradients of biochemical signals providing positional information to cells (*Rogers and Schier, 2011*; *Wolpert, 1969*). In plants, graded distribution of the hormone auxin is not only essential for embryogenesis, but also for post-embryonic development, where it regulates the reiterative organogenesis characteristic of plants (*Dubrovsky et al., 2008*; *Vanneste and Friml, 2009*; *Benková et al., 2003*). Plant shoots develop post-embryonically through rhythmic organ generation in the shoot apical meristem (SAM), a specialized tissue with a stem cell niche in its central zone (CZ; *Figure 1A*). In *Arabidopsis thaliana*, as in a majority of plants, organs are initiated sequentially in the SAM peripheral zone (PZ surrounding the CZ) at consecutive relative angles of close to 137°, either in a clockwise or anti-clockwise spiral (*Figure 1A*; *Galvan-Ampudia et al., 2016*). SAM organ patterning or phyllotaxis has been extensively analyzed using mathematical models (*Douady and Couder, 1996*; *Mitchison, 1977*; *Veen and Lindenmayer, 1977*). A widely accepted model proposes that the time interval between organ initiations (the plastochron) and the spatial position of organ initiation emerge from the combined action of inhibitory fields emitted by pre-existing organs and the SAM center (*Douady and Couder, 1996*). Tissue growth then self-organizes organ patterning by moving organs away from the stem cells and leaving space for new ones.

Auxin is the main signal for positional information in phyllotactic patterning (*Reinhardt et al., 2003a*; *Reinhardt et al., 2000*). Auxin, has been proposed to be transported directionally toward incipient primordia where it activates a transcriptional response leading to organ specification (*Benková et al., 2003*; *Reinhardt et al., 2003a*; *Heisler et al., 2005*; *Vernoux et al., 2000*). PIN-FORMED1 (PIN1) belongs to a family of auxin efflux carriers whose polarity determines the direction of auxin fluxes (*Benková et al., 2003*; *Gälweiler et al., 1998*). PIN1 proteins are present throughout

**eLife digest** Plants, like animals and many other multicellular organisms, control their body architecture by creating organized patterns of cells. These patterns are generally defined by signal molecules whose levels differ across the tissue and change over time. This tells the cells where they are located in the tissue and therefore helps them know what tasks to perform.

A plant hormone called auxin is one such signal molecule and it controls when and where plants produce new leaves and flowers. Over time, this process gives rise to the dashing arrangements of spiraling organs exhibited by many plant species. The leaves and flowers form from a relatively small group of cells at the tip of a growing stem known as the shoot apical meristem.

Auxin accumulates at precise locations within the shoot apical meristem before cells activate the genes required to make a new leaf or flower. However, the precise role of auxin in forming these new organs remained unclear because the tools to observe the process in enough detail were lacking.

Galvan-Ampudia, Cerutti et al. have now developed new microscopy and computational approaches to observe auxin in a small plant known as *Arabidopsis thaliana*. This showed that dozens of shoot apical meristems exhibited very similar patterns of auxin. Images taken over a period of several hours showed that the locations where auxin accumulated were not fixed on a group of cells but instead shifted away from the center of the shoot apical meristems faster than the tissue grew. This suggested the cells experience rapidly changing levels of auxin. Further experiments revealed that the cells needed to be exposed to a high level of auxin over time to activate genes required to form an organ. This mechanism sheds a new light on how auxin regulates when and where plants make new leaves and flowers. The tools developed by Galvan-Ampudia, Cerutti et al. could be used to study the role of auxin in other plant tissues, and to investigate how plants regulate the response to other plant hormones.

the SAM and regulate the spatio-temporal distribution of auxin cooperatively with other carriers (*Reinhardt et al., 2003a*; *Bainbridge et al., 2008*). Convergence of PIN1 carriers toward sites of organ initiation was proposed to control an accumulation of auxin that triggers organ initiation. This spatial organization of PIN1 polarities was also proposed to deplete auxin around organs, locally blocking initiation and thus establishing auxin-based inhibitory fields (*Reinhardt et al., 2003a*; *Heisler et al., 2005*; *Vernoux et al., 2011*; *de Reuille et al., 2006*; *Stoma et al., 2008*; *Jonsson et al., 2006*; *Smith et al., 2006a*). In addition, a reduced responsivity of the CZ to auxin has been demonstrated, providing an auxin-dependent mechanism for the inhibition of organogenesis in the CZ (*Vernoux et al., 2011*; *de Reuille et al., 2006*). Several models converge to suggest that together, these auxin-dependent regional cues determine new organ locations in the growing SAM.

The genetically-encoded biosensor DII-VENUS, a synthetic protein degraded directly upon sensing of auxin, recently allowed an unprecedented qualitative visualization of spatial auxin gradients in the SAM (*Vernoux et al., 2011*; *Brunoud et al., 2012*). However, quantification of the spatio-temporal dynamics of auxin is required to fully evaluate both experimental and theoretical understanding of the action of auxin in SAM patterning. This is all the more important given that the continuous helicoidal reorganization of auxin distribution in the growing SAM, suggests that auxin might convey complex positional information. Here, we used a quantitative imaging approach to question the nature of the auxin-dependent positional information. We further investigate how efflux and biosynthesis regulate the 4D dynamics of auxin, and explore how this information is processed in the SAM to generate rhythmic patterning.

## Results

### Spatio-temporal auxin distribution

In the SAM, DII-VENUS fluorescence reports auxin concentration with cellular resolution (*Vernoux et al., 2011*; *Brunoud et al., 2012*). To extract quantitative information about auxin distribution, we generated a DII-VENUS ratiometric variant, hereafter named qDII (quantitative DII-

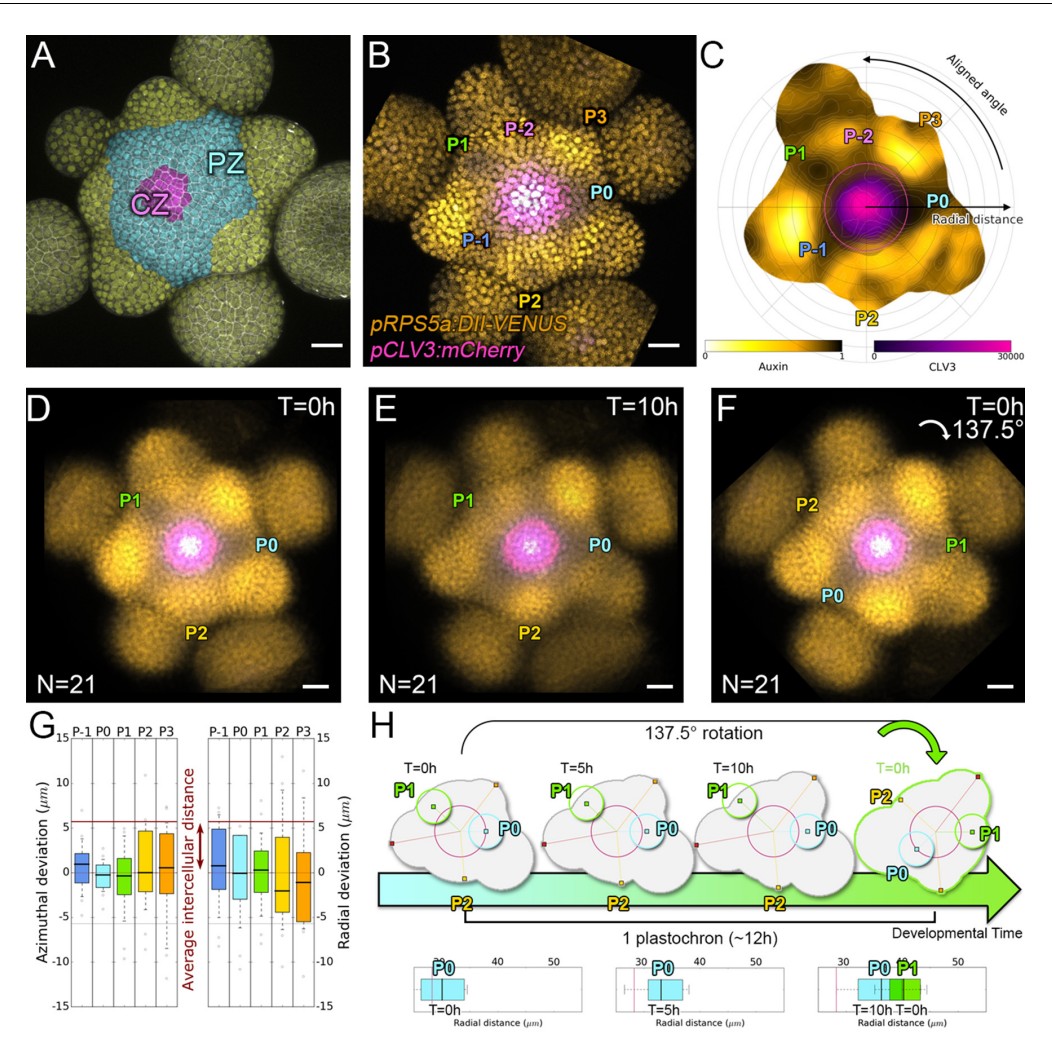

**Figure 1.** Spatial auxin distribution in the SAM follows a precise reiterative pattern. (A) SAM radial organization. The CZ (magenta) is surrounded by the PZ (cyan). Emerging flower primordia and flowers are colored in yellow. (B). Representative expression patterns of DII-VENUS-N7 (yellow) and *pCLV3:mCherry* transcriptional reporter line (magenta). Primordia are indicated by color and rank. (C). Auxin map (1-qDII, yellow to black) of (B). *CLV3* expression (magenta) and radial extension (circle) are shown. Black arrows depict radial distance from the center and aligned angle. (D–F). Superposition of 21 aligned SAM images at time 0 hr (D), and 10 hr (E). (F) 137.5° clockwise rotation of (D) results in a quasi-identical image of (E). See *Figure 1—figure supplement 2A* for non-aligned image superposition. Scale bars = 20 μm. (G). Precision in auxin maxima positioning measured using angular position deviation (azimuthal deviation, left panel) and radial direction (right panel). Red lines indicate the average cellular distance. N = 21 meristems. Colors indicate primordium ranks (P$_{-1}$ blue, P$_0$ cyan, P$_1$ green, P$_2$ yellow, P$_3$ orange) (H). Space can be used as a proxy for time, as a rotation of 1 divergence angle is equivalent to a translation of 1 plastochron in time.

The online version of this article includes the following figure supplement(s) for figure 1:

**Figure supplement 1.** Expression pattern of qDII and *CLV3*.

**Figure supplement 2.** Precision in the positioning of auxin maxima enables time extrapolation.

**Figure supplement 3.** Auxin depletion dynamics.

---

VENUS). qDII differs from previously used tools (*Liao et al., 2015*) in producing DII-VENUS and a non-degradable TagBFP reference stoichiometrically from a single RPS5A promoter (*Wend et al., 2013*; *Goedhart et al., 2011*; *Figure 1—figure supplement 1A–H*). By introducing a stem cell-specific *pCLV3:mCherry* nuclear transcriptional reporter into plants expressing qDII (*Pfeiffer et al.,*

*2016*) we generated a functional and robust geometrical reference for the SAM center (*Figure 1B,C* and *Figure 1—figure supplement 1I–M*).

All analyzed meristems (21 individual SAM) showed qDII patterns similar to those obtained with DII-VENUS, with locations of auxin maxima following the phyllotactic pattern (*Vernoux et al., 2011*; *Figure 1B–E*). Despite the fact that SAMs were imaged independently and not synchronized, qDII patterns appeared highly stereotypical with easily identifiable fluorescence maxima and minima. This was confirmed by image alignment using SAM rotations (applying prior mirror symmetry if necessary; *Figure 1D* and *Figure 1—figure supplement 2A–C*). All images could be superimposed preserving the spatial distribution of auxin maxima and minima (*Figure 1—figure supplement 2B*). Our analysis shows that auxin distribution follows the same synchronous pattern across a population of SAMs, with low angular and rhythmic variability (*Figure 1—figure supplement 2D–E*, Appendix 2), with apparent stationarity up to a 137° rotation (*Figure 1H*).

To further quantify auxin distribution, we developed a mostly automated computational pipeline to measure SAM fluorescence (Appendix 3) (*Cerutti et al., 2020*). We used the spatial distribution of 1-DII-VENUS/TagBFP as a proxy for auxin distribution, hereafter named 'auxin' (*Figure 1C*) and focused on the epidermal cell layer (L1) where organ initiation takes place (*Jonsson et al., 2006*; *Kierzkowski et al., 2013*; *Smith et al., 2006b*; *Reinhardt et al., 2003b*). The location of the absolute auxin maximum value was defined as Primordium 0 ($P_0$). Other local maxima with lower auxin values were called $P_n$ (Appendix 1), with n corresponding to their rank in the phyllotactic spiral (*Figure 1C* and *Figure 1—figure supplement 2B*). Note that the dynamic range of qDII allows measuring an auxin value for the vast majority of cells in the PZ and only a few cells at P0 had undetectable values of DII-VENUS, leading to an auxin value of 1. The pipeline then permits the quantification of nuclear signals and aligns all the SAMs onto a common clockwise reference frame with standardized x,y,z-orientation and with the $P_0$ maximum to the right. This automatic registration confirmed that auxin maxima follow a phyllotactic pattern with a divergence angle close to 137.5° (*Figure 1—figure supplement 2F*). It also demonstrated that maxima are positioned with a precision close to the size of a cell both in distance from the SAM center and in azimuth (angular distance) with a maximal standard deviation of 8.4 μm or 1.5 cell diameters (*Figure 1G*).

We then considered the temporal changes in auxin distribution by using time-lapse images over one plastochron, which corresponds to the period of this rhythmic system. $P_0$ and successive auxin maxima moved radially (*Figure 1—figure supplement 2D*). Remarkably, while the average radial distance from each local maximum $P_n$ to the SAM center progresses (*Figure 1—figure supplement 2G*), the spatial deviation of this distance does not change significantly over time, reflecting the synchronized movement of local maxima, with limited meristem to meristem variation. After 10 hr, every $P_n$ local maximum has almost reached the starting position of the next local maximum, $P_{n+1}$, but after 14 hr they have passed this position (*Figure 1—figure supplement 2G*). This suggests that a rotation of 137.5°, which replaces $P_n$ by $P_{n+1}$, corresponds to a temporal progression of 10 to 14 hr (*Figure 1H*). This was supported by dissimilarity measurements obtained using different rotation angles between maps (*Figure 1—figure supplement 2H*), allowing us to confirm that plastochron last 12h ± 2h. We could thus derive a continuum of primordium development by placing $P_{n+1}$ time series one plastochron (12 hr) after $P_n$ time series on a common developmental time axis (*Figure 1H*). Together with the observed developmental stationarity, this permitted the

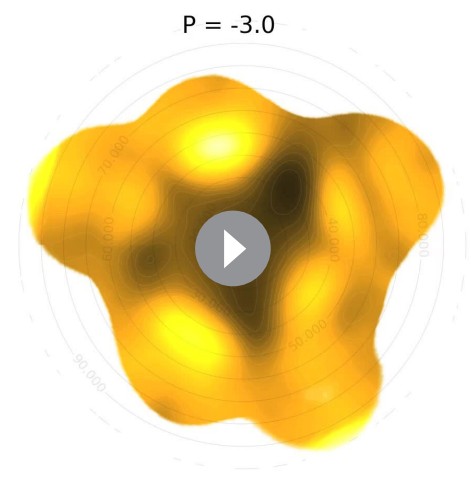

P = -3.0

**Video 1.** Auxin developmental continuum over nine plastochrons. Auxin distribution dynamics in the SAM obtained from population averaging and temporal extrapolation. The developmental stage indicated at the top p=n corresponds to the area located on the right. Color code: yellow = low auxin, to black = high auxin.
https://elifesciences.org/articles/55832#video1

reconstruction of auxin dynamics over several plastochrons from observations spanning only one. The resulting quantitative temporal map of auxin distribution in the SAM reveals the dynamic genesis of auxin maxima in the PZ first as finger-like protrusions (visible at $P_{-2}$, $P_{-1}$ and $P_0$) from a permanent high auxin zone at the center of the SAM (*Figure 1—figure supplement 2I–L* and *Video 1*), as previously predicted (*de Reuille et al., 2006*). At later stages, auxin maxima become confined to fewer cells while auxin minima are progressively established precisely in between auxin maxima and the CZ (*Figure 1—figure supplement 3*).

We next wondered whether the motion of auxin maxima and minima could result purely from cellular growth, an hypothesis used in several theoretical models (*Douady and Couder, 1996*; *Jonsson et al., 2006*; *Smith et al., 2006b*; *Heisler and Jönsson, 2006*). By following a $P_1$ maximum, we observed that cells within the auxin maximum zone closest to the CZ at time 0 hr gradually transfer to the depletion zone at time 10 hr (*Figure 2A–C*; nuclei circled in white). At the same time, cells on the distal edge of the maximum zone show a progressive increase in their auxin level (*Figure 2A–C*; nuclei circled in red), suggesting a spatial shift of the auxin maximum relatively to the cellular canvas. To explore further this phenomenon, we used nuclear motion to estimate cell motion vectors and compare them with the motion of the center of auxin maximum zones, we further found that the average radial speed of auxin maxima between stages $P_1$ and $P_4$ can surpass the average displacement of individual nuclei, with a peak velocity of more than 1 µm/h at the $P_2$ stage (*Figure 2D–E*). These results show that auxin maxima are not attached to specific cells; instead they travel through the tissue, resulting in an apparent centrifugal wave of auxin accumulation. Consequently, the SAM cellular network provides a dynamic medium in which auxin maximum zones can move radially with their own apparent velocity relative to the growing tissue (*Figure 2D–E*). Analysis on time-courses of up to 14 hr revealed significant auxin variations in certain cells over one plastochron while auxin levels remained unchanged in others (*Figure 2F–G*). However, neighboring cells always showed limited differences in their temporal auxin profiles (*Figure 2F–G*). We concluded from these observations that there is a high definition spatio-temporal distribution of auxin, with auxin apparent movement occurring faster than growth within the tissue and providing cells with graded positional information in space and time (*Figure 2H*).

## Spatio-temporal control of auxin efflux and biosynthesis

The creation of auxin maxima first as protrusions of a high auxin zone in the CZ contrasts with the current vision of organogenesis being triggered by local auxin accumulation at the periphery of the CZ with concomitant auxin depletion around auxin maxima (*Reinhardt et al., 2003a*; *de Reuille et al., 2006*; *Stoma et al., 2008*; *Jonsson et al., 2006*; *Smith et al., 2006b*). This, in addition to the partial uncoupling of auxin distribution dynamics and growth, led us to reevaluate the spatio-temporal patterns of PIN1 localization, given their central role in controlling auxin distribution (*Reinhardt et al., 2003a*; *de Reuille et al., 2006*; *Jonsson et al., 2006*; *Smith et al., 2006b*). Co-visualization of a functional PIN1-GFP (*Benková et al., 2003*) and qDII/*CLV3* fluorescence over time showed that PIN1 concentration increases from $P_0$ and reaches a maximum at $P_2$ before decreasing (*Figure 3A,H* and *Figure 3—figure supplement 2*), consistent with previous observations (*Heisler et al., 2005*; *Bhatia et al., 2016*; *Caggiano et al., 2017*). To quantify PIN1 cell polarities, we used confocal images after cell wall staining with the fluorescent dye propidium iodide (PI) as a reference to position the PIN1-GFP signal relative to the L1 anticlinal cell walls at each cell-cell interface (*Shi et al., 2017*; *Figure 3B* and Appendix 4). This allowed us to compute PIN1-GFP polarity for each cell-cell interface of the SAM by extracting the 3D distribution of fluorescence for PI and GFP and quantifying the difference of intensity on membranes on both sides of the cell wall (*Figure 1C* and Appendix 4). These cell interface polarities measure in which direction each cell interface locally contributes to orient the flow of auxin transport. Using super-resolution radial fluctuation (SRRF) microscopy (*Gustafsson et al., 2016*) on the same samples, we could show that this method recovers cell interface PIN1 polarities with an error below 10% (8 out of 94 interfaces analyzed). When calculating cellular PIN1 polarity vectors by integrating the cell interface polarity information for each cell, we could further show that more than 80% of the cellular polarities deviate by less than 30° between the two approaches. This quantitative evaluation (*Figure 3D–G*, *Figure 3—figure supplement 1* and Appendix 4) validates the robustness of our method, showing that, in spite of a coarse image resolution, a vast majority of cellular polarity directions are consistent with super resolution imaging techniques. Our approach is therefore particularly suitable for monitoring global trends at

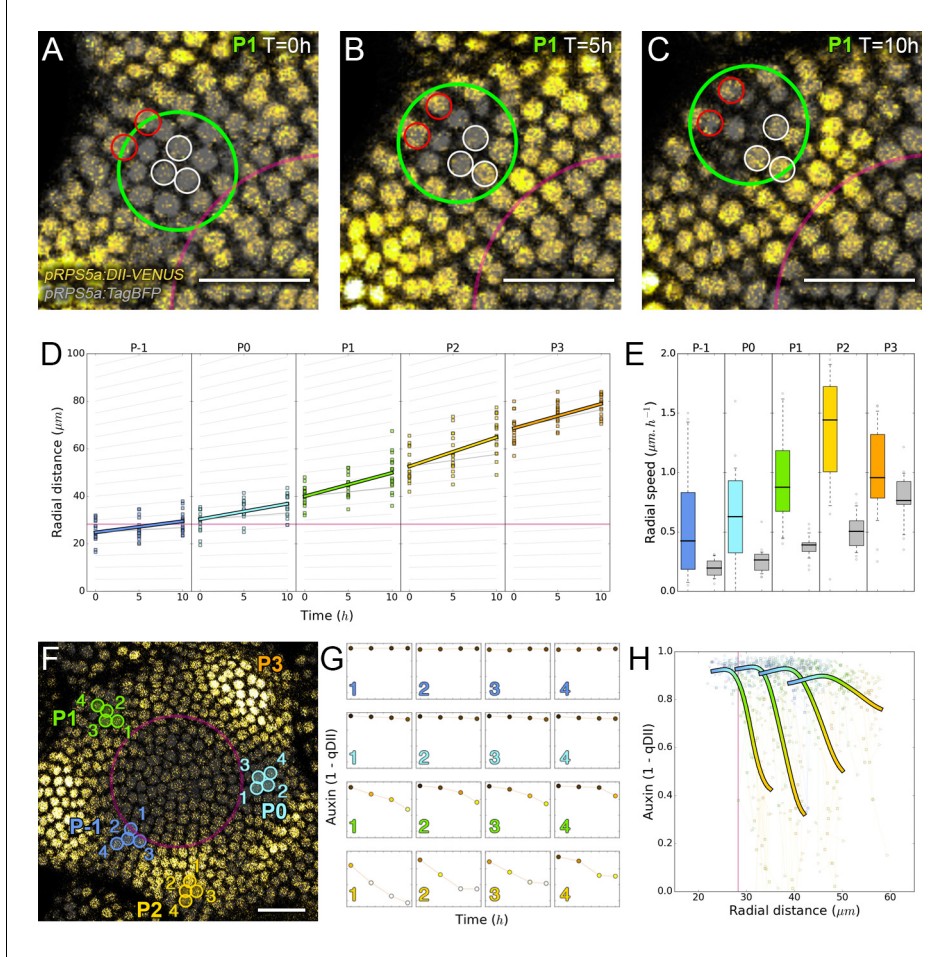

**Figure 2.** Auxin information travels as centrifugal waves in the meristem. (**A–C**) Representative projection of $P_1$ nuclei showing DII-VENUS-N7 (yellow) and TagBFP nuclei (grey) intensity changes in time. Time tracked nuclei are marked by white and red circles showing rapid decrease or increase of auxin over 10 hr, respectively. The green circle is centered on the position of the auxin maxima at each time point. The magenta line indicates the limit of the *CLV3* domain. Scale bars = 20 µm. (**D**) Average motion of maxima (colored lines) is faster than average cell motion (grey lines). The magenta line indicates the *CLV3* domain border. N = 21 meristems. (**E**) Compared distributions of radial motion speeds of auxin maxima (color boxplots) estimated as the slope of a linear regression per individual. Individual nucleus radial speed (gray boxplots) at the location of the maxima. N = 21 meristems. (**F–G**) Individual cells experience different auxin histories. Tracked cells at different locations (F; colored circles) and corresponding auxin levels (ordinate) over time (0, 5, 10, 14 hr). Scale bar = 20 µm. (**H**) Cellular mean auxin trajectories as a function of radial distance. Each line represents an extrapolated cell-size sector moving accordingly to cellular radial motion by its Gaussian average trajectory in radial distance (abscissa) and auxin value (ordinate). The color indicates the developmental stages at a given radial distance ($P_{-1}$ = blue, $P_0$ = cyan, $P_1$ = green, $P_2$ = yellow).

the scale of a tissue. Local averaging of the cellular vectors obtained from confocal images was then used to calculate continuous PIN1 polarity vector maps in order to identify the dominant trends in auxin flux directions in the SAM (*Figure 3I*, and Appendix 4). At the tissue scale, the vector maps demonstrate a strong convergence of PIN1 toward the center of the SAM (*Figure 3I–J* and *Figure 3—figure supplement 2*). In addition, PIN1 polarities deviate locally toward the radial axes followed by auxin maxima when they protrude from the CZ. We detected the previously observed inversion of PIN1 polarities at organ boundaries (*Heisler et al., 2005*) and our quantifications show that this occurs only from $P_7$ (*Figure 3—figure supplement 2C*), thus isolating the flower from the rest of the SAM from this late stage. $P_3$ to $P_5$ show a general flux toward the SAM that is locally deflected around the zones of auxin minima before converging back toward the SAM center

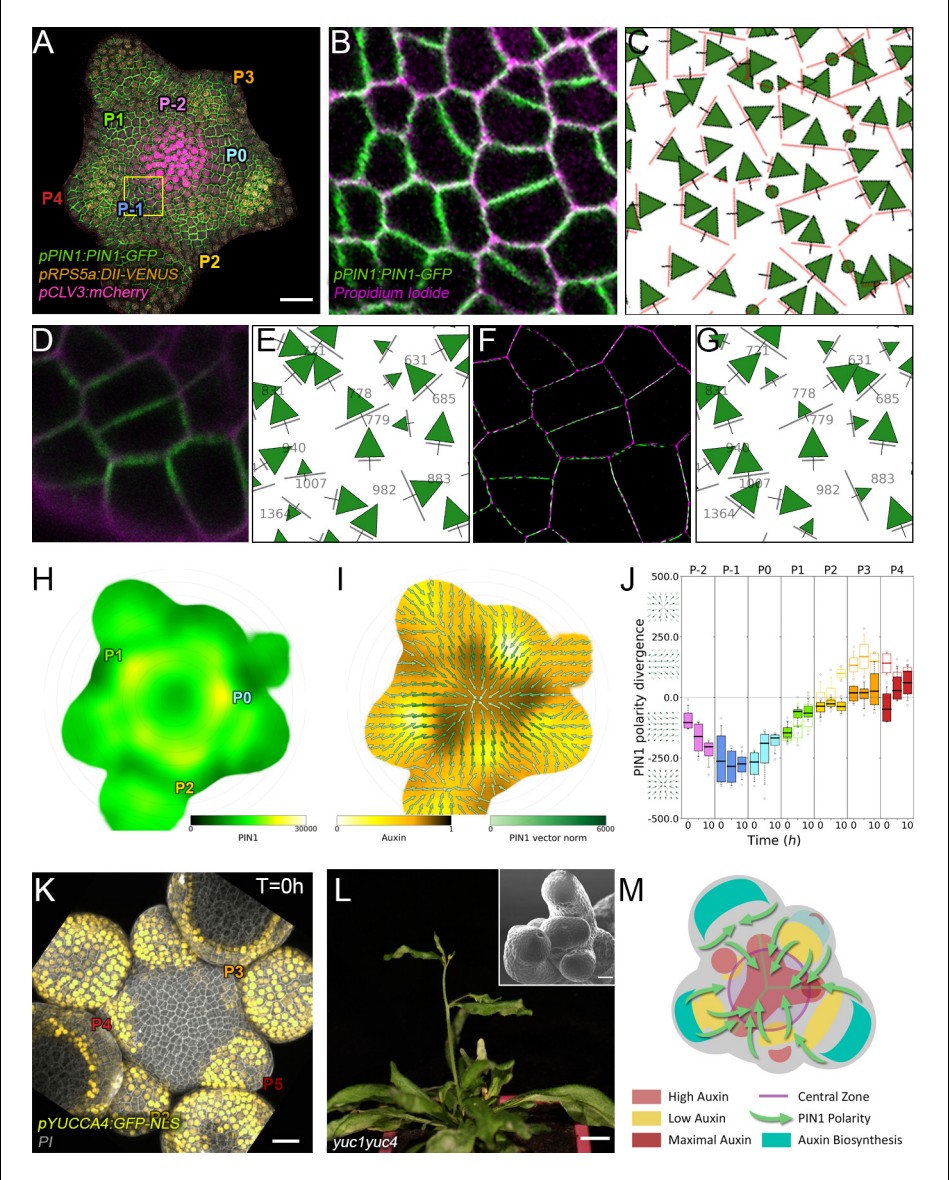

**Figure 3.** Spatio-temporal organization of auxin fluxes and biosynthesis. (**A**) Co-visualization of PIN1-GFP (green), DII-VENUS-N7 (yellow) and *pCLV3:mCherry* (magenta). Scale bar = 20 µm. Square shows P$_{-1}$ sector. (**B**) Magnified P$_{-1}$ region of (**A**) PIN1-GFP (green) and PI (magenta). (**C**) Computed PIN1 cell interface polarities of (**B**). Green arrows indicate polarities with a p-value<0.1, small arrows < 0.25 and dots > 0.25. (**D–G**). Image of PIN1-GFP (green) and cell wall (magenta) obtained using confocal (**D**) or super resolution (SRRF) microscopy (**F**) and respective PIN1 cell interface polarities (**E,G**). (**H**) Quantification of PIN1-GFP expression. N = 4 meristems. (**I**) PIN1 vector map (green arrows) organization correlated with auxin distribution (yellow to black). N = 4 meristems. (**J**) PIN1 polarity divergence index at auxin maxima (color filled boxplots) or auxin minima (white filled boxplots) positions during organ initiation. N = 4 meristems. (**K**) The YUC4 auxin biosynthesis limiting enzyme is specifically expressed in developing flowers. *YUC4:GFP* transcriptional reporter in yellow, cell wall (PI) staining in grey. Scale bars = 20 µm. (**L**) *yuc1yuc4* mutant inflorescence and meristem morphological defects (inset). Scale bars are 10 mm and 20 µm (inset). (**M**) Schematic representation of the tissue-scale organization of auxin transport and biosynthesis in relation to auxin distribution.

The online version of this article includes the following figure supplement(s) for figure 3:

**Figure supplement 1.** Validation of PIN1 polarity vectors with super resolution microscopy.
**Figure supplement 2.** Tissue-scale PIN polarity distribution.
**Figure supplement 3.** Expression patterns of *YUC* genes and phenotypes of *yuc* mutants.

(*Figure 3I* and *Figure 3—figure supplement 2*). Over the course of one plastochron, only limited changes in the PIN1 polarities are observed (*Figure 3—figure supplement 2*), suggesting that changes in auxin distribution at this time resolution do not require major adjustments in the direction of auxin efflux at the tissue scale.

We next asked where auxin could be produced in the SAM. YUCCAs (YUCs) have been shown to be limiting enzymes for auxin biosynthesis (*Cheng et al., 2006*; *Liu et al., 2016*). We thus mapped expression of the eleven YUC encoding genes in the SAM, using GFP reporter lines with a promoter fragment size shown to be functional for *YUC1,2* and 6 (*Figure 3—figure supplement 3A–N*; *Liu et al., 2016*; *Robert et al., 2013*). Only *YUC1,4,6* were expressed (*Figure 3K*, *Figure 3—figure supplement 3A–F*). While *YUC6* showed a very weak expression in the CZ, both *YUC1* and *YUC4* are expressed in the L1 layer on the lateral sides of the SAM/flower boundary from P$_3$ for *YUC4* (*Figure 3K*) and P$_4$ for *YUC1* (*Figure 3—figure supplement 3A and D*; *Cheng et al., 2006*). From P$_4$, *YUC4* expression extends over the entire epidermis of flower primordia. This is coherent with genetic and other expression data (*Supplementary file 1*; *Cheng et al., 2006*; *Armezzani et al., 2018*). In addition, *yuc1yuc4* loss-of-function mutants show severe defects in SAM organ positioning and size (*Shi et al., 2018*; *Pinon et al., 2013*; *Figure 3L* and *Figure 3—figure supplement 3O–U*). Taken with the organization of PIN1 polarities, these results suggest that P$_3$-P$_5$ are auxin production centers for the SAM that regulate phyllotaxis and that PIN1 polarity organization allows for pumping auxin away from these production centers and towards the meristem.

In conclusion, our results suggest a scenario in which auxin distribution depends on high concentrations of auxin at the center of the SAM, and also at P$_{-1}$ and P$_0$, acting as flux attractors and on auxin production primarily in P$_3$-P$_5$ (*Figure 3M*).

## The role of time in transcriptional responses to auxin

To assess quantitatively whether and how the spatio-temporal distribution of auxin is interpreted in the SAM, we next introduced the synthetic auxin-induced transcriptional reporter DR5 (*Friml et al., 2003*; *Sabatini et al., 1999*; *Ulmasov, 1997*) driving mTurquoise2 into the qDII/*CLV3* reporter line (*Figure 4A–D*). Cells expressing DR5 closest to the CZ were robustly positioned at an average distance of 32 µM ± 7 (SD) from the center. This corresponds to a distance at which the intensity of *CLV3* reporter expression is less than 5% of its maximal value (*Figure 4—figure supplement 1A*). The distance from the center at which transcription can be activated by auxin is thus defined with a near-cellular precision.

To obtain a global vision of how auxin-controlled transcription is related to auxin concentration, we performed a Principal Component Analysis (PCA) using quantified levels of DR5, auxin and *CLV3* in each nucleus of the PZ during a 10 hr time series, together with their distance from the center (*Figure 4E*). With the first two axes accounting for around 75% of the observed variability, we unexpectedly observed orthogonality between auxin input and DR5 output, clearly marking the absence of a general correlation in the SAM (*Figure 4E*, inset). This unexpected finding was confirmed by the low Pearson correlation coefficients between DR5 and auxin values at the cell-level (*Figure 4—figure supplement 1B*). We refined our analysis by focusing on the different primordia regions. We assembled all the observed couples of values (auxin, DR5), averaged over each primordium region, on a single graph (*Figure 4F*). This demonstrated that, spatially, a given auxin value does not in general determine a specific DR5 value. However, values corresponding to primordia at consecutive stages follow loop-like counter-clockwise trajectories in the auxin x DR5 space (indicated by the arrow in *Figure 4F*). Such trajectories are symptomatic of hysteresis reflecting the dependence of a system on its history. In other words, it appears that the relationship between auxin level and DR5 expression is not direct, but is affected by another factor depending on the previous developmental trajectory of each cell (determined by parameters such as genetic activity, protein content, signal exposure, chromatin state).

We then tried to identify what in this developmental history can explain the observed differences in DR5 response to auxin. We first used our reconstructed continuum of primordium development to study the joint temporal variations of DR5 and auxin within a group of cells during primordium initiation (Appendix 5). This showed that the start of auxin-induced transcription follows the build-up of auxin concentration with a delay of nearly one plastochron (*Figure 4G–H*). The duration of the observed phenomenon suggests the existence of an additional process, over and above fluorescent protein maturation (*Vernoux et al., 2011*; *Balleza et al., 2018*), that creates a significant auxin

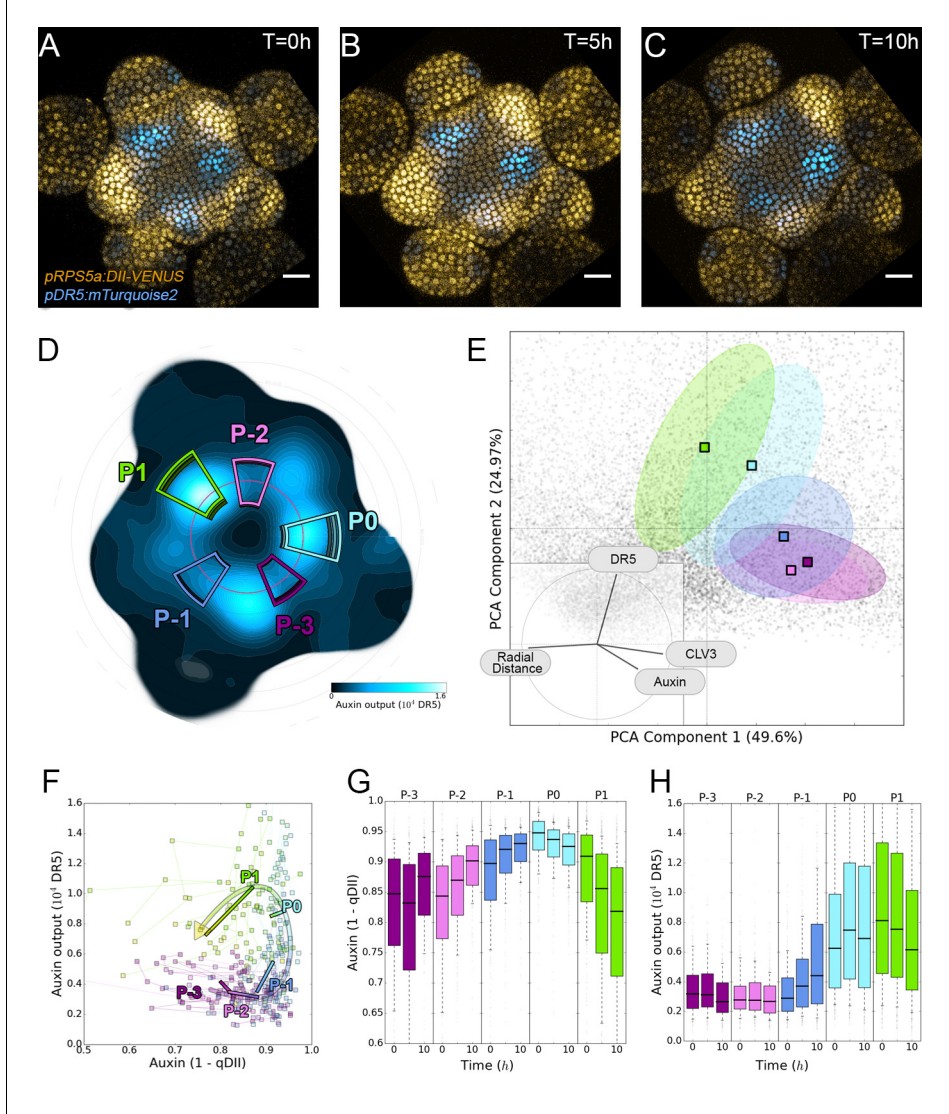

**Figure 4.** Auxin and its transcriptional output show a complex non-linear relationship. (A–C). Time-lapse images of representative projections of DII-VENUS-N7 (yellow), TagBFP nuclei (grey) and *pDR5:mTurquoise2* (cyan). Scale bars = 20 μm. (D). Quantified *DR5* expression map (black to cyan). Colored sectors show the tissue areas where primordia are located ($P_{-3}$ to $P_1$). N = 21 meristems. (E). Principal Component Analysis (PCA) showing absence of correlation (orthogonality) between auxin and *DR5* at the tissue scale. Colored ellipses show the consistent pattern associated with each primordium stage (from $P_{-2}$ to $P_1$ using the same colors as in (D)). (F). Auxin and DR5 non-linear relationship in primordia. Cells from $P_{-3}$ to $P_1$ are indicated with the color code used in (D). Lines represent the regression of auxin and DR5 medians in time. (G–H). Auxin (G) and *DR5* (H) expression in primordia. Boxplots use the same color code for primordia as in (D). N = 21 meristems.

The online version of this article includes the following figure supplement(s) for figure 4:

**Figure supplement 1.** Precision in *DR5* radial positioning and Pearson correlation analysis of DR5 expression and auxin levels.

response delay in primordium cells during development. Due to this delay, DR5 is not a direct read-out of auxin concentration, explaining the absence of correlation between DR5 expression and auxin levels in these cells.

We next wondered what could explain a time-dependent acquisition of cell competence to respond to auxin. A first possible scenario is that cells exiting the CZ proceed through different stages of activation of an auxin-independent developmental program enabling them to sense auxin

only after a temporal delay. A second possibility is that auxin controls this developmental program through a time integration process. In this scenario, cells exiting the CZ would need to be exposed to high auxin concentrations for a given time to build up an auxin transcriptional response. To test these scenarios, we treated SAMs with auxin for different periods using physiologically relevant concentrations (*Reinhardt et al., 2000*; *Figure 5A–I*). All treatments, even the shorter ones, equally degraded DII-VENUS throughout the PZ (*Figure 5—figure supplement 1A–I*). This suggests that auxin uptake was similar throughout the PZ, although we cannot totally discard that some differences exist. In the shorter auxin treatments (30' and 120'), the auxin transcriptional response was mainly enhanced at $P_{-1}$ and $P_{-2}$ and to a lesser extent at the position of the predicted $P_{-3}$ that is where cells are already being exposed to auxin (*Figure 5I*). The longer auxin treatments (300') lead to an activation of signaling in most cells in the PZ and organs, with the strongest activation being observed again at $P_{-1}$ and $P_{-2}$ but also at the predicted azimuth for $P_{-3}$, $P_{-4}$ and $P_{-5}$ (*Figure 5H–I*). We could further show that a 300' treatment with a lower auxin concentration (200 nM) activated signaling similarly (at $P_{-1}$) or more strongly (at $P_{-2}$, $P_{-3}$, $P_{-4}$ and $P_{-5}$) than a 120' 1 mM auxin treatment. Conversely, a 120' treatment with higher auxin concentration (5 mM) lead to an activation of signaling almost as strongly as a 300' 1 mM treatment at $P_{-1}$, although the activation was lower at $P_{-2}$ (*Figure 5I*). In all treatments, no significant effect was detected at $P_0$, consistent with the fact that DR5 activation is already maximal at this stage of development (*Figure 4*). We next treated *pinoid* (*pid*) mutant SAMs with exogenous auxin. *pid* mutants are strongly affected in polar auxin transport and in aerial organ production (*Reinhardt et al., 2003a*; *Friml et al., 2004*; *Christensen et al., 2000*). DR5 expression was low and radially uniform in *pid* SAMs, suggesting a uniform auxin distribution (*Figure 5J*; *Friml et al., 2004*). When treated with 1 mM auxin, DR5 could be activated in all cells of the periphery of the SAM (suggesting an uptake throughout the PZ as in the wild-type) only with a 300' treatment, while a 120' treatment had only a weak effect (*Figure 5J–M* and *Figure 5—figure supplement 2A–C*). This indicates that, even with the reduced complexity in PZ patterning of the *pid* mutant (*Friml et al., 2004*), activation of auxin signaling is still dependent on the time of exposure to auxin in all cells surrounding the CZ. Taken together, our observations support the second scenario, with the activation of signaling being a function of both time of exposure to auxin and auxin concentration. Conversely, our results are incompatible with the first scenario, where the capacity of the cells to respond to auxin is intrinsic and is not dependent upon auxin exposure time. Notably, the results with *pid* SAMs suggest that all cells at the SAM periphery show no intrinsic differences in their capacity to respond to auxin, in agreement with published data (*Reinhardt et al., 2003a*; *Heisler et al., 2005*; *Smith et al., 2006a*). Our results thus support the hypothesis that temporal integration of auxin concentration is required for downstream transcriptional activation in the SAM.

The Auxin Response Factor (ARF) ETTIN (ETT/ARF3) plays an important role in promoting organogenesis in the SAM (*Wu et al., 2015*; *Chung et al., 2019*). Despite the fact that ETT is a non-canonical ARF, genetic data indicate that it acts together with ARF4 and MONOPTEROS/ARF5 to promote organogenesis at the SAM. We found that in a loss-of function *ett3* mutant the expression of DR5 was restricted to only 2–3 cells at sites of organogenesis, an observation consistent with a role for ETT in promoting organogenesis. In addition, a 300' 1 mM auxin treatment did not induce DR5 in the SAM (*Figure 5N–Q* and *Figure 5—figure supplement 2D–E*). Auxin signaling and ARF3 in particular have been shown to act by modifying acetylation of histones (*Wu et al., 2015*; *Chung et al., 2019*; *Long et al., 2006*). Pharmacological inhibition of histone deacetylases (HDACs) alone was able to trigger concomitant activation of DR5 at $P_0$ and $P_{-1}$ sites in the SAM (*Figure 5—figure supplement 1Q–S*). Taken together, these results suggest that auxin signal integration likely depends on a functional ARF-dependent auxin nuclear pathway.

Phyllotaxis is perturbed in *ett* mutant SAMs (*Figure 5—figure supplement 1M–P*; *Simonini et al., 2017*). Our results thus suggest that a perturbation of the temporal reading of auxin information can result in phyllotaxis defects. Supporting this idea, we also found that daily exogenous auxin treatments at the SAM affected phyllotaxis and that the efficiency of the treatment increased with both auxin concentration and treatment length. This was particularly evident for 30' and 120' treatments (*Figure 5—figure supplement 1J–L*). 300' treatments were less efficient at higher auxin concentrations, possibly due to compensation mechanisms. These results suggest that temporal integration of auxin information at the SAM is essential for phyllotaxis.

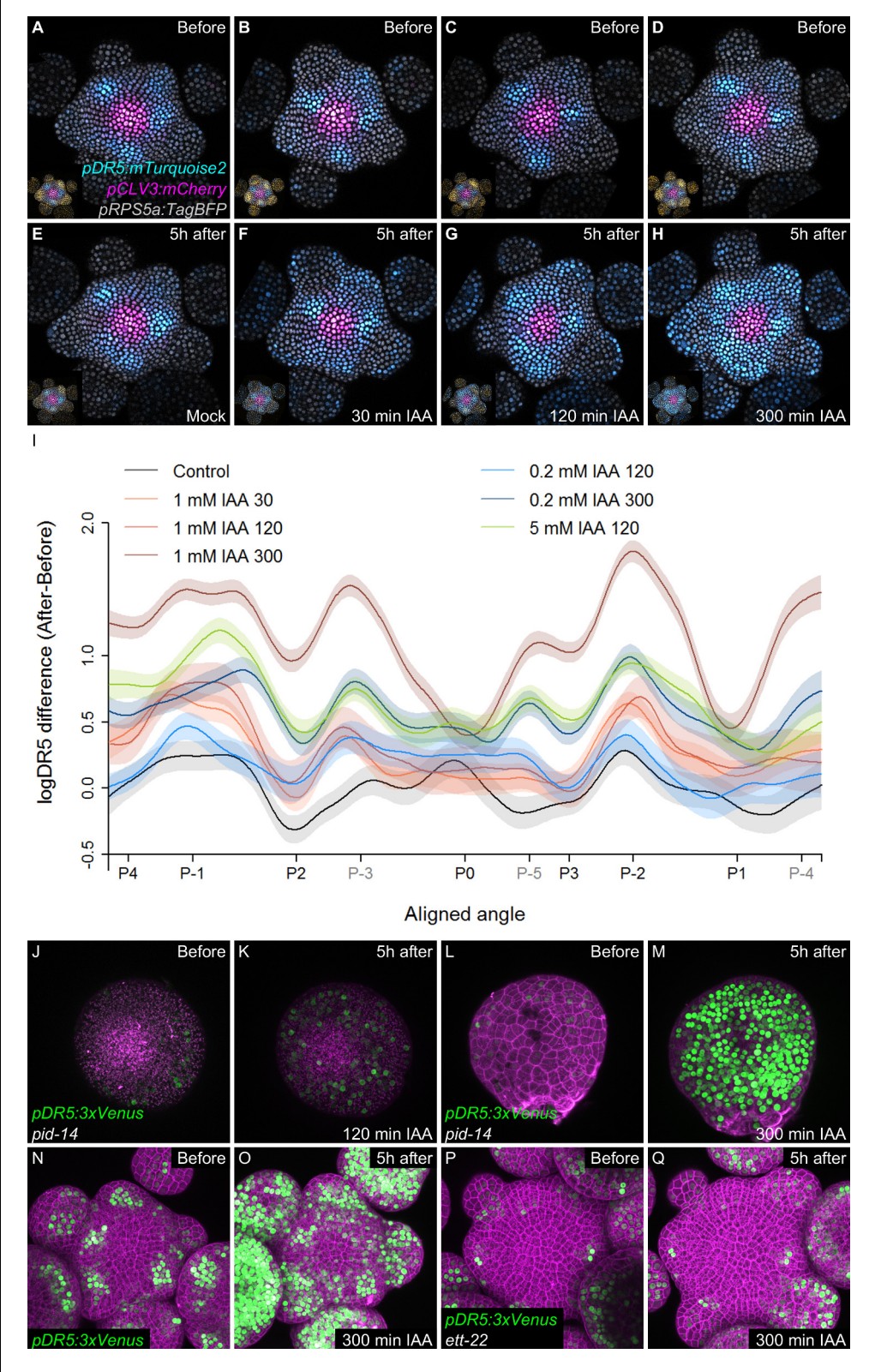

**Figure 5.** Temporal integration of auxin concentration regulates transcription. (**A–I**) Activation of the *DR5* reporter with different concentrations of auxin and durations of treatments. *pDR5:mTurquoise2* expression before auxin treatment (**A–D**) and 5 hr after the end of the auxin treatment: mock (**E**) or 1 mM IAA treatment for 30' (**F**), 120' (**G**) or 300' (**H**). *pDR5:mTurquoise* (cyan), *TagBFP* driven by *pRPS5a* (gray) and *pCLV3:mCherry* (magenta) labelled

*Figure 5 continued on next page*

*Figure 5 continued*

nuclei are shown. Inset: DII-VENUS-N7 (yellow) from the same meristem. Quantification of *DR5* expression in the PZ after auxin treatments. (I). Average *DR5* response in the PZ with different auxin concentrations and treatment durations. Confidence intervals (shade) and regression (line) shows log(*DR5*) expression along the circumference of the PZ (aligned angle) of control (gray) or IAA (color) treated meristems. For simplicity only the angular position of primordia are indicated (in grey, presumptive positions). (J–M). Transcriptional response to auxin treatment of different durations in *pid-14*. *pid-14 pDR5:3xVENUS* SAM treated with IAA for 120' (J,K) or 300' (L,M) are shown. (N–Q). Transcriptional response to a 300' auxin treatment in *ett* mutants. Control Col-0 *pDR5:3xVENUS-N7* (N,O) and *ett-22/arf3 pDR5:3xVENUS-N7* (P,Q) meristems treated with auxin for 300'.

The online version of this article includes the following figure supplement(s) for figure 5:

**Figure supplement 1.** Time integration in the control of transcription in response to auxin.

**Figure supplement 2.** The effect of auxin treatments of different durations on auxin transcriptional responses in *pid-14* and *ett-22/arf3*.

## Discussion

In a recent modeling study, a stochastic induction of organ initiation based on temporal integration of morphogenetic information was proposed (*Refahi et al., 2016*). Here we provide evidence that organ initiation in the SAM is indeed dependent on temporal integration of the auxin signal. Our quantitative analysis of the dynamics of auxin distribution and response supports a scenario in which rhythmic organ initiation at the SAM is driven by the combination of high-precision spatio-temporal graded distributions of auxin with the use of the duration of cell exposure to auxin, to temporally differentiate sites of organ initiation (*Figure 6*). Importantly our results suggest that a time integration mechanism is essential for rhythmic organ patterning in the SAM since auxin-based spatial information pre-specifies several sites of organ initiation and is thus unlikely to provide sufficient information (*Video 1*). Whether temporal integration of auxin information exists in other tissues remains to be established.

We provide evidence that temporal integration of the auxin signal likely requires the effectors of the auxin signaling pathway. Activation of transcription downstream of auxin by ARFs relies on chromatin remodeling, increasing the accessibility of ARF targets and possibly allowing for the recruitment of histone acetyltransferases (*Wu et al., 2015*), together with the release of histone deacetylases (HDACs) from target loci through degradation of Aux/IAA repressors (*Long et al., 2006*). Chromatin state change is one mechanism that allows the temporal integration of signals in eukaryotes, including plants (*Angel et al., 2011*; *Coda et al., 2017*; *Nahmad and Lander, 2011*;

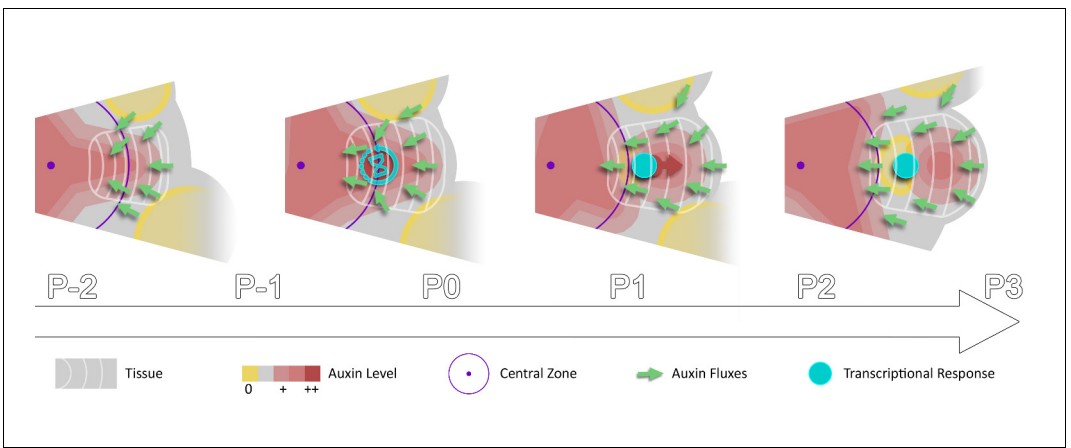

**Figure 6.** Spatio-temporal gradients of auxin translate into rhythmic organ patterning through time integration. A maximum of auxin protrudes from a high auxin concentration zone at the CZ faster than the cell radial movement. Cells exiting the CZ that are exposed to high auxin levels progressively acquire competence for transcriptional response. This leads to activation of transcriptional responses with a delay close to the system period, the plastochron.

*Sun et al., 2009*). It is thus plausible that time integration of the auxin signal in cells leaving the CZ is set by progressive acetylation of histones triggered by ARFs at their target loci. As chromatin deacetylation also represses auxin signaling in the CZ (*Ma et al., 2019*), balancing the acetylation status of ARF target loci could provide a mechanism to tightly link stem cell maintenance to differentiation by precisely positioning organ initiation at the boundary of the stem cell niche, while at the same time allowing sequential organ initiation. Temporal integration might as well rely on mechanisms that fine-tune the intracellular distribution of auxin, such as auxin metabolism but also intracellular transport (*Sauer and Kleine-Vehn, 2019*). Determining how different mechanisms might act in parallel to provide a capacity to activate target genes as a function of auxin concentrations over time will require further analyses. It will notably be important to determine whether other ARF than ARF3 act in the temporal integration of auxin.

The existence of high definition spatio-temporal auxin gradients suggests that as for several morphogens in animals (*Nahmad and Lander, 2011*; *Dessaud et al., 2007*; *Scherz et al., 2007*; *Maden, 2002*) the robustness of SAM patterning results from highly reproducible spatio-temporal positional information. Our results indicate that auxin maxima could first emerge from the CZ at the confluence of centripetal auxin fluxes. Confluences creating auxin maxima would at the same time divert fluxes away from areas where auxin minima appear (*Figure 3M*). Our analysis raises the question of how auxin transport could generate this high definition signal distribution and whether the different models that have been proposed can explain this distribution (*Bainbridge et al., 2008*; *Stoma et al., 2008*; *Jonsson et al., 2006*; *Smith et al., 2006a*; *van Berkel et al., 2013*). Further analysis of the spatio-temporal control of auxin distribution needs also to consider that early developing flowers act as auxin production centers. These flowers could not only provide a memory of the developmental pattern through lateral inhibition but also contribute positively to a self-sustained auxin distribution pattern by providing auxin to the system (*Figure 3M*). Finally, our work indicates that the stem cell niche could act as a system-wide organizer of auxin transport, consistent with previous work (*de Reuille et al., 2006*). This could provide another layer of regulation tightly coordinating differentiation with the presence of a largely auxin-insensitive stem cell niche (*Vernoux et al., 2011*; *Ma et al., 2019*).

# Materials and methods

## Key resources table

| Reagent type (species) or resource | Designation | Source or reference | Identifiers | Additional information |
|---|---|---|---|---|
| Genetic reagent *Arabidopsis thaliana* | *pPIN1:PIN1-GFP (Col-0)* | *Benková et al., 2003* | | |
| Genetic reagent *Arabidopsis thaliana* | *pCLV3:mCherry-NLS (Col-0)* | *Pfeiffer et al., 2016* | | |
| Genetic reagent *Arabidopsis thaliana* | *pYUC1-11:GFP (Col-0)* | *Liu et al., 2016*; *Robert et al., 2013* | | |
| Genetic reagent (*Arabidopsis thaliana*) | *yuc1 yuc4/+ pDR5 rev::GFP (Col-0)* | *Robert et al., 2013* | | |
| Genetic reagent (*Arabidopsis thaliana*) | *ett-22 (Col-0)* | *Pekker et al., 2005* | | |
| Genetic reagent (*Arabidopsis thaliana*) | *pid-14 (Col-0)* | *Huang et al., 2010* | | |
| Genetic reagent *Arabidopsis thaliana* | *pRPS5a:DII-VENUS-N7-p2A-TagBFP-SV40 (Col-0)* | This study | qDII | Request to teva.vernoux@ens-lyon.fr |
| Genetic reagent *Arabidopsis thaliana* | *pDR5rev:2x-mTurquoise 2-SV40 (Col-0)* | This study | | Request to teva.vernoux@ens-lyon.fr |
| Chemical compound, drug | Trichostatin A | Invivogen | met-tsa-1 | 0.005 mM |
| Chemical compound, drug | Indole-3-acetic acid sodium salt | Sigma-Aldrich | I5148 | 0.2, 1.0, 5.0 mM |

*Continued on next page*

*Continued*

| Reagent type (species) or resource | Designation | Source or reference | Identifiers | Additional information |
|---|---|---|---|---|
| Software, algorithm | RStudio | *RStudio Team, 2015* | RRID:SCR_000432 | |
| Software, algorithm | Image J | | RRID:SCR_003070 | https://imagej.net |
| Software, algorithm | NumPy | | RRID:SCR_008633 | http://www.numpy.org |
| Software, algorithm | SciPy | *Virtanen et al., 2020* | RRID:SCR_008058 | http://www.scipy.org |
| Software, algorithm | VTK | *Schroeder et al., 2006* | RRID:SCR_015013 | http://www.vtk.org |
| Software, algorithm | scikit-image | *van der Walt et al., 2014* | | http://scikit-image.org |
| Software, algorithm | sam_spaghetti | This study *Cerutti et al., 2020* | | https://gitlab.inria.fr/mosaic/publications/sam_spaghetti/ |
| Other | Propidium iodide solution | Sigma-Aldrich | P4864 | 0.1 mM |

## Plant material and growth conditions

Seeds were directly sown in soil, vernalized at 4 ˚C, and grown for 24 days at 21 ˚C under long day condition (16 hrs light, LED 150µmol/m$^2$/s). Shoot apical meristems from inflorescence stems with a length between 0.5 and 1.5 cm were dissected and cultured in vitro as described in *Prunet et al. (2016)* for 16 hrs. When required, meristems were stained with 100 µM propidium iodide (PI; Merck) for 5 min. Auxin treatments were performed by immersing meristems in solutions containing indicated concentrations of indole-acetic acid (IAA) and 10 mM MES-hydrate (buffer) for indicated periods of time. Trichostatin A (TSA – Invivogen) was added to the culture medium to a final concentration of 5 µM. Meristems were cultured in TSA for 16 hrs prior to auxin treatment. For time lapses, the first image acquisition (T=0) corresponds to 2 hrs after the end of the dark period. In planta treatments were carried out on 24 day-old Col-0 plants by dropping 10 µL of IAA solution (IAA at different concentrations, 10 mM MES-hydrate and 0.01% v/v Tween-20) onto the SAM, followed by incubation for indicated lengths of time. Meristems were then washed with 100 µL of 10 mM MES buffer with 0.01% v/v Tween-20. Treatments were carried out on 5 consecutive days and perturbations in organ positioning were recorded 7 days after the end of the treatments.

Previously published transgenic lines used in this study are PIN1-GFP (*Benková et al., 2003*), pCLV3:mCherry-NLS (*Pfeiffer et al., 2016*), pYUC1-11:GFP and *yuc1 yuc4/+* pDR5rev::GFP (*Liu et al., 2016*; *Robert et al., 2013*), *ett-22* (*Pekker et al., 2005*), *pid-14* (*Huang et al., 2010*). pRPS5a:DII-VENUS-N7-p2A-TagBFP-SV40 (qDII) and pDR5rev:2x-mTurquoise2-SV40 constructs were cloned cloned using Gateway technology (Life Sciences), and transformed in *Arabidopsis thaliana* (Col-0). Stable qDII homozygous lines were then crossed with pCLV3:mCherry-NLS, pDR5rev:2x-mTurquoise2-SV40 and PIN1-GFP reporter lines.

## Microscopy

All confocal laser scanning microscopy was carried out with a Zeiss LSM 710 spectral microscope or a Zeiss LSM700 microscope. Multitrack sequential acquisitions were always performed using the same settings (PMT voltage, laser power and detection wavelengths) as follows: VENUS, excitation wavelength (ex): 514 nm, emission wavelength (em): 520–558 nm; mTurquoise2, ex: 458 nm, em: 470–510 nm; EGFP, ex: 488 nm, em: 510–558 nm; TagBFP, ex:405 nm, em: 430–460 nm; mCherry, ex: 561 nm, em: 580–640 nm; propidium iodide, ex: 488, em: 605–650 nm.

Scanning electron microscopy of meristems were carried out using a HIROX SH-3000 microscope.

Time lapses for Super Resolution Radial Fluctuation (SRRF) imaging were performed on an inverted Zeiss microscope (AxioObserver Z1, Carl Zeiss Group, http://www.zeiss.com/) equipped with a spinning disk module (CSU-W1-T3, Yokogawa, www.yokogawa.com) and a Prime95B SCMOS camera (https://www.photometrics.com) using a 63x Plan-Apochromat objective (numerical aperture 1.4, oil immersion), pixel size 175 nm or a 100x Plan-Apochromat objective (numerical aperture 1.46,

oil immersion), pixel size 110 nm. GFP was excited with a 488 nm laser (150 mW) and fluorescence emission was filtered using a 525/50 nm BrightLine single-band bandpass filter (Semrock, http://www.semrock.com/). PI was excited with a 561 nm laser (80 mW) and fluorescence emission was filtered using a 609/54 nm BrightLine single-band bandpass filter (Semrock, http://www.semrock.com/). To obtain high resolution images, 200 frames were acquired with 50% laser power and 70 ms exposure time using Stream Acquisition mode. The green and red channels were acquired sequentially. For drift correction, 200 nm TetraSpeck beads (Life Technologies) were added to samples. Images were processed using the NanoJ-SRRF plugin (*Gustafsson et al., 2016*) with the following parameters: Ring Radius 0.5, Radiality Magnification 5, Axes in ring 6, Temporal Analysis TRPPM. SRRF time-lapses were produced by running SRRF analysis on groups of 50 frames. If aberrant PSF of Tetraspeck beads were observed, datasets were discarded.

## Quantification and statistical analysis

All confocal images were pre-processed using the ImageJ software (http://rsbweb.nih.gov/ij/) for the delimitation of the region of interest. Then the CZI image files were processed using a computational pipeline relying on the numpy, scipy, pandas, czi_file Python libraries, as well as other custom libraries. Extensive details about the computational methods and algorithms are given in Appendix 3, 4 and 5.

Given the non- linear positive DR5 response, the raw values were logarithmically transformed in order to obtain a symmetric distribution of the noise. Nadaraya-Watson estimates and confidence intervals were then calculated with a confidence level of 95% in the R environment (*RStudio Team, 2015*). The boxplots displayed in the article were obtained by computing the median (central line), first and third quartiles (lower and upper bound of the box) and first and ninth deciles (lower and upper whiskers) using the R environment or numpy percentile function and rendered using the matplotlib Python library. Linear regressions were performed using the polyfit and polyval numpy functions. P-values were obtained using the scipy anova implementation in the f_oneway function. Principal component analysis was performed using the PCA implementation from the scikit-learn Python library. All data were generated with at least three independent sets of plants.

## Data and software availability

All experimental data and quantified data that support the findings of this study are available from the corresponding authors upon request.

Generic quantitative image and geometry analysis algorithms are provided in Python libraries timagetk, cellcomplex, tissue_nukem_3d and tissue_paredes (https://gitlab.inria.fr/mosaic/) made publicly available under the CECILL-C license. Specific SAM sequence alignment and visualization algorithms are provided in a separate project providing Python scripts to perform the complete analysis pipelines (*Cerutti, 2020*; copy archived at https://github.com/elifesciences-publications/sam_spaghetti).

## Acknowledgements

We thank Fabrice Besnard and the members of the SIGNAL team for insightful discussions; Antoine Larrieu for helping with RNA-Seq analysis; Hélène Robert-Boisivon for the YUC transcriptional and mutant lines; Sophie Ribes for her contribution to the nuclei image processing pipeline; Gwyneth Ingram, Olivier Hamant, Arezki Boudaoud and Jan Traas for feedback on the manuscript. We acknowledge the contribution of SFR Biosciences (UMS3444/CNRS, US8/Inserm, ENS de Lyon, UCBL) facilities PLATIM, for assistance with microscopy; of the PSMN (Pôle Scientifique de Modélisation Numérique) of the ENS de Lyon for providing us access to their computing resources. This work was supported by Human Frontier Science Program organization (HFSP) Grant RPG0054-2013 to TV and CG, ANR-12-BSV6-0005 grant (AuxiFlo) and ANR-18-CE12-0014-02 (ChromAuxi) to TV and DFG FOR2581 to JUL.

## Additional information

### Funding

| Funder | Grant reference number | Author |
|---|---|---|
| Agence Nationale de la Recherche | ANR-12-BSV6-005 | Teva Vernoux |
| Deutsche Forschungsgemeinschaft | FOR2581 | Christian Wenzl<br>Jan U Lohmann |
| Human Frontier Science Program | RPG0054-2013 | Christophe Godin<br>Teva Vernoux |
| Agence Nationale de la Recherche | ANR-18-CE12-0014-02 | Teva Vernoux |

The funders had no role in study design, data collection and interpretation, or the decision to submit the work for publication.

### Author contributions

Carlos S Galvan-Ampudia, Guillaume Cerutti, Conceptualization, Resources, Data curation, Software, Formal analysis, Supervision, Validation, Investigation, Visualization, Methodology, Writing - original draft, Project administration, Writing - review and editing; Jonathan Legrand, Resources, Software, Formal analysis, Validation, Investigation, Methodology; Géraldine Brunoud, Raquel Martin-Arevalillo, Investigation, Methodology; Romain Azais, Formal analysis, Validation; Vincent Bayle, Formal analysis, Investigation; Steven Moussu, Methodology; Christian Wenzl, Resources; Yvon Jaillais, Conceptualization, Methodology, Writing - review and editing; Jan U Lohmann, Resources, Writing - review and editing; Christophe Godin, Teva Vernoux, Conceptualization, Supervision, Funding acquisition, Writing - original draft, Project administration, Writing - review and editing

### Author ORCIDs

Carlos S Galvan-Ampudia https://orcid.org/0000-0002-4779-3568
Guillaume Cerutti https://orcid.org/0000-0003-2003-9335
Yvon Jaillais http://orcid.org/0000-0003-4923-883X
Jan U Lohmann http://orcid.org/0000-0003-3667-187X
Christophe Godin https://orcid.org/0000-0002-1202-8460
Teva Vernoux https://orcid.org/0000-0002-8257-4088

### Decision letter and Author response

Decision letter https://doi.org/10.7554/eLife.55832.sa1
Author response https://doi.org/10.7554/eLife.55832.sa2

## Additional files

### Supplementary files

- Supplementary file 1. RNAseq expression of YUCCA genes at the SAM.

- Transparent reporting form

### Data availability

All data generated or analyzed during this study are included in the manuscript and supporting files. Generic quantitative image and geometry analysis algorithms are provided in Python libraries timagetk, cellcomplex, tissue_nukem_3d and tissue_paredes (https://gitlab.inria.fr/mosaic/) made publicly available under the CECILL-C license. Specific SAM sequence alignment and visualization algorithms are provided in a separate project providing Python scripts to perform the complete analysis pipelines (https://gitlab.inria.fr/mosaic/publications/sam_spaghetti.git, copy archived at https://github.com/elifesciences-publications/sam_spaghetti).

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

## Appendix 1

### Definition of a conceptual frame for models and analysis

The shoot apical meristem dynamically produces organ primordia, issuing from a central dome-shaped area, into a complex spatio-temporal pattern that is referred as phyllotaxis. In an abstract view of this structure, the meristem can be seen as dynamic collection of organ primordia characterized by their spatial trajectory relatively to the central zone (CZ) and by the evolution of their inner state. We propose a formal definition of such a system, which we name a 'phyllotactic dynamical system'.

### Definition 1 (phyllotactic dynamical system)

Let a phyllotactic dynamical system $\mathcal{S}$ be a finite set of primordia considered over a time interval $\mathcal{T} = [t_{\min}, t_{\max}] \subset \mathbb{R}$ and such that:

- At every time $t \in \mathcal{T}$, each primordium $p \in \mathcal{S}$ is characterized by its current state $\{\tau_p(t), x_p(t), y_p(t)\}$ where:
  - $\tau_p \in [\tau_{\min}, \tau_{\max}] \subset \mathbb{R}$ is called the developmental state of the primordium.
  - $x_p = [r_p(t), \theta_p(t), z_p(t)] \in \mathbb{R}^3$ is the spatial position of the primordium in a cylindrical coordinate system, the origin of which is called the center of the system.
  - $y_p(t) \in \mathbb{R}^d$ is a vector describing the physiological state of the primordium.
- The developmental state $\tau_p$ is a continuous strictly increasing function of time. Note that consequently, for every $p \in \mathcal{S}$, $\tau_p$ is a bijection between $\mathcal{T}$ and $\tau_p(\mathcal{T})$.
- In the case where $0 \in \tau_p(\mathcal{T})$, the time $t_{0,p} \overset{\text{def}}{=} \tau_p^{-1}(0)$ is called the initiation time of the primordium $p$.
- The spatial position and the physiological state are conditioned by the developmental state of the primordium in such way that:

$$\begin{cases} \exists X : [\tau_{\min}, \tau_{\max}] \longrightarrow \mathbb{R}^3 & | \, \forall p \in \mathcal{S}, \exists X_p \in \mathbb{R}^3 \, | \, \forall t \in \mathcal{T}, x_p(t) = X_p + X(\tau_p(t)) \\ \exists Y : [\tau_{\min}, \tau_{\max}] \longrightarrow \mathbb{R}^d & | \, \forall p \in \mathcal{S}, \forall t \in \mathcal{T}, y_p(t) = Y(\tau_p(t)) \end{cases} \quad (1)$$

- $\mathcal{S}$ is equipped with a strict total order denoted $<$ that verifies:

$$\forall p, q \in \mathcal{S}, p < q \Longrightarrow \forall t \in \mathcal{T}, \tau_p(t) < \tau_q(t) \quad (2)$$

This definition reflects the idea that for any primordium, there exists an underlying physiological state, a hidden variable that determines all processes, both geometrical and physiological, that characterizes primordium development. This state can be used to rank the different organs among them, and to run through the sequence of primordia in the order of their respective development. It is actually more common to refer to primordia by their integer rank in this developmental order:

### Property 1

There exists a morphism between $(\mathcal{S}, <)$ and $(\mathbb{Z}, <)$, and we can use it to denote the consecutiveness relationship in the strict total order of $\mathcal{S}$ by:

$$\nexists r \in \mathcal{S} \, | \, p < r < q \overset{\text{def}}{\Longleftrightarrow} q = p + 1 \quad (3)$$

In a phyllotactic system, the notion of plastochron refers to the time elapsed between two consecutive organ initiations. However it is common to speak about the plastochron as a characteristic of the system when this duration does not vary over time:

## Definition 2 (plastochron)

We say that a phyllotactic dynamical system $\mathcal{S}$ has a plastochron $T \in \mathbb{R}$ if two consecutive primordia in the strict total order of $\mathcal{S}$ always have their initiation times separated by a time interval of length $T$:

$$\forall p, p+1 \in \mathcal{S}, t_{0,p+1} = t_{0,p} - T \tag{4}$$

A stronger assumption that is generally made on a phyllotactic system is that it develops in a steady regime, meaning that it maintains a constant rate of development. This translates into linear functions for the developmental states of primordia with a common strictly positive slope. If we add the existence of a plastochron, then this slope is naturally equal to the inverse of the plastochron:

## Definition 3 (steady development)

We say that a system $\mathcal{S}$ with a plastochron $T$ has a steady development if all primordia in $\mathcal{S}$ have their developmental states increasing at the same constant rate $1/T$:

$$\forall p \in \mathcal{S}, \forall t \in \mathcal{T}, \tau_p(t) = \frac{1}{T}\left(t - t_{0,p}\right) \tag{5}$$

In such a regularly developing system, the plastochron constitutes the natural unit on the developmental scale of the primordia. Indeed, the corollary to the previous definition is that the developmental state of the primordia increases by one unit every plastochron/

## Property 2

In a system $\mathcal{S}$ with a steady development of plastochron $T$, all primordia increase their developmental state by 1 after a period $T$:

$$\forall p \in \mathcal{S}, \forall t \in \mathcal{T}, \tau_p(t+T) = \tau_p(t) + 1 \tag{6}$$

Another consequence is that, at the moment where a new primordium initiates, the developmental state of its immediate predecessor is exactly equal to 1. Given the steadiness of the system, this gap of one developmental unit is actually maintained throughout the evolution of primordia.

## Property 3

In a system $\mathcal{S}$ with a steady development of plastochron $T$, two consecutive primordia in the strict total order of $\mathcal{S}$ always have their developmental states separated by 1:

$$\forall p, p+1 \in \mathcal{S}, \forall t \in \mathcal{T}, \tau_{p+1}(t) = \tau_p(t) + 1 \tag{7}$$

The fact that, in such a system, the primordia are all regularly staged in terms of developmental time allows to refer to them unambiguously by an integer rank from the lastly initiated primordium. Due to the previous properties, considering for each primordium the closest integer to its developmental state ensures a one-to-one mapping of primordia with a series of consecutive integers. This rank, that will remain constant during a period of one plastochron, can conversely be seen as a developmental stage through which all primordia will pass, one after the other.

## Property 4 (developmental stage)

In a system $\mathcal{S}$ with a steady development of plastochron $T$, at any time $t \in \mathcal{T}$, the rounding function of the developmental state:

$$\mathbb{P}(\bullet, t) : \mathcal{S} \longrightarrow [\![\mathbb{P}_{\min}(t), \mathbb{P}_{\max}(t)]\!] \subset \mathbb{Z}$$
$$p \longmapsto \mathbb{P}(p, t) = \left\lfloor \tau_p(t) + \tfrac{1}{2} \right\rfloor \tag{8}$$

is an isomorphism. We call $\mathbb{P}(p, t)$ the developmental stage of primordium $p$ at time $t$. If $\mathbb{P}(p, t) = k \in \mathbb{Z}$, we say that the primordium $p$ has the label $\mathbb{P}_k$ at time t.

Intuitively, consecutive primordia should find themselves in consecutive developmental stages. The definition of developmental stage as the rounding of developmental state, and the fact that there exists a constant gap of 1 between developmental states of consecutive primordia ensures this natural property:

## Property 5

In a system $\mathcal{S}$ with a steady development of plastochron $T$, two consecutive primordia in the strict total order of $\mathcal{S}$ always have consecutive developmental stages:

$$\forall p, p + 1 \in \mathcal{S}, \forall t \in \mathcal{T}, \mathbb{P}(p + 1, t) = \mathbb{P}(p, t) + 1 \tag{9}$$

The isomorphism between a steady phyllotactic system and a subset of consecutive integers allows to simplify the notations for the ranking of the primordia. If it was natural to denote $p + 1$ the predecessor of primordium $p$, it is now possible to extend the notation to gaps of more than one unit, with an integer number that reflects the actual gap in developmental stages between two primordia.

## Definition 4

In a system $\mathcal{S}$ with a steady development of plastochron $T$, we can extend the notation of the consecutiveness relationship in the strict total order of $\mathcal{S}$ using the isomorphism $\mathbb{P}(\bullet, t)$ to identify the primordia by their relative developmental stages. We write this relationship as follows:

$$\forall p, q \in \mathcal{S}, \quad \exists k \in \mathbb{Z} \mid \forall t \in \mathcal{T}, \mathbb{P}(q, t) = \mathbb{P}(p, t) + k \quad \overset{\text{def}}{\Longleftrightarrow} \quad q = p + k \tag{10}$$

In addition to its intrinsically regular dynamics, an ideal phyllotactic system is characterized by a geometrical regularity, and the formation of spiral-like patterns. The spirals issue from the successive emergence of primordia at evenly spaced angular locations combined with an identical radial motion. In our conceptual frame, the geometrical arrangement of primordia is represented by the vectors $x_p$ of cylindrical coordinates, for which there is a common component that depends only of developmental state, and a constant part $X_p$ that depends of the considered primordium. For the system to be considered regular from a geometrical point of view, these primordium-dependent components have to follow a rigorous angular pattern, with a constant divergence angle $\alpha$ that separates two consecutive primordia.

## Definition 5 (regular phyllotaxis)

We say that a system $\mathcal{S}$ with a steady development of plastochron $T$ has a regular phyllotaxis of divergence angle $\alpha$ if the constant parts of spatial positions of two consecutive primordia in the strict total order of $\mathcal{S}$ only differ by a rotation of angle $\alpha$ around the vertical axis:

$$\forall p, p + 1 \in \mathcal{S}, \quad X_{p+1} = R_z(\alpha) \cdot X_p = X_p + \begin{pmatrix} 0 \\ \alpha \\ 0 \end{pmatrix} \tag{11}$$

If such a regularity property is achieved, then the system becomes highly auto-similar, so that a rotation of angle $\alpha$ corresponds to a translation in time of one plastochron $T$. In terms of primordia characteristics, this means that the spatial position $x_p$ and the physiological state

$y_p$ of a primordium will be strictly identical, after one plastochron, to those of its predecessor, only up to a rotation of angle $\alpha$.

## Property 6 (spatio-temporal periodicity)

In a system $\mathcal{S}$ with a steady development of plastochron $T$ and a regular phyllotaxis of divergence angle $\alpha$, the system verifies at all times the following spatio-temporal periodicity equation:

$$\forall p, p+1 \in \mathcal{S}, \forall t \in \mathcal{T}, \begin{cases} x_{p+1}(t) & = R_z(\alpha) \cdot x_p(t+T) \\ y_{p+1}(t) & = y_p(t+T) \end{cases} \tag{12}$$

Phyllotaxis regularity offers a way to access the ranking of primordia simply by looking at their spatial positions. If the divergence angle is such that two different primordia can not be aligned on the same direction, then the angular positions alone can be enough to provide a robust ranking of organ primordia.

## Definition 6 (clear regular phyllotaxis)

In a system $\mathcal{S}$ with a steady development of plastochron $T$ and a regular phyllotaxis of divergence angle $\alpha$, if $\alpha$ is not a simple fraction of $2\pi$, that isif:

$$\nexists a, b \in [\![ 1, |\mathcal{S}| ]\!] \mid \alpha = \frac{a}{b} 2\pi \tag{13}$$

we say that $\mathcal{S}$ has a clear regular phyllotaxis.

## Property 7 (ordering on a clear regular phyllotaxis)

In a system $\mathcal{S}$ with a steady development of plastochron $T$ and a clear regular phyllotaxis of divergence angle $\alpha$, then the primordia angles $\{\theta_p \mid p \in \mathcal{S}\}$ are sufficient to determine the strict total order on $\mathcal{S}$:

$$\forall p, q \in \mathcal{S}, \quad \exists t \in \mathcal{T}, \exists k \in \mathbb{Z} \mid \theta_q(t) - \theta_p(t) = k\alpha[2\pi] \implies q = p + k \tag{14}$$

With this conceptual frame in mind, we can define thoroughly the general problem addressed when labeling the primordia on a meristem observation, that is typically when one tries to estimate where is $\mathbb{P}_1$ and where is $\mathbb{P}_2$ on a microscopy image. In that problem, only a partial state is observed for each primordium, containing mostly its spatial position and possibly some quantified features. Using this information only, the goal is to stage the visible primordia, by affecting them a label that is as close as possible to their actual developmental state, in such way that if the method is used on different observations, the primordia assigned the same label really have close actual developmental states.

## Problem 1 (assignation of developmental stages)

Given a system $\mathcal{S}$, observed at $K_t$ discrete time points $\{t_i \mid i \in [\![ 0, K_t ]\!]\}$ in which, for every $p \in \mathcal{S}$ and for every $t_i$, only a partially observed state $\{\tilde{x}_p(t_i), \tilde{y}_p(t_i)\}$ is available;

Find for every $t_i \mid i \in [\![ 0, K_t [\![$ an estimated developmental stage function $\widehat{\mathbb{P}}(\bullet, t_i)$ that verifies:

$$\forall p, p+1 \in \mathcal{S}, \widehat{\mathbb{P}}(p+1, t_i) = \widehat{\mathbb{P}}(p, t_i) + 1 \tag{15}$$

and that minimizes the average staging error of the primordia:

$$\epsilon_{\mathbb{P}} \frac{1}{|\mathcal{S}|} \sum_{p \in \mathcal{S}} \epsilon_{\mathbb{P}}(p)$$

$$\text{where} \quad \epsilon_{\mathbb{P}}(p) = \frac{1}{K_t} \sum_{i=0}^{K_t - 1} \left| \widehat{\mathbb{P}}(p, t_i) - \tau_p(t_i) \right| \tag{16}$$

Now, if we make the assumption that the system shows the regularity properties detailed above (steady development and clear regular phyllotaxis of known divergence angle $\alpha$), the assignation problem can be made much simpler. Provided the system is observed over a time frame such that there is always one primordium labelled $\mathbb{P}_k$, a solution to the problem can be found by identifying at each time point which of the primordia has its developmental stage equal to $k$. Indeed, the steady development property ensures its uniqueness, and the regular phyllotaxis allows to propagate the assignation by successive rotations of angle $\alpha$.

## Definition 7 ($\mathbb{P}_k$-maintaining system)

We say that a system $\mathcal{S}$ is $\mathbb{P}_k$-maintaining ($k \in \mathbb{Z}$) if at all times, there is a primordium that has the label $\mathbb{P}_k$:

$$\forall t \in \mathcal{T}, \exists p \in \mathcal{S}, \mathbb{P}(p, t) = k \tag{17}$$

## Property 8 (reduced $\mathbb{P}_k$ assignation problem in a clear regular phyllotaxis)

Let $\mathcal{S}$ be a $\mathbb{P}_k$-mantaining system with a steady development of plastochron $T$ and a clear regular phyllotaxis of divergence angle $\alpha$. The solution to the assignation of developmental stages problem can be reduced to:

Find for every $t_i \mid i \in [\![0, K_t[\![$, the primordium $p \in \mathcal{S}$ such that $\widehat{\mathbb{P}}(p, t_i) = k$.

The next question in order to solve this reduced problem is how to identify at each time point the primordium to label as $\mathbb{P}_k$ based on the observations. A prerequisite to do this is that some information in the physiological state of the primordia is sufficient to know that they are currently at stage $k$. In other terms, there must be a subset of the physiological state space that is characteristic of a primordium's developmental state $\tau$ being close to the value $k$.

## Definition 8 ($\mathbb{P}_k$-characteristic state function)

Let $\mathcal{S}$ be a system with a steady development of plastochron $T$. We say that the physiological state function $Y$ is $\mathbb{P}_k$-characteristic ($k \in \mathbb{Z}$) if there exists a value set $\Gamma_k \subset \mathbb{R}^d$ such that:

$$\forall \tau \in [\tau_{\min}, \tau_{\max}], \quad Y(\tau) \in \Gamma_k \iff |\tau - k| < \frac{1}{2} \tag{18}$$

In this case, the assignation of the label $\mathbb{P}_k$ to the primordium for which the physiological state lies in the $\mathbb{P}_k$-characteristic subset provides a solution to the assignation problem for which the error to minimize is, if not optimal, at least bounded by 1/2.

## Property 9 ($\mathbb{P}_k$-characterization solution)

Let $\mathcal{S}$ be a $\mathbb{P}_k$-mantaining system with a steady development of plastochron $T$ and a clear regular phyllotaxis of divergence angle $\alpha$ and a $\mathbb{P}_k$-characteristic state function. The reduced solution to the assignation of developmental stages problem given by:

$$\forall i \in [\![0, K_t[\![, \quad \widehat{\mathbb{P}}(p, t_i) = k \iff \tilde{y}_p(t_i) \in \Gamma_k \tag{19}$$

has a staging error $\epsilon_{\mathbb{P}}(p)$ bounded by 1/2 for every $p \in \mathcal{S}$.

When the system is observed over a relatively short time period, it might be convenient to consider, for simplicity reasons, that the primordia do not change stage, and that the morphism existing between $\mathcal{S}$ and $\mathbb{Z}$ at time $t_{\min}$ is preserved until $t_{\max}$. Actually, if we make this assumption for a system observed during less than one plastochron, we can show that the resulting staging error is again bounded by 1/2.

## Property 10 (stage stationarity condition)

Let $\mathcal{S}$ be a system with a steady development of plastochron $T$. We say that a developmental stage assignation $\widehat{\mathbb{P}}$ is stationary if it is the same at all times of observation, that is if:

$$\exists \widehat{\mathbb{P}}^* : \mathcal{S} \longrightarrow \mathbb{Z} \mid \forall p \in \mathcal{S}, \forall t \in \mathcal{T}, \widehat{\mathbb{P}}(p,t) = \widehat{\mathbb{P}}^*(p) \tag{20}$$

If $\mathcal{S}$ is observed during a time interval $\mathcal{T} = [t_{\min}, t_{\max}]$ smaller than its plastochron $T$, then there exists a stationary developmental stage assignation with a staging error bounded by 1/2 for every $p \in \mathcal{S}$:

$$t_{\max} - t_{\min} < T \implies \exists \widehat{\mathbb{P}}^* : \mathcal{S} \longrightarrow \mathbb{Z} \mid \forall p \in \mathcal{S}, \epsilon_{\mathbb{P}}(p) = \frac{1}{t_{\max} - t_{\min}} \int_{t_{\min}}^{t_{\max}} \left| \widehat{\mathbb{P}}^*(p) - \tau_p(t) \right| dt < \frac{1}{2} \tag{21}$$

This final observation leads us to consider that, when it applies to a system observed over less than a plastochron, the stage assignation problem has a stationary solution that guarantees an average staging error of at most 1. By labelling as $\mathbb{P}_k$ the primordium for which the average physiological state is characteristic of stage $k$, we can obtain a staging of all primordia with the same stage at all time points without making an error of more than one developmental unit.

## Property 11 (stationary $\mathbb{P}_k$-characterization solution)

Let $\mathcal{S}$ be a $\mathbb{P}_k$-mantaining system with a steady development of plastochron $T$ and a clear regular phyllotaxis of divergence angle $\alpha$ and a $\mathbb{P}_k$-characteristic state function. If $\mathcal{S}$ is observed during a time interval $\mathcal{T} = [t_{\min}, t_{\max}]$ smaller than its plastochron $T$, then the reduced stationary solution to the assignation of developmental stages problem given by:

$$\widehat{\mathbb{P}}^*(p) = k \iff \frac{1}{K_t} \sum_{i=0}^{K_t - 1} \tilde{y}_p(t_i) \in \Gamma_k \tag{22}$$

has a staging error $\epsilon_{\mathbb{P}}(p)$ bounded by one for every $p \in \mathcal{S}$:

$$t_{K_t - 1} - t_0 < T \implies \forall p \in \mathcal{S}, \epsilon_{\mathbb{P}}(p) < 1 \tag{23}$$

In a classical inhibitory field model of phyllotaxis, the developmental state $\tau_p = 0$ of an organ primordium $p$ corresponds to the time where an initiation is decided in the peripheral zone (PZ) and would therefore match a local spatio-temporal minimum of the global inhibition field. With the idea in mind that the depletion of auxin has very often been related to the concept of 'inhibition' from those models of phyllotaxis, we consider that the instant where initiation happens corresponds to a local spatio-temporal maximum of auxin in the meristem. In other terms a characteristic of the primordium labelled $\mathbb{P}_0$ should be that it has the maximal auxin level across the PZ.

Therefore if the local auxin maximality is the $j^{\text{th}}$ component $Y_j \in \{0, 1\}$ of the systems's primordia state function, then we consider that the function $Y$ is $\mathbb{P}_0$-characteristic with $\Gamma_0 = \mathbb{R} \times \ldots \times ]1/2, 1] \times \ldots \times \mathbb{R}$. We observed the meristems over a time interval of 10 hr, which is less that the estimated plastochron in our experimental conditions. Therefore, by Property 9, we can define a stable assignation of developmental stages to the visible primordia of bounded error by assigning the label $\widehat{\mathbb{P}}^* = 0$ to the primordium that has most often the maximal value of auxin across the PZ over the times of observation. If the meristems prove to be close enough to phyllotactic systems with a plastochron and a regular phyllotaxis,

then this first assignation will be enough to derive the developmental stages of all the other organ primordia based on their spatial positions. The method developed to perform this developmental stage assignation heuristic on experimental data is detailed in Appendix 2. Evidence for the regularity of the observed phyllotactic systems is discussed in Appendix 1.

## Appendix 2

### Effects of variability on a theoretical phyllotactic system

In this section we develop a formal study on regularity in a phyllotactic system. Notably, we wondered to which extent the apparent similarity of the observed SAMs could be informative on the level of precision in the process of organogenesis. To answer this, we simulated a sample of phyllotactic patterns assuming that i) they are all aligned with respect to the position of their $\mathbb{P}_0$ ii) their plastochrons and divergence angles are drawn from random distributions centered on a common average value. By varying the levels of noise on both angular positions and plastochrons, we assess how variability impacts the overlapping of phyllotactic patterns at the scale of a population.

Let us consider a 2D phyllotactic dynamical system $\mathcal{S}$ (see Appendix 1) formed by consecutive organ primordia observed on a temporal interval $\mathcal{T}$. At every time $t \in \mathcal{T}$, each primordium labelled $\mathbb{P}_p, p \in [\![0, p_{\max}]\!] \subset \mathbb{Z}$ is represented by its developmental stage $\tau_p$ and by its coordinates in a 2D cylindrical reference frame:

$$\mathcal{S}(t) = \{\, (\, \tau_p(t), r_p(t), \theta_p(t)\,) \mid p \in [\![0, p_{\max}]\!]\} \tag{24}$$

If we assume that the system has a plastochron $T$ and has a steady development, all primordia develop at the same constant rate $1/T$. In that case, we can derive:

$$\forall t \in \mathcal{T}, \forall p \in ]\!]0, p_{\max}]\!], \tau_p(t) = \tau_{p-1}(t) + 1 \tag{25}$$

In addition, if we consider that the system has a regular phyllotaxis with a divergence angle $\alpha$, that prmordia emerge on the contour of a central zone of radius $R$, and then move radially following an exponential motion law of coefficient $\beta$ we can write the state equations of the system as follows:

$$\forall t \in \mathcal{T}, \forall p \in [\![0, p_{\max}]\!], \begin{cases} \theta_p(t) & = \theta_0 + p \cdot \alpha \\ r_p(t) & = R \cdot e^{\beta \tau_p(t)} \end{cases} \tag{26}$$

This can be translated into incremental equations to obtain a recursive definition of the state of the system, knowing the state of the primordium labelled $\mathbb{P}_0$ at each time $t \in \mathcal{T}$:

$$\forall t \in \mathcal{T}, \forall p \in ]\!]0, p_{\max}]\!], \begin{cases} \theta_p(t) & = \theta_{p-1}(t) + \alpha \\ r_p(t) & = r_{p-1}(t)e^{\beta} \end{cases} \tag{27}$$

We set ourselves in a context where all the considered phyllotactic systems have previously been aligned on $\mathbb{P}_0$, that is where $\forall t \in \mathcal{T}, \theta_0(t) = 0$ and $\tau_0(t) \in [-0.5, 0.5]$.

We study what happens if we introduce variability into this system, by adding noise on two of the key variables of the system:

- A Gaussian noise of standard deviation $\sigma_\alpha$ on the divergence angle $\alpha$
- A Gaussian noise of normalized standard deviation $\sigma_\tau$ on the plastochron time $T$

To be more precise, we consider that the system still has a plastochron and a constant development rate $1/T$, but that the gap between the initiation times $t_{0,p-1}$ and $t_{0,p}$ of two consecutive primordia that should always be equal to $T$ is actually a random variable:

$$t_{0,p-1} - t_{0,p} \sim T \cdot \mathcal{N}(1, \sigma_\tau) \tag{28}$$

which translates into:

$$\forall t \in \mathcal{T}, \tau_p(t) - \tau_{p-1}(t) \sim \mathcal{N}(1, \sigma_\tau) \tag{29}$$

Consequently, we can formulate the recursive definition of the system as the drawing of $2p_{\max}$ random variables:

$$\forall t \in \mathcal{T}, \forall p \in ]\!]0, p_{\max}]\!], \begin{cases} \theta_p(t) - \theta_{p-1}(t) & \sim \mathcal{N}(\alpha, \sigma_\alpha) \\ r_p(t) - r_{p-1}(t) & \sim e^{\beta \mathcal{N}(1, \sigma_\tau)} \end{cases} \tag{30}$$

We simulate a population of such systems by generating $K_S$ single-time instances that are all aligned on $\mathbb{P}_0$. To do so, we draw for each instance a value for $\tau_0$ from a uniform distribution in [-0.5, 0.5], then use the initial values $(r_0, \theta_0) = (e^{\beta \tau_0}, 0)$ and construct the system recursively by drawing the corresponding random variables. This way, we obtain a population of organ primordia positions identified by their rank $p$ (**Appendix 2—figure 1A**).

In this random population, we are interested in which extent the generated phyllotactic patterns overlap. To do so, we estimate whether the points corresponding to primordia of the same rank can be grouped into separable clusters. Therefore, we consider the obtained primordia as a point cloud of 2D cylindrical coordinates labelled by a primordium rank:

$$\mathcal{P} = \left\{ ((r_i, \theta_i), p_i) \mid i \in [\![0, (p_{\max} + 1)K_S[\![ \right\} \tag{31}$$

To answer the separability question, we measure to which extent the identically labeled points $\mathcal{P}_p = \{(r_i, \theta_i), \mid p_i = p\}$ are separable by applying an unsupervised clustering algorithm. For this, we clustered the points using a k-means algorithm. The resulting clusters can be separated by linear boundaries in the Voronoi diagram associated with their centroids $\left\{ (\widehat{\theta_p}, \widehat{r_p}) \mid p \in [\![0, p_{\max}]\!] \right\}$. We measure the linear separability of primordia by looking if points with the same label gather inte the same Voronoi cell.

We use prior knowledge by setting the number of components of the k-means algorithm to $p_{\max} + 1$ and by setting the initial centroids $\left\{ (\widehat{\theta_p^0}, \widehat{r_p^0}) \mid p \in [\![0, p_{\max}]\!] \right\}$ on the positions of the primordia in a model without any noise:

$$\forall p \in [\![0, p_{\max}]\!], \begin{cases} \widehat{\theta_p^0} & = p \cdot \alpha \\ \widehat{r_p^0} & = R \cdot e^{\beta p} \end{cases} \tag{32}$$

After convergence, the algorithm returns $p_{\max} + 1$ centroid points that we use to construct the Voronoi diagram. This actually defines a predictor for the estimated primordium rank $\widehat{p_i}$ by looking inside which cell of the diagram lies a given point (**Appendix 2—figure 1B**).

$$\forall i \in [\![0, (p_{\max} + 1)K_S[\![, \widehat{p_i} = \operatorname*{argmin}_{p \in [\![0, p_{\max}]\!]} \left( \left\| \begin{pmatrix} r_i \\ \theta_i \end{pmatrix} - \begin{pmatrix} \widehat{r_i} \\ \widehat{\theta_i} \end{pmatrix} \right\| \right) \tag{33}$$

Finally, we estimate the Voronoi separability $v$ of our cloud of primordia points by computing the accuracy of the primordium rank prediction (noting $\delta_{ij}$ the Kronecker delta on integers):

$$v = \frac{1}{(p_{\max} + 1)K_S} \sum_{i=0}^{(p_{\max}+1)K_S - 1} \delta_{\widehat{p_i} p_i} \tag{34}$$

If $v$ equals to 1, it means that the primordium points group into perfectly identifiable clusters. This can be interpreted as the fact that $K_S$ randomly sampled individuals can be superimposed perfectly, once they have been centered and aligned on their primordium which is the closest to the $\mathbb{P}_0$ stage.

This will obviously be the case (as long as the motion coefficient $\beta$ remains realistic, typically <1) if no noise is introduced into the system. If $\sigma_\alpha = \sigma_\tau = 0$, then all individuals proceed from the same regular exact pattern, and the only variability will be the one of the instant of sampling $\tau_0$. We wondered up to which level of noise this separability property could be maintained, in order to understand what a high observed separability could tell us on the intrinsic regularity of a phyllotactic system.

To do so, we scanned the parameter space by varying $\sigma_\alpha$ between 0° and 20° and $\sigma_\tau$ between and 2 plastochrons, first with the values $R = 30\mu m$, $\alpha = 137.51°$ and $\beta = 0.23$ corresponding to actual measured data (**Figure 1—figure supplement 2D–E**). As expected,

increasing the angular variability creates more elongated clusters (*Appendix 2—figure 1C*) that still appear separated. Yet, the Voronoi separation introduces confusion between neighboring primordia, more specifically making $\mathbb{P}_p$ and $\mathbb{P}_{p+3}$ overlap (*Appendix 2—figure 1F*). Interestingly, when we increase the plastochron variability (*Appendix 2—figure 1D*), the confusion concerns rather $\mathbb{P}_p$ and $\mathbb{P}_{p+5}$ (*Appendix 2—figure 1G*). In both illustrated cases, the separability score drops markedly below 95%, while the separability of the actual observed data has been evaluated in the same way at 100% (*Appendix 2—figure 1E*).

The landscape of separability in the $\sigma_\alpha \times \sigma_\tau$ parameter space gives an insight on the effects of variability on a population of individuals (*Figure 1—figure supplement 2F–G*). With no surprise, primordia points appear to be less and less identifiable as azimuthal or plastochron variability increase, and even worse when both do. But it shows that there exists a maximal level in variability up to which the clusters are still perfectly separable (*Figure 1—figure supplement 2E*, red contour).

The standard deviation in primordia angles measured on our observed SAMs is equal to 6.7˚. We measured the resulting angular deviation in the simulated primordia points, and we could show that it depends only on the divergence angle variability $\sigma_\alpha$. Moreover, we determined the value of $\sigma_\alpha$ that matches best angular deviation of observed data for primordia ranks ranging from 1 to 5 (where the model tends to show more angular variability than the observations). This value is equal to 3.6˚ (*Appendix 2—figure 1I*).

If we fix the angular variability $\sigma_\alpha$ to this value (*Figure 1—figure supplement 2E*, white vertical line), then the maximal plastochron variability that is allowed for the separability to remain at 100% is close to 0.4. This means that it would be impossible to see the near-perfect superposition observed in our data if the phyllotactic system that produced it had a plastochron variability greater than 0.4, which would translate into an uncertainty on organ initiation times of nearly 5 hr. From this, we conclude that a plastochron variability of 5 hr is an upper bound of the rhythmic variability achieved by real SAMs.

Yet such a value of $\sigma_T$ produces primordia distributions that show much more radial variability than the observed one (*Appendix 2—figure 1A* vs. *Appendix 2—figure 1E*). To get a more precise approximation of this parameter value, we measured the resulting radial deviation in the simulated primordia points. This deviation only depends on the plastochron variability $\sigma_T$, and determined the value that matches best the observed radial deviations for primordia ranks ranging from 0 to 3 (where the model tends to show more radal variability than the observations). This value equals 0.22 (*Appendix 2—figure 1J*), which corresponds to a plausible rhythmic variability of nearly 2 hr for real SAMs. The obtained pattern is more representative of the observed deviations (*Appendix 2—figure 1H*) even if a more accurate model of 3D primordia distribution would be required to estimate exactly the variability parameters of the system. The approximation of 2 hr for the plastochron variability consequently validates our first order assumption that all considered SAMs are in a steady regime of development with the same plastochron duration, and that the whole set of individual primordia of a given rank forms a homogeneous population in terms of developmental state.

Another interesting feature evidenced by this analysis is the influence of the motion speed of primordia separability. We explored the $\beta \times \sigma_\tau$ parameter space by fixing the value of $\sigma_\alpha$ to the observed one, and varying the motion coefficient $\beta$ between 0 and 0.4 (*Appendix 2—figure 1K*). It appears that lowering the speed reduces the maximal possible value of $\sigma_\alpha$ to achieve 100% separability, as the points tend to overlap more in the radial dimension, leading to a decreasing separability at fixed angular variability.

On the other hand, increasing the speed seems to greatly affect the tolerance to plastochron variability. For instance a value of $\sigma_\tau = 1$ that translates into a separability of 95% when $\beta = 0.1$ suddenly drops to a separability of only 50% if $\beta$ is increased up to 0.4. However increasing motion speed does not affect this much the maximal value of $\sigma_\tau$ required to achieve 100% separability, which always remains close to 0.3. This consolidates our previous conclusions, even in the case of an underestimation of the radial speed.

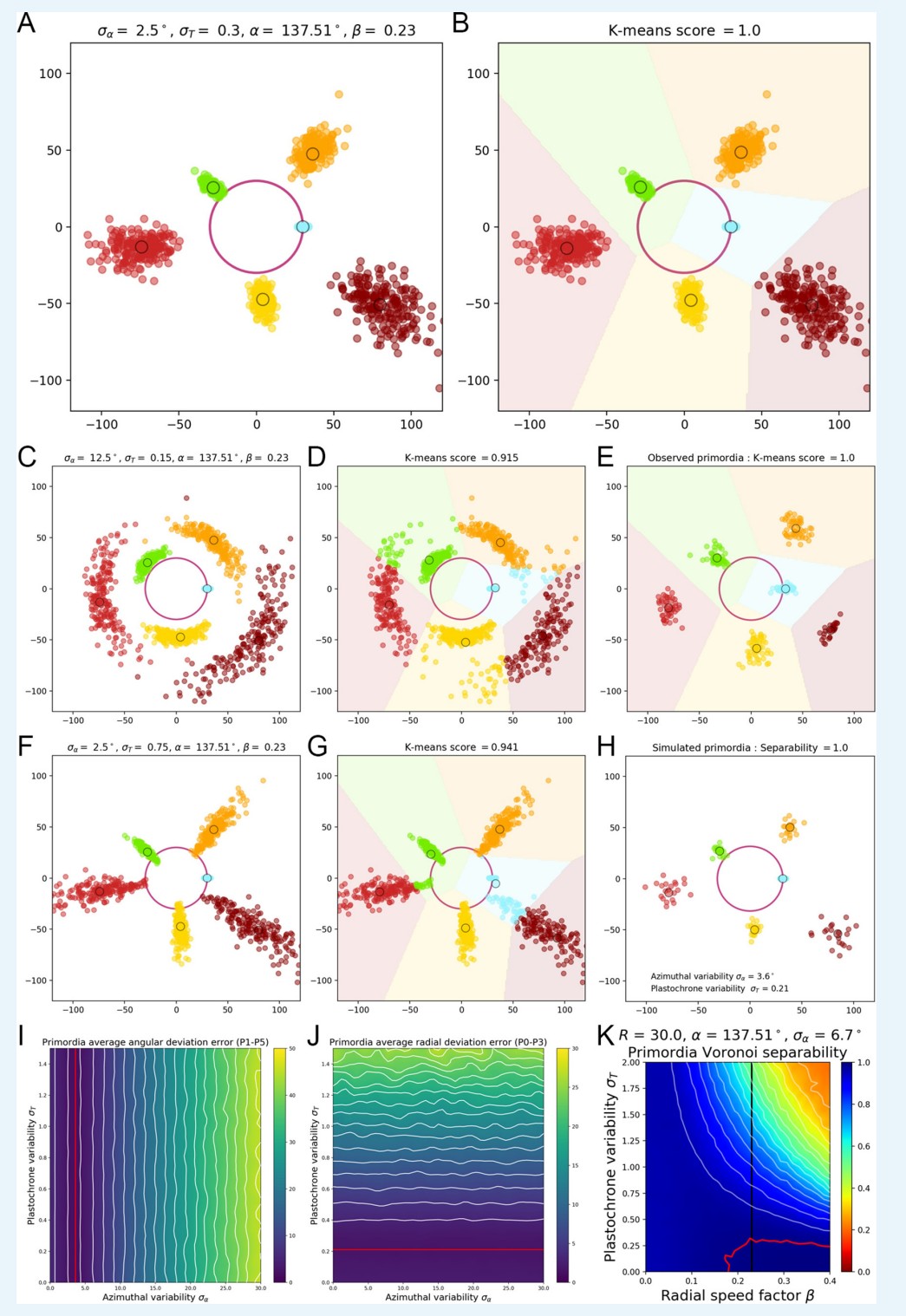

**Appendix 2—figure 1.** A geometrical model of primordia distribution enables estimating the plastochron variability of the SAM. (**A**) Primordium points are generated from a computational phyllotactic model with a control over the variability in angular positioning and plastochron duration. (**B**) Linearly separable clusters in the resulting point cloud are identified using an unsupervised algorithm with prior information. The obtained labeling in primordia ranks is compared with the theoretical one to compute a separability measure. (**C–D**) Increasing the azimuthal variability creates clusters that are more difficult to separate and generates

confusion between $\mathbb{P}_p$ and $\mathbb{P}_{p+3}$. (**E**) The primordia points from the observed experimental data form perfectly separable clusters (**F–G**) Increasing the plastochron variability creates clusters that are more difficult to separate and generates confusion between $\mathbb{P}_p$ and $\mathbb{P}_{p+5}$. (**H–J**) Measured anglular and radial deviations of both observed and simulated primordia allow to determine plausible values for the angular varibility $\sigma_\alpha = 3.6°$ (**I**) and plastochron variability $\sigma_T = 0.22$ (**J**). The prmordia sdistriibution generated with this parameters (**H**) is the one that metches best the observed deviation of primordia clusters. (**K**) Separability evaluated by varying speed coefficient and plastochron variability. Modifying the radial speed of primordia changes the tolerance of the system to azimuthal and plastochron variability. High rhythmic precision is always required to achieve seamless superposition. Red contour indicates 100% separability, white contours every lower 5%. Black line indicates experimental value of speed coefficient.

**Appendix 3**

## Quantitative analysis of nuclei image signals

Going from microscopy images to aligned quantitative data requires a complex computational pipeline that involves several steps of image analysis, computational geometry and data manipulation. The main goal of this pipeline is to provide a representation of the signals contained in the images that allows for quantitative individual comparison and identification of invariant trends in the spatial patterns of the signals across a population. To achieve this, we need to perform three basic tasks:

- Extraction of cellular objects and signal quantification from the raw voxel intensities of the images:

Images are essentially structured grids with an information of signal covering a discretized space, without any explicit notion of what is an object of interest or not. It is then necessary to identify those objects within the image grid, to associate them with a spatial position and extent and to use the signal intensity information to assign quantitative values to each extracted cellular object.

- Geometrical transformation of all individual data into a common spatial reference frame:

In order to be able to compare individual meristems and compute statistics at the scale of a population, we need to align the spatialized data extracted from the images so that it becomes possible for instance to match organs in a comparable developmental state. To do so, we chose to use a common coordinate system into which we transform all the meristem geometries.

- Computation of a continuous representation of the signal that allows point-wise comparison:

The information we extract from the images is defined at the scale of cells, which provides a discrete representation of signals. If we want to compute statistics, we would need to estimate a one-to-one pairing between cells of different individuals, without being sure it exists. Instead, we decided to use a continuous representation of the signals that allows the aggregation of spatialized data. At any location in space, it would be possible to obtain a value of signal for each individual, and to compute statistics without the need for cell matching.

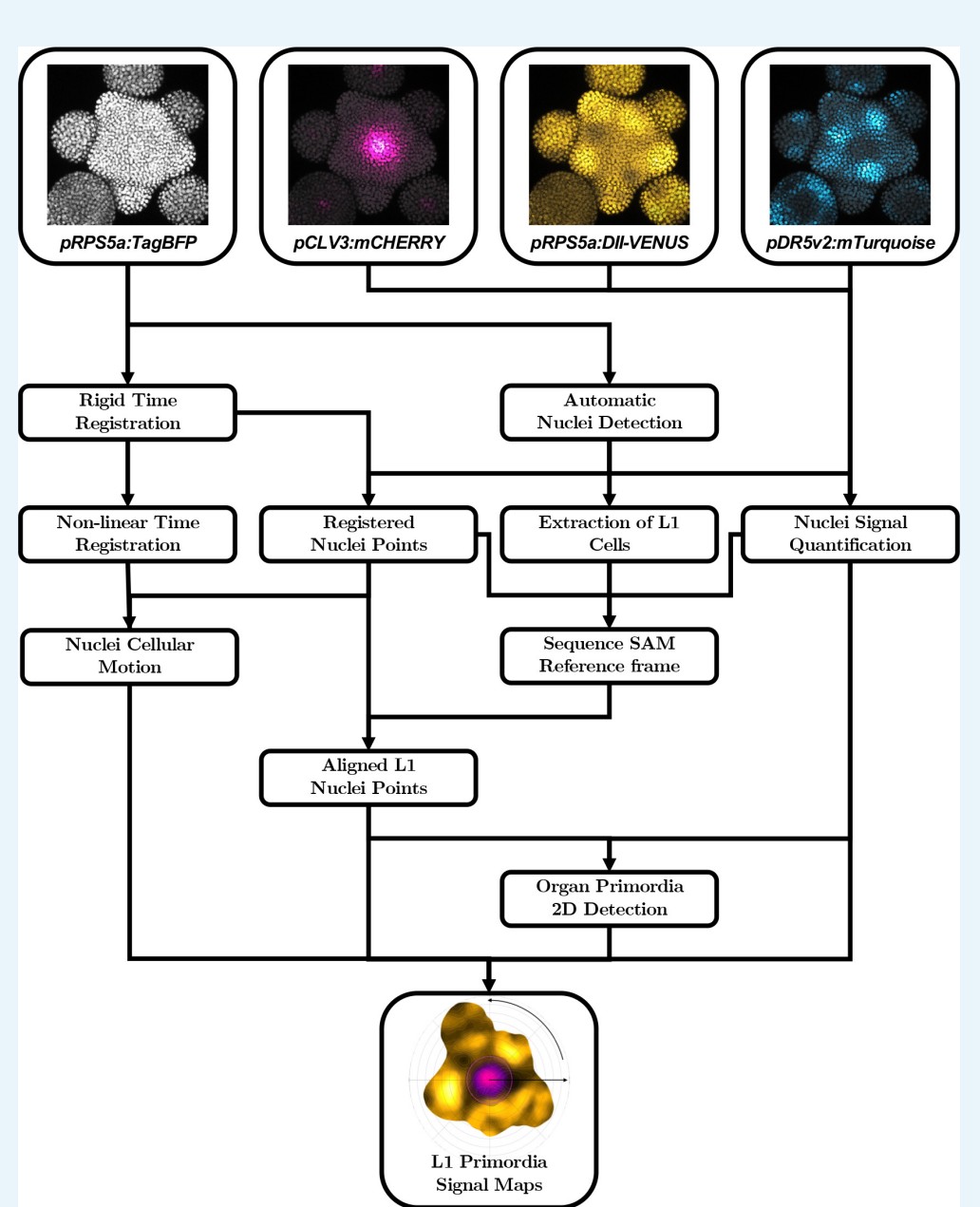

**Appendix 3—figure 1.** Automatic quantification pipeline for the time-lapse microscopy images of nuclei-targeted fluorescence signals. To obtain quantitative data from the images produced under the microscope, various sequential processing steps need to be performed, from the extraction of the relevant objects (nuclei positions with their different channel intensity values) to the geometrical characterization and the spatio-temporal registration of the tissues, to finally get a complete, aligned and consistent dataset gathering all the imaged meristems.

The succession of computational steps that were necessary to perform these tasks is depicted in *Appendix 3—figure 2*. Together, they allow to reconstruct dynamic continuous representations of biological signals averaged over a population of meristems, as the maps displayed in *Figure 1B*, *Figure 3G and I*, *Figure 4D* and *Video 1*. Some tools, such as the estimation of 2D maps of epidermal signal, were used extensively throughout the analysis, even though they do not figure as boxes in the pipeline. The methods used for each of these steps are addressed in detail in the following of this Appendix.

## Automatic nuclei detection

We consider that a 3D image consists of an array $I$ of size $K_x \times K_y \times K_z$ filled with values taken in an integer intensity interval $\mathcal{I} \subset \mathbb{N}$. The elements of this 3D array are called voxels. In the case of a 16-bit-encoded unsigned integer image, the intensity interval $\mathcal{I} = [\![0, 2^{16} - 1[\![$. We denote:

$$I = \{I_{ijk} \in \mathcal{I} \ | \ (i,j,k) \in [\![0, K_x[\![ \times [\![0, K_y[\![ \times [\![0, K_z[\![\}. \tag{35}$$

The voxels in the array grid can be projected into a physical space $\Omega \subset \mathbb{R}^3$ through a mapping function $\mathbf{x} : [\![0, K_x[\![ \times [\![0, K_y[\![ \times [\![0, K_z[\![ \to \Omega$ that associates the array indices with a discrete, evenly spaced, 3D lattice $\Omega_I \subset \Omega$. The voxel size $v = (v_x, v_y, v_z) \in \mathbb{R}^3$ defines a spacing of the lattice that is potentially different on each dimension:

$$\Omega_I = \{\mathbf{x}(i,j,k) = (i \cdot v_x, j \cdot v_y, k \cdot v_z) \ | \ (i,j,k) \in [\![0, K_x[\![ \times [\![0, K_y[\![ \times [\![0, K_z[\![\}. \tag{36}$$

This mapping allow to define the image as a function $I : \Omega_I \to \mathcal{I}$ that associates a 3D point $\mathbf{x} = (x, y, z)$ in the image definition space $\Omega_I$ with a fluorescence intensity value $I(\mathbf{x}) \in \mathcal{I}$, such that $\forall (i,j,k) \in [\![0, K_x[\![ \times [\![0, K_y[\![ \times [\![0, K_z[\![,$

$$I(\mathbf{x}(i,j,k)) = I_{ijk}. \tag{37}$$

In the case of a multichannel image, we denote $I_S$ the image channel corresponding to the signal $S$. Nuclei are detected for each meristem acquisition independently using the *pRPS5a: TagBFP* (Tag) channel, which we denote $I_{\mathrm{Tag}}$.

As a first approximation, cell nuclei can be considered to appear as a roughly spherical blob of fluorescence intensity in the image $I_{\mathrm{Tag}}$, with a limitedly variable radius. To detect them we convolve the image with a sequence of $K_\sigma$ 3D isotropic Gaussian kernels of increasing standard deviations $\Omega_\sigma = \{\sigma_l \ | \ l \in [\![0, K_\sigma[\![\}$, with $\sigma_0 < \ldots < \sigma_{K_\sigma - 1}$. The response to this filtering is expected to be maximal for homogeneous spheres of intensity with a radius close to the standard deviation of the Gaussian kernel.

The sequence of filtered images can be seen as a 4D Gaussian scale-space (**Lindeberg, 1994**) resulting in a 4D image, which by extension we denote $I_{\mathrm{Tag}} : \Omega_I \times \Omega_\sigma \subset \mathbb{R}^4 \to \mathcal{I}$. In this image, the fourth dimension $\sigma$ corresponds to scale. If we note $G(\sigma)$ the discrete Gaussian kernel of standard deviation $\sigma$, the scale-space transform we use is defined by, $\forall (\mathbf{x}, \sigma) \in \Omega_I \times \Omega_\sigma$,

$$I_{\mathrm{Tag}}(\mathbf{x}, \sigma) = \sqrt{\sigma}\big(I_{\mathrm{Tag}} * G(\sigma)\big)(\mathbf{x}). \tag{38}$$

We detect nuclei as a set of local 4D response maxima in this scale-space representation. To conform with the scale-space theory, we define the scale interval $\Omega_\sigma$ as a geometric sequence varying from $\sigma_{\min}$ to $\sigma_{\max}$. These two bounds are to be chosen in the typical range of variation of nuclei radius.

A 4D point $(\mathbf{x}, \sigma)$ is then considered a local maximum if its response $I_{\mathrm{Tag}}(\mathbf{x}, \sigma)$ is higher than a threshold $I_{\min}$, and higher that all the neighbouring responses at all scales. More formally, if we note $\mathcal{B}(\mathbf{x}, \sigma) = \{\mathbf{x}' \in \Omega_I \ | \ \|\mathbf{x}' - \mathbf{x}\| < \sigma\}$ the discrete ball of radius $\sigma$ in the image lattice $\Omega_I$, the 4D point $(\mathbf{x}, \sigma)$ is a *local maximum* if it realizes the maximal value of response over $\mathcal{B}(\mathbf{x}, \sigma) \times \Omega_\sigma$, in other terms:

$$(\mathbf{x}, \sigma) \text{ is a } local \ maximum \text{ of } I_{\mathrm{Tag}} \iff (\mathbf{x}, \sigma) \in \underset{(\mathbf{x}', \sigma') \in \mathcal{B}(\mathbf{x}, \sigma) \times \Omega_\sigma}{\mathrm{argmax}} (I_{\mathrm{Tag}}(\mathbf{x}', \sigma')). \tag{39}$$

Let us denote $\mathcal{P}$ the set of points $\mathbf{x} \in \Omega_I$ corresponding to a *local maximum* of $I_{\mathrm{Tag}}$ whose response is strictly greater than $I_{\min}$. Each point in $\mathcal{P}$ corresponds to a detected nucleus, identified by an integer index $n$ and associated with a single spatial position $P_n$ in physical coordinates, so that we can write

$$\mathcal{P} = \{P_n \mid n \in \mathcal{N} = [\![1, |\mathcal{P}|]\!]\}. \tag{40}$$

This detection method has been evaluated on a set of 4 manually expertized SAM images acquired at different voxel sizes with a 16 bit encoding, representing more than 5000 cells. Since the acquisition parameters for those images matched the one use in the rest of our analysis, we adjusted the method parameters to perform best on this expertized dataset. The parameter testing led to the determination of the optimal values $K_\sigma = 3$, $\sigma_{\min} = 0.8\mu m$, $\sigma_{\max} = 1.4\mu m$, $I_{\min} = 3000$, corresponding to an evaluated performance of 95.6% recall (percentage of the manually labelled cells that were indeed detected) and 98.5% precision (percentage of the detected cells that were actually labelled by experts).

## Nuclei signal quantification

Each image channel is quantified at the level of every detected nucleus. The signal intensity value is obtained by computing a weighted average of the channel intensity $I_S$ around the position of the nucleus. The signal images showing some local subcellular noise, the raw voxel value $I_S(P_n)$ might not be fully representative of the whole nucleus. We chose to use a distance-based Gaussian weight, of constant radius $\sigma_\mathcal{N}$ for all channels, to account for as much as possible of the signal information inside the nuclei, for which the typical measured diameter is $\sim 5\mu m$. Practically for the signal $S$, we first filter the image channel $I_S$ by a Gaussian kernel of radius $\sigma_\mathcal{N}$ and retrieve the values at the voxel positions of all detected nuclei, so that $\forall n \in \mathcal{N}$,

$$S_n = (I_S * G(\sigma_\mathcal{N}))(P_n). \tag{41}$$

For example, the local level of expression of the *CLV3* gene, imaged using *pCLV3: mCHERRY* in the channel $I_{\mathrm{CLV3}}$ would we quantified as $\mathrm{CLV3}_n = (I_{\mathrm{CLV3}} * G(\sigma_\mathcal{N}))(P_n)$. In the case of the ratiometric auxin sensor qDII, we combine the information of two fluorescence channels $I_{\mathrm{Tag}}$ and $I_{\mathrm{DII}}$ to compute the ratio of estimated signals for each nuclei point, so that $\forall n \in \mathcal{N}$,

$$\mathrm{qDII}_n = \frac{\mathrm{DII}_n}{\mathrm{Tag}_n}. \tag{42}$$

## Extraction of L1 cells

For the purpose of the analysis, we want to discriminate between the first layer of cells ($L_1$) and the rest of the tissue. We use an automatic method to do so, which in our case cannot rely on adjacency to the background as one would do on a segmented membrane-marker image. Instead we will use the distance of the nuclei to the estimated surface of the tissue.

This surface is computed based on the *pRPSa:TagBFP* channel as a 3D triangle mesh $\mathcal{M} = \{\mathcal{V}, \mathcal{T}\}$, where:

- $\mathcal{V}$ is a set of vertices
- Each vertex $v \in \mathcal{V}$ is associated with a 3D position $M_v \in \mathbb{R}^3$
- $\mathcal{T}$ a set of triangles defined by triplets of indices $(v_1, v_2, v_3) \in \mathcal{V}^3$
- $\mathcal{T}$ is such that the resulting simplicial complex forms a 2-manifold (**Agoston, 2005**) (Chapter 5.3: Topological Spaces, Chapter 6.3: Simplicial Complexes).

## Computation of the surface mesh

To obtain this triangle mesh, the image $I_{\mathrm{Tag}}$ is filtered by a large Gaussian kernel to diffuse nuclei intensity between cells, and is thresholded to obtain a binary region, which is meshed by applying a Marching Cubes algorithm (**Lorensen and Cline, 1987**) on a resampled version of the image. This mesh undergoes a phase of triangle decimation (**Garland and Heckbert, 1997a**) and isotropic remeshing (**Botsch and Kobbelt, 2004a**) to obtain a surface composed of roughly 50000 regular faces.

At this stage, we generally obtain a surface that goes both above and below the tissue (notably due to weaker intensity in the inner tissue). The remove the lower part that does not correspond to the epidermal surface, we estimate the face normal vectors to keep only the triangles for which the normal points towards the upper side of the meristem. The largest edge-connected component of this set of triangles is kept and used as the estimated meristem surface mesh $\mathcal{M}$.

## Detection of $L_1$ nuclei

With each vertex $v$ of the surface mesh $\mathcal{M}$, we associate the index $n(v)$ of the *closest point* in the set $\mathcal{P}$ of nuclei points: $\forall v \in \mathcal{V}$,

$$n(v) = \underset{n \in \mathcal{N}}{\arg\min}(\|P_n - M_v\|). \tag{43}$$

We define $\mathcal{L}_1$ as the subset of $\mathcal{N}$ formed by indices $n$ of nuclei points $P_n$ that are the *closest point* to at least one vertex of $\mathcal{M}$.

## Rigid time registration

The previous steps were performed individually on each frame of the time-lapse acquisitions. In our study, we focused on sequences of observation of the same individual over its development, consisting of $K_t$ multichannel images $\{I(t_i) \mid i \in [\![0, K_t]\!]\}$, indexed by their temporal position $t_i \in \mathbb{N}$ in hours relatively to the first time of acquisition $t_0 = 0h$. In the remaining, we will consistently index data computed from the $i$-th acquisition $I(t_i)$ by the temporal index. For instance $\mathcal{P}(t_i) = \{P_n \mid n \in \mathcal{N}(t_i)\}$ denotes the set of nuclei points detected in $I(t_i)_{\mathrm{Tag}}$.

To study consistently the dynamics at the scale of the sequence of images, we need to place the quantitative nuclei information in the same spatial reference frame. To do this, we estimate 3D rigid transformations between consecutive time frames of the sequence. This estimation is performed using a block matching algorithm (*Ourselin et al., 2000*) applied on the *pRPS5a:TagBFP* channel $\{I(t_i)_{\mathrm{Tag}} \mid i \in [\![0, K_t]\!]\}$ of the consecutive images. This produces $K_t - 1$ isometry matrices in homogeneous coordinates $R_{t_i \leftarrow t_{i+1}}$ that can be inverted and/or multiplied to transform any frame of the sequence into the spatial reference frame of any other.

## Registered nuclei points

We use the registration output to transform all the detected nuclei points into the coordinate system of the first frame of the sequence. By applying the resulting rigid transforms to the the nuclei points detected at time $t_i$, we obtain a new point cloud $\mathcal{P}(t_i)^0$, indexed by the same set of integers $\mathcal{N}(t_i)$, such that $\forall n \in \mathcal{N}(t_i)$,

$$P_n^0 = \left( \prod_{j=i-1}^{0} R_{t_j \leftarrow t_{j+1}} \right) P_n. \tag{44}$$

## Non-linear time registration

In a second time, we want to estimate the local deformation of the tissue, notably to get a quantitative measure of individual cell motion and to approximate local cellular growth.

We compute this new transformation as a dense vector field that maps two consecutive *pRPS5a:TagBFP* images, that have previously been rigidly registered into the coordinate system of $I(t_0)$. We denote $\{I(t_i)_{\mathrm{Tag}}^0 \mid i \in [\![0, K_t]\!]\}$ the sequence of such registered images.

The vector field transforming $I(t_i)_{\mathrm{Tag}}^0$ into $I(t_{i+1})_{\mathrm{Tag}}^0$ is estimated using the block matching framework for non-linear registration (*Ourselin et al., 2000*). This approach has proven to be

efficient on plant tissues in the case of deformations of moderate amplitude (*Fernandez et al.,* *2010*; *Michelin et al., 2016*) which is the case for the meristematic tissue we are considering with a $4h$ to $5h$ time interval between two frames.

The resulting vector field is actually a vectorial image $\mathbf{U}^0_{t_i \to t_{i+1}} : \Omega_{I(t_0)} \to \mathbb{R}^3$ defined over the same grid as $I(t_0)$. It contains at each voxel position a 3D vector measuring the local total deformation of the tissue to go from the current frame to the next one.

We also perform the backwards non-linear registration by computing another set of non-linear transformations between $I(t_{i+1})^0_{\mathrm{Tag}}$ and $I(t_i)^0_{\mathrm{Tag}}$ in the exact same manner and obtain the vector fields $\mathbf{U}^0_{t_i \leftarrow t_{i+1}}$ mapping the next frame to the current one.

## Nuclei cellular motion

The cellular motion between two consecutive sequence frames taken at times $t_i$ and $t_{i+1}$ in the forward direction can be estimated using the registered nuclei points $P(t_i)^0 = \{P^0_n \mid n \in \mathcal{N}(t_i)\}$ and the transformation $\mathbf{U}^0_{t_i \to t_{i+1}}$ that maps the current frame into the next one by a vector field. Each nucleus $n$ can then be assigned a local displacement vector by looking at $\mathbf{U}^0_{t_i \to t_{i+1}}(P^0_n)$.

We deduce speed vectors $\mathbf{v}^0_n$ measuring the local speed of cellular motion by dividing the displacement vector by the time interval: $\forall n \in \mathcal{L}_1(t_i)$,

$$\mathbf{v}^0_n = \frac{1}{t_{i+1} - t_i} \mathbf{U}^0_{t_i \to t_{i+1}}(P^0_n) \tag{45}$$

## 2D maps of epidermal signal

To infer a signal value on any 3D point in space from the values quantified at discrete nuclei points, we used an approximation strategy. Our method makes the signal continuous in space by computing a local weighted average of signal values. The weighting function $\eta$ we use is a parametric sigmoid density function of the distance $r$ to a given point, $\eta : \mathbb{R}^+ \to [0, 1]$, which relies on two parameters: an extent parameter $R$, and a sharpness parameter $k$ such that $\eta(0) \sim 1$, $\eta(R) = \frac{1}{2}$, and $\dot{\eta}(R) = -\frac{k}{2}$ (*Appendix 3—figure 3A*). This sigmoid function takes the form:

$$\eta(r) = \frac{1}{2} - \frac{1}{2} tanh(k \cdot (r - R)) \tag{46}$$

The continuous signal map $\widehat{S}$ is then defined based on a point cloud $\mathcal{P} = \{P_n \in \mathbb{R}^3 \mid n \in \mathcal{N}\}$ and the associated signal values $\{S_n \mid n \in \mathcal{N}\}$ as : $\forall \mathbf{x} \in \mathbb{R}^3$,

$$\widehat{S}(\mathbf{x}) = 1 \sum_{n \in \mathcal{N}} \eta(\|\mathbf{x} - P_n\|) \sum_{n \in \mathcal{N}} \eta(\|\mathbf{x} - P_n\|) S_n \tag{47}$$

Note that, for any point $\mathbf{x}$ where the total density $H(\mathbf{x}) = \sum_{n \in \mathcal{N}} \eta(\|\mathbf{x} - P_n\|)$ equals , the signal map is not defined. To make the map outlines closer to their actual support, we consider that $\widehat{S}(\mathbf{x})$ is defined only if $H(\mathbf{x}) \geq \frac{1}{2}$. This constraint is equivalent to consider that the implicit surface obtained with the point cloud $\mathcal{P}$ as generator and $\eta$ as potential forms the boundary of the definition domain of the function $\widehat{S}$.

The first layer of cells forms a continuous surface and in the case of the shoot apical meristem, the curvature of this surface is such that it generally does not create overlaps in a 2D projection made along the meristem axis. Therefore, to study processes taking place at the $L_1$, we compute maps using only the subset $\mathcal{L}_1$ of first-layer nuclei and their 3D point cloud projected on the plane $z = 0$ (*Appendix 3—figure 2C-E*), which we call *2D projected maps of epidermal signal*. In this case, by extension we denote $\widehat{S}(x, y) = \widehat{S}(x, y, 0)$.

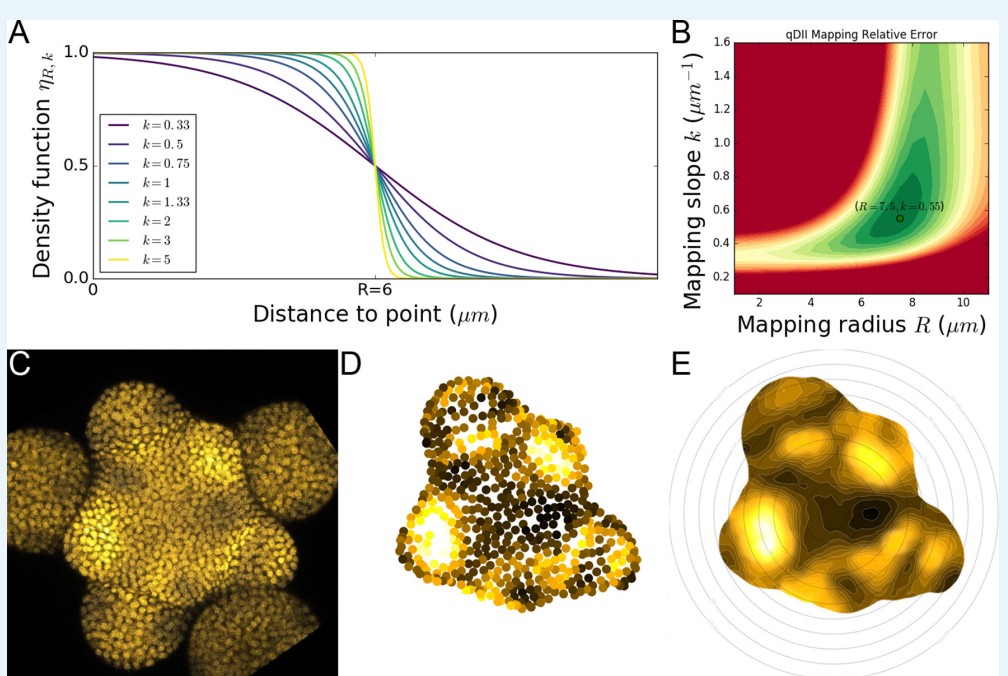

**Appendix 3—figure 2.** 2D continuous maps of epidermal signals. (**A**) To build a continuous 2D map, we diffuse the signal in space by computing a local average of discrete signal values using a kernel function whose extent and sharpness are set by two parameters $R$ and $k$. (**B**). Optimal values of these parameters were determined on the signal of main interest qDII and are the ones used throughout the analyses. (**C–E**). Using the L1 nuclei detected in the confocal image (**C**) and their quantified signal values projected in 2D (**D**) we compute a 2D map of epidermal signal (**E**) in this case qDII.

To determine the best parameter values $R$ and $k$ for the function $\eta$ we ran an extensive parameter exploration and measured the error made by mapping the epidermal signal to retain the values that yield the minimal error. This is measured by the average relative error between the actual signal value $S_n$ of a nucleus and the value of the 2D map $\widehat{S}$ computed with all nuclei *but the considered one* at the projected position of the nucleus.

We searched for optimal values on the qDII signal, as it is our main focus in this work, using the whole set of 21 available SAM image sequences. The values we obtain, shown in *Appendix 3—figure 2B*, are $R^* = 7.5\mu m$ and $k^* = 0.55\mu m^{-1}$. From now on, we will consider that, if not explicitly mentioned otherwise, the maps we use are *2D projected maps of epidermal signal* computed with the optimal parameters $R = R^*$ and $k = k^*$ for the density function $\eta$.

## Sequence SAM reference frame determination

In order to aggregate the data quantified on several individuals and expressed in their own image reference frame, we need to perform a geometrical alignment that superimposes organs with a similar developmental state.

To do so, we chose to map a common coordinate system onto the point clouds extracted from the images by landmarking a set of key geometrical features for each meristem (*Appendix 3—figure 3A-B*):

- the position $\mathbf{c} = (x_{\mathbf{c}}, y_{\mathbf{c}}, z_{\mathbf{c}}) \in \mathbb{R}^3$ of the apex center in the central zone (CZ) of the meristematic dome
- the unitary vector $\mathbf{a} \in \mathbb{R}^3$ of the main radial symmetry axis of the meristematic dome
- the unitary radial vector $\mathbf{r} \in \mathbb{R}^3$ ($\mathbf{a} \perp \mathbf{r}$) of the direction of the last initiated organ primordium (labelled $\mathbb{P}_0$) relatively to $\mathbf{c}$

- the binary orientation $o \in \{-1, 1\}$ of the phyllotactic spiral (clockwise or counter-clockwise)

Together, these landmarks define a new reference frame $\mathcal{R}^*$ (**Appendix 3—figure 3C**) into which we will rigidly transform all the nuclei points and images.

## Detection of the CZ center $\mathbf{c}$

The *CLV*3 peptide, imaged in the *pCLV3:mCHERRY* channel, is a marker of the central zone (CZ) of the meristem that is expressed notably on the first layer of cells. We use the quantified signal values $\{\mathrm{CLV3}_n \mid n \in \mathcal{L}_1\}$ considering nuclei points from the whole sequence (registered on the reference frame $\mathcal{R}^0$ of the first image of the sequence) to estimate the center position.

In the resulting 2D map $\widehat{\mathrm{CLV3}}$, the CZ appears as a wide isotropic peak of signal intensity (**Appendix 3—figure 3D**). We assume that the CZ is the largest area of high *CLV*3 intensity in the tissue we consider. We extract a central zone domain by thresholding the 2D map $\widehat{\mathrm{CLV3}}$ and keeping the largest connected component, from which we compute the 2D center $(x_{\mathbf{c}}, y_{\mathbf{c}})$ and the area $A_{\mathbf{c}}$.

To make the estimation more robust, we perform this CZ extraction using a range of $K_{\mathbf{c}}$ threshold values that depend on the average signal intensity of the sequence $\{c_k \cdot \overline{\mathrm{CLV3}} \mid k \in [\![0, K_{\mathbf{c}}[\![\}$. In the end, we estimate $(x_{\mathbf{c}}, y_{\mathbf{c}})$ as the average 2D center and $A_{\mathbf{c}}$ and the average area of all these central zone domains obtained with $K_{\mathbf{c}} = 7$ and $c_k = 1.2 + 0.1k$.

The resulting distribution of estimated CZ radii $r_{\mathbf{c}} = \sqrt{\frac{A_{\mathbf{c}}}{\pi}}$ among the considered individuals showed a very peaked distribution around an average value of 28 µm (**Appendix 3—figure 3E**).

Finally, the third coordinate $z_{\mathbf{c}}$ of the 3D center of the CZ is computed as the estimated value $\widehat{z^0}$ of the $z$ coordinate of $L_1$ nuclei points in $\mathcal{R}^0$, taken at the center point $(x_{\mathbf{c}}, y_{\mathbf{c}})$ detected above.

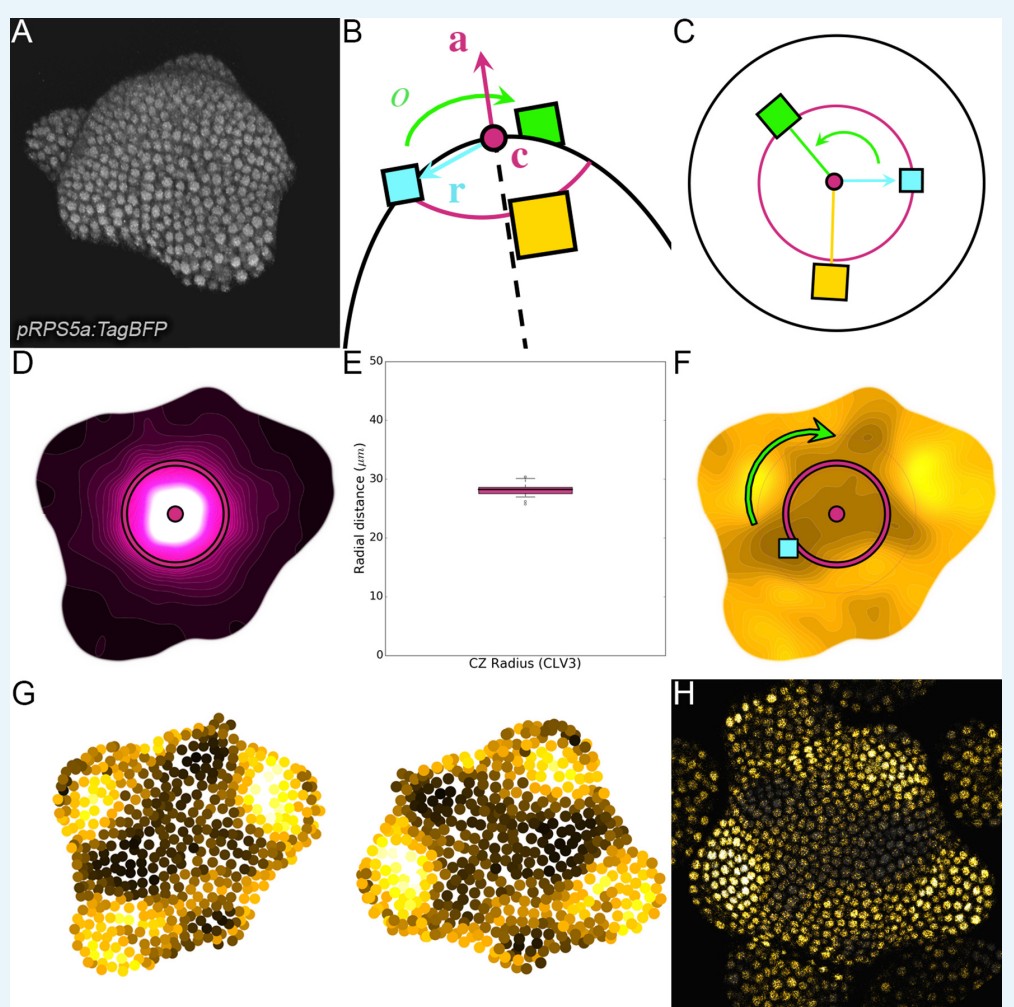

**Appendix 3—figure 3.** Landmark-based alignment of SAMs using 2D continuous maps. (**A**) Seen in 3D, the SAM shows a dome-like structure surrounded by a spiral of primordia. (**B**). The center and the main axis of the dome, the direction of the last intiated primordium and the orientation of the phyllotactic spiral are key landmarks of the SAM geometry. (**C**). Knowing their position allows to define a reference frame $\mathcal{R}^*$ in which all individuals can be superimposed. (**D**). The 2D projected map of epidermal CLV3 signal allows locating precisely the center **c** and main axis **a** of the meristem and estimating the extent of the CZ. (**E**). The extents of CZ estimated using the CLV3 maps show a very limited variability around the 28μ m value among the observed individuals (N = 21 SAMs). (**F**). The 2D projected map of epidermal qDII signal allows detecting the direction **r** of the $\mathbb{P}_0$ primordium as the maximal concentration of auxin in the PZ. The phyllotactic orientation $o$ is set manually. (**G**) The rigid transformation is applied to the detected nuclei to align the individuals in the reference frame $\mathcal{R}^*$. (**H**) The same is done with images and a curved L1 slice is used to display only epidermal cells.

## Estimation of the vertical axis of the apex a

In the reference frame of the image, the surface of the meristematic dome around the central zone might be slightly tilted. To correct this, we estimate the vector **a** that corresponds to the direction around which the surface of the meristem is radially symmetrical.

This vector is characterized by a rotation matrix $R_{\mathbf{a}}$ to allows to transform the reference frame $\mathcal{R}^0$ of the image to a new reference frame $\mathcal{R}_{\mathbf{a}}$ where the origin is **c** and the $z$ axis corresponds to **a**.

We estimate the rotation matrix by minimizing the tilting of the surface of the CZ in this new reference frame. To do so, we look at the dispersion of $z$ coordinates of the nuclei expressed in this reference frame as a function of their radial distance. We minimize the error between the $z$ coordinate of the $L_1$ nuclei points and their local average $\bar{z}$. To take into account the extent of the meristematic dome, this error is weighted by a Gaussian function of the distance of points to $\mathbf{c}$ of standard deviation $\sigma_r = 20\mu m$.

Note that in principle, at this stage we should re-estimate the position $\mathbf{c}$ of the center using the nuclei points expressed in the reference frame $\mathcal{R}_\mathbf{a}$. In our case, we considered that we could avoid this iterative estimation since the meristems are imaged from above and we can make the approximation that the vector $\mathbf{a}$ forms a small angle with the $z$ axis of the images. Consequently, the initial estimation of $\mathbf{c}$ in the $xy$ plane of the image does not induce a significant error.

## Detection of the $\mathbb{P}_0$ direction r

Within the previous reference frame $\mathcal{R}_\mathbf{a}$ equipped with a cylindrical coordinate system $(r, \theta, z)$, we now look for the azimuthal direction $\theta_0$ of the primordium labelled $\mathbb{P}_0$. For this, we need to locate the local spatio-temporal maximum of average auxin concentration in the peripheral zone (PZ), as explained in Appendix id1. Therefore, we look for the minimal value of the 2D map $\widehat{\text{qDII}}$ (computed using the nuclei points of the whole sequence), which corresponds to a signal projected orthogonally to $\mathbf{a}$.

We limit the search to the PZ by constraining that $r \in [\rho_{\min}r_\mathbf{c}, \rho_{\max}r_\mathbf{c}]$ to avoid artifactual detections inside the CZ or in older organs (we used $\rho_{\min} = 0.9$ and $\rho_{\max} = 1.6$ in the analysis). In the end we get the coordinates of an auxin maximum out of which the azimuthal coordinate $\theta_0$ defines the unitary vector $\mathbf{r} = (\cos(\theta_0), \sin(\theta_0), 0)$ in the reference frame $\mathcal{R}_\mathbf{a}$ (**Appendix 3—figure 3F**). We denote the $R_\mathbf{r}$ the rotation matrix that allows to transform the $x$ axis of $\mathcal{R}_\mathbf{a}$ into $\mathbf{r}$.

## Determination of the phyllotactic orientation

Finally, rather than estimating the orientation of the meristem from the data, we chose to rely on manual expertise to determine visually wheter the arrangement of organs arond the meristem showed a clockwise ($o = -1$) or counterclockwise ($o = 1$) phyllotactic spiral. This produces a simple identity or reflection matrix $R_o$, depending on the case:

$$R_o = \begin{pmatrix} 1 & 0 & 0 \\ 0 & o & 0 \\ 0 & 0 & 1 \end{pmatrix}. \tag{48}$$

## Aligned L1 nuclei points

In the end, the determination of the SAM landmarks $\mathbf{c}$, $\mathbf{a}$, $\mathbf{r}$ and $o$ allows to transform the sequence registered points $\mathcal{P}^0$ into the common 3D reference frame $\mathcal{R}^*$ in which we will be able to compare different individuals locally (**Appendix 3—figure 3G**). We denote $R_{*\leftarrow t_0}$ the corresponding rigid transform, that can be written in homogeneous coordinates as

$$R_{*\leftarrow t_0} = \begin{pmatrix} R_o \cdot R_r \cdot R_a & -c \\ o & 1 \end{pmatrix} \tag{49}$$

We note $\mathcal{P}^* = \{P_n^* \mid n \in \mathcal{L}_1\}$ the positions of first-layer nuclei points in this common reference frame, defined by, $\forall n \in \mathcal{L}_1, P_n^* = R_{*\leftarrow t_0} \cdot P_n^0$. From now on, we will always consider that the nuclei points have been aligned in the common SAM reference frame $\mathcal{R}^*$, and that the signal maps are computed using the aligned positions $\mathcal{P}^*$.

## Aligned L1 image slices

We also use the resulting alignment transformation to register the original images into the common reference frame by applying the same transform $R_{* \leftarrow t_0}$ to the registered image $I(t_i)_S^0$. This creates a registered image $I(t_i)_S^*$ expressed over a new voxel grid $\Omega_I^*$ that is centered on 0 in x and y and slightly shifted in z so that:

$$\Omega_I^* = \Omega_I - \begin{pmatrix} \frac{K_x}{2} \cdot v_x \\ \frac{K_y}{2} \cdot v_y \\ \frac{K_z}{8} \cdot v_z \end{pmatrix}. \tag{50}$$

We used the aligned 2D epidermal maps in order to create the 2D projected views of these registered images displaying only a single layer of cells that can be seen in **Appendix 3— figure 3H**, **Figure 2A–C**, **Figure 3A–B**, **Figure 4A–C** and **Figure 5A–H**. More specifically, we compute the values of the 2D map $\widehat{z^*}$ over the $(x, y)$ coordinates of the centered grid $\Omega_I^*$, and produce a 2D image $I(t_i)_S^*$ defined over the same $(x, y)$ grid as the original one but keeping the voxel value of one $z$ slice per pixel, so that $\forall (x, y, z) \in \Omega_I^*$,

$$I(t_i)_S^*(x, y) = I(t_i)_S^* \left( x, y, \left\lfloor \frac{\widehat{z^*}(x, y)}{v_z} \right\rfloor \cdot v_z \right). \tag{51}$$

The produced 2D image displays the intensity levels of a single curved image slice that goes through the nuclei of first cell layer, making information appear more clearly than a simple maximal intensity projection that would also include intensity from the inner layers.

## Organ primordia 2D detection

In the aligned SAM reference frame $\mathcal{R}^*$, we expect to find organs in comparable developmental stages at very close spatial locations for all individuals, under the global hypothesis of a stationary and regular development. Therefore we use some a priori knowledge on regular phyllotaxis to detect the positions of the ranked organ primordia in the meristem, namely $\mathbb{P}_0$, $\mathbb{P}_1$, $\mathbb{P}_2$ and so on.

Previous works on auxin dynamics in the meristem suggest that primordia correspond to local accumulation of auxin, which would be detectable as local minima in the $\widehat{\mathrm{qDII}}$ map, but also that soon after organ initiation, an auxin depletion area is formed, creating a local maximum in $\widehat{\mathrm{qDII}}$. In that respect, our primordia detection procedure consists first in detecting extremal points (both maxima and minima) in this 2D map and in labeling them in a second time by organ primordium rank.

## Detection of extremal regions in the auxin map

To locate extremal regions in 2D, we are not only interested in absolute local extremality (namely peaks and troughs of the map) but also in points that are extremal in a given direction, and we therefore look for ridges and valleys of the $\widehat{\mathrm{qDII}}$ map (**Appendix 3—figure 4A**). Peaks, which are maximal in any direction, will appear as convergence points of ridges, troughs as convergence points of valleys, and saddle points as convergence points of a ridge and a valley.

If we consider a direction defined by a unitary vector $\mathbf{u} \in \mathbb{R}^2$, the local minimality of the 2D field $\widehat{\mathrm{qDII}}$ along $\mathbf{u}$ can be defined as a scalar field $\Lambda_{\mathbf{u}}^- : \mathbb{R}^2 \to \{0, 1\}$ that takes the value 1 only if the sign of the derivative of $\widehat{\mathrm{qDII}}$ along $\mathbf{u}$ (noted $\vec{\nabla}_{\mathbf{u}}$) changes from negative to positive. The same goes for the local maximality of the signal along $\mathbf{u}$, noted $\Lambda_{\mathbf{u}}^+ : \mathbb{R}^2 \to \{0, 1\}$, except that the sign of the derivative should go from positive to negative. We define the functions $\Lambda_{\mathbf{u}}^-$ and $\Lambda_{\mathbf{u}}^+$ by:

$$\Lambda_{\mathbf{u}}^{-} = \begin{cases} 1 & \text{if} \begin{cases} \vec{\nabla}_{\mathbf{u}}\widehat{\text{qDII}} = 0 \\ \vec{\nabla}_{\mathbf{u}}\vec{\nabla}_{\mathbf{u}}\widehat{\text{qDII}} > 0 \end{cases} \\ 0 & \text{otherwise} \end{cases} \qquad \Lambda_{\mathbf{u}}^{+} = \begin{cases} 1 & \text{if} \begin{cases} \vec{\nabla}_{\mathbf{u}}\widehat{\text{qDII}} = 0 \\ \vec{\nabla}_{\mathbf{u}}\vec{\nabla}_{\mathbf{u}}\widehat{\text{qDII}} < 0 \end{cases} \\ 0 & \text{otherwise} \end{cases} \tag{52}$$

If we consider all possible directions, defined by their angle $\omega \in [-\pi, \pi]$, a valley point (respectively a ridge point) is a point that achieves local minimality (respectively local maximality) along a large proportion of unitary vectors $\mathbf{u}(\omega) = (\cos(\omega), \sin(\omega))$. Consequently, we define the scalar fields $\Lambda^{-} : \mathbb{R}^2 \to [0,1]$ and $\Lambda^{+} : \mathbb{R}^2 \to [0,1]$ measuring respectively the global *valleyness* and *ridgeness* of a point in the field $\widehat{\text{qDII}}$ as:

$$\Lambda^{-} = \frac{1}{2\pi}\int_{-\pi}^{\pi} \Lambda_{\mathbf{u}(\omega)}^{-} d\omega \qquad \Lambda^{+} = \frac{1}{2\pi}\int_{-\pi}^{\pi} \Lambda_{\mathbf{u}(\omega)}^{+} d\omega. \tag{53}$$

We first look for saddle points in order to eliminate them and disconnect the otherwise continuous networks of ridges and valleys. Saddle regions $\mathcal{L}^0$ are formed by points that belong to both a valley and a ridge and are detected using the geometric mean of valleyness and ridgeness fields, then valley regions $\mathcal{L}^-$ and ridge regions $\mathcal{L}^+$ can be found outside the saddle regions, using different thresholds $\lambda^0$, $\lambda^+$ and $\lambda^-$ in each case:

$$\begin{aligned} \mathcal{L}^0 &= \left\{ \mathbf{x} \in \mathbb{R}^2 \mid \sqrt{\Lambda^+(\mathbf{x})\Lambda^-(\mathbf{x})} \geq \lambda^0 \right\} \\ \mathcal{L}^- &= \left\{ \mathbf{x} \in \mathbb{R}^2 \setminus \mathcal{L}^0 \mid \Lambda^-(\mathbf{x}) \geq \lambda^- \right\} \\ \mathcal{L}^+ &= \left\{ \mathbf{x} \in \mathbb{R}^2 \setminus \mathcal{L}^0 \mid \Lambda^+(\mathbf{x}) \geq \lambda^+ \right\} \end{aligned} \tag{54}$$

We perform this region extraction with the values $\lambda^0 = 0.01$ and $\lambda^+ = \lambda^- = 0.03$. Then we consider the union of the connected components of each set $\mathcal{L}^0$, $\mathcal{L}^-$ and $\mathcal{L}^+$, which results in a set of connected regions $\mathcal{C} = \{c_l \mid l \in [\![0, K_{\mathcal{C}}]\!]\}$ as the ones delineated in *Appendix 3—figure 4B*. These regions are guaranteed to have no overlap as long as $\lambda^0 < \lambda^+$ and $\lambda^0 < \lambda^-$.

## Description of extremal regions

Each of these connected extremal regions can potentially match the auxin accumulation or depletion zone characterisitic of a given primordium. In order to determine this matching we characterize the information contained in each of the extremal regions by defining a set of descriptor functions:

- *type* : $\mathcal{C} \to \{-1, 0, 1\}$: whether the region is a connected component of $\mathcal{L}^-$, $\mathcal{L}^0$ or $\mathcal{L}^+$ respectively.
- *pos* : $\mathcal{C} \to \mathbb{R}^2$: a single spatial position corresponding to the point of extremal value of the $\widehat{\text{qDII}}$ field.
- *area* : $\mathcal{C} \to \mathbb{R}$: the area of the connected region.
- *extr* : $\mathcal{C} \to [0, 1]$: the maximal value of extremality (valleyness, geometric mean or ridgeness).
- $\widehat{\text{qDII}}$ : $\mathcal{C} \to \mathbb{R}$: the extremal value of the $\widehat{\text{qDII}}$ field in the connected region.

For each region, we define two significance scores $\gamma^- : \mathcal{C} \to [0, 1]$ and $\gamma^+ : \mathcal{C} \to [0, 1]$ measuring how good a candidate the region would be respectively for a minimal or a maximal area of the $\widehat{\text{qDII}}$ map. Each of these two scores is computed using the following elementary score functions, which only use the region descriptors:

- A central zone exclusion score $\gamma_{CZ}$ impeding the regions from lying within the central zone of the meristem, defined as:

$$\gamma_{CZ}(c_l) = \begin{cases} 1 & \text{if } \|pos(c_l)\| > \frac{r_c}{2} \\ 0 & \text{otherwise.} \end{cases} \tag{55}$$

- An area score $\gamma_{area}$ that is closer to one when the considered region is large, defined as:

$$\gamma_{area}(c_l) = 1 - \frac{1}{area(c_l)}. \tag{56}$$

- An extremality score $\gamma_{ext}$ that is closer to one when the considered region has a large value of extremality, defined as:

$$\gamma_{extr}(c_l) = \min\left(\frac{1}{2} + extr(c_l), 1\right). \tag{57}$$

- A signal value score, $\gamma_{qDII}$ that is closer to one when the value of $\widehat{qDII}$ is low for minimal regions and high for maximal regions (saddles being considered as maximal regions of lower signal), defined as:

$$\gamma_{qDII}(c_l) = (1 - qDII(c_l))\delta_{type(c_l),-1} + (0.2 + qDII(c_l))\delta_{type(c_l),0} + (qDII(c_l))\delta_{type(c_l),1} \tag{58}$$

where $\delta$ is the Kronecker symbol.

We compute the significance scores $\gamma^-$ and $\gamma^+$ as a multiplicative combination of these elementary scores to obtain maximal values respectively for large, marked valleys of low $\widehat{qDII}$ value and large, marked ridges or marked saddles of high $\widehat{qDII}$ value outside the middle of the CZ (**Appendix 3—figure 4C**). Therefore we define the significance scores of extremal regions as:

$$\begin{aligned}
\gamma^-(c_l) &= \gamma_{extr}(c_l) \cdot \gamma_{CZ}(c_l) \cdot \gamma_{qDII}(c_l) \cdot (\delta_{type(c_l),0} + \gamma_{area}(c_l) \cdot \delta_{type(c_l),1}). \\
\gamma^+(c_l) &= \gamma_{extr}(c_l) \cdot \gamma_{CZ}(c_l) \cdot \gamma_{qDII}(c_l) \cdot \gamma_{area}(c_l) \cdot \delta_{type(c_l),-1}
\end{aligned} \tag{59}$$

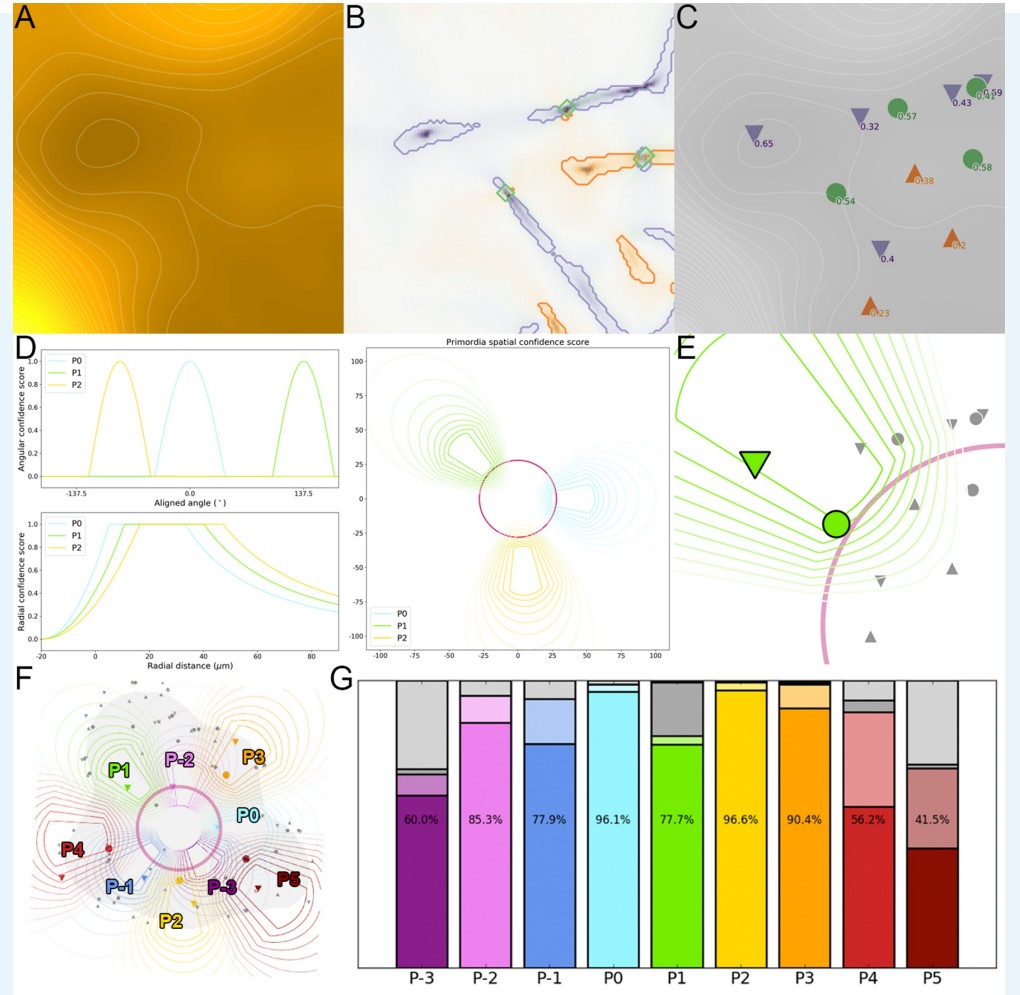

**Appendix 3—figure 4.** Detection of primordia as extremal regions of auxin. (**A**) The auxin distribution forms a landscape where accumulation froms or troughs or valleys in the qDII map, whereas depletion forms peaks or ridges. (**B**). Extremal regions (valleys in purple, ridges in orange and saddles in green) are identified from the qDII map. (**C**). Each region is associated with a unique spatial position and characterized by a significance score. (**D**). A geometrical model of primordia distribution is used to compute a confidence score for each extremal region, relatively to each considered primordium rank, depending on its spatial postion. (**E**). The primordium rank $k$ is assigned to the extremal regions with the highest combination of significance and spatial confidence score relatively to $\mathbb{P}_k$ (**F**). This allows to identify the auxin accumulation zoned and the potential auxin depletion zones associated with each primordium in the considered SAM. (**G**). Comparison of automatically detected auxin extremal points with expertized ones demonstrate a very accurate detection between ranks P0 and P3, with a decreased performance when the features are less well defined (no absolute minimum before P0, several maxima after P4). Color indicates the rank of primordia. Filled color indicates accurate detection, light color indicates correct detection but inaccurate location, dark grey indicates false negatives, light grey indicates false positives.

## Geometrical model of primordia organization

To help predict the position of primordia, we used a simple a priori geometrical model of primordia spatial organization in a 2D projected space. This model associates each point $(r, \theta) \in \mathbb{R}^2$ expressed in polar coordinates with a score $\gamma_k(r, \theta)$ reflecting the confidence of finding the $k^{\text{th}}$ primordium at the position $(r, \theta)$.

This score is decomposed into an angular score $\gamma'_k(\theta)$ and a radial score $\gamma''_k(r)$ and such that $\gamma_k(r,\theta) = \gamma'_k(\theta) \cdot \gamma''_k(r)$. The angular score makes the assumption that pimordia are organized in a regular spiral of divergence angle $\alpha^* = \frac{2\pi}{\phi^2} \sim 137.5°$, and is therefore close to one when the angular coordinate of a 2D point is close to the theoretical angle $k \cdot \alpha^*$ of the $k^{\text{th}}$ primordium, with an angular tolerance $\Delta\theta(k)$ (**Appendix 3—figure 4D**), so that:

$$\gamma'_k(\theta) = \cos\left( \min\left( \pi \cdot \frac{|\theta - k \cdot \alpha^*|}{\Delta\theta(k)}, \frac{\pi}{2} \right) \right). \tag{60}$$

The radial score makes the assumption that primordia move radially so that they are more likely to be found in a range of radial distances $[r_{\min}(k), r_{\max}(k)] \subset \mathbb{R}^+$ that gradually increases with primordium rank $k$ (**Appendix 3—figure 4D**). We define the radial score $\gamma''_k$, so that it is equal to one within the range and decreasing outside, by:

$$\gamma''_k(r) = \min\left( \left(\frac{r_{\max}(k)}{r}\right)^2, 1 \right) \cdot \min\left( \left(\frac{r}{r_{\min}(k)}\right)^2, 1 \right). \tag{61}$$

We define the radial distance range as a relative measure on the CZ radius $r_{\mathbf{c}}$, so that $r_{\min}(k) = \rho_{\min}(k) \cdot r_{\mathbf{c}}$ and $r_{\max}(k) = \rho_{\max}(k) \cdot r_{\mathbf{c}}$. The relative distances $\rho_{\min}$ and $\rho_{\max}$ are affine functions of the primordium rank $i$, such that:

$$\begin{aligned} \rho_{\min}(k) &= \rho^0_{\min} + \rho^1_{\min} \cdot k \\ \rho_{\max}(k) &= \rho^0_{\max} + \rho^1_{\max} \cdot k. \end{aligned} \tag{62}$$

The product of these two scores results in the delineation of smooth windows in the 2D space corresponding to the areas of high likelihood of encountering the primordium $\mathbb{P}_k$ (**Appendix 3—figure 4D**).

## Assignment of primordia to extremal regions

We consider that primordia are characterized by the following specifications:

- A primordium $p$ is necessarily associated with the existence of a valley in $\mathcal{C}$ and possibly with the additional existence of a ridge or a saddle in $\mathcal{C}$.
- A primordium $p$ must respect several conditions:

The valley associated with $p$ must locally be the most significant minimal region, that is have a locally maximal value of $\gamma^-$.
If $p$ is labelled $\mathbb{P}_k$, the valley point associated with $p$ must have a high spatial confidence score $\gamma_k$.
The ridge or saddle point associated with $p$ must lie between the center of the meristem and the radial position of the valley point.
The ridge or saddle associated with $p$ must locally be the most significant maximal region, that is have a locally maximal value of $\gamma^+$.
If $p$ is labelled $\mathbb{P}_k$, the ridge or saddle point associated with $p$ must have a high spatial confidence score $\gamma_k$.

For each primordium rank $k \in [\![k_{\min}, k_{\max}]\!] \subset \mathbb{Z}$, we make the decision of assigning the label $\mathbb{P}_k$ to a valley region $c_l \in \mathcal{C}$, if the region achieves the highest combined score $\gamma^-_k(c_l) = \gamma_k(pos(c_l)) \cdot \gamma^-(c_l)$ among the extremal regions, and that this score is greater than a threshold $\gamma_{\min}$:

$$l^-(k) = \begin{cases} \underset{l \in [\![0, K_C[\![}{\arg\max} \left( \gamma^-_k(c_l) \right) & \text{if} \quad \underset{l \in [\![0, K_C[\![}{\max} \left( \gamma^-_k(c_l) \right) > \gamma_{\min} \\ \emptyset & \text{otherwise.} \end{cases} \tag{63}$$

When the valley point exists ($l^-(k) \neq \emptyset$) we use its position to restrict the search of a ridge or saddle point to the area located between the center the center and the valley region $c_{l^-(k)}$ by defining an additional score function $\gamma^\pm_k : \mathcal{C} \to \{0,1\}$ as:

$$\gamma_k^{\pm}(c_l) = \begin{cases} 1 & \text{if } \|pos(c_l)\| < \|pos(c_{l^-(k)})\| \\ 0 & \text{otherwise.} \end{cases} \tag{64}$$

In a second time, we make the decision of assigning the label $\mathbb{P}_k$ to a ridge or saddle region, if it achieves the highest combined score $\gamma_k^+(c_l) = \gamma_k(pos(c_l)) \cdot \gamma^-(c_l) \cdot \gamma_k^{\pm}(c_l)$ among the extremal regions, and that this score is greater than $\Gamma_{\min}$:

$$l^+(k) = \begin{cases} \underset{l \in [\![0,K_C[\![}{\operatorname{argmax}}\left(\gamma_k^+(c_l)\right) & \text{if } \underset{l \in [\![0,K_C[\![}{\max}\left(\gamma_k^+(c_l)\right) > \gamma_{\min} \\ \emptyset & \text{otherwise.} \end{cases} \tag{65}$$

Each identified primordium extremal point is then associated with a spatial position $(r_i^-, \theta_i^-)$ or $(r_i^+, \theta_i^+)$ (**Appendix 3—figure 4F**) which can be used to compute estimates of all the spatial signals (starting with $\widehat{qDII}$) and to track signals at the level of organ primordia.

## Evaluation of primordia detection

On all the meristems presented in the article, we preformed a manual correction of the primordium assignment of auxin extremal points, to make sure that we recover information from biologically meaningful locations. The assignment of primordium ranks was made on the detected extremal regions $\mathcal{C}$ so that manual and automatic assignment could be compared quantitatively.

This manual labeling allowed us to perform an evaluation of the detection method at the scale of the whole population of meristems (N = 63). We evaluated the performance by counting:

The number $VP$ of correctly assigned primordia extremal points (corresponding to the case where the primordium rank $k$ has been assigned to an extremal region in the automatic method whereas it has been assigned to the same one in the manual labelling)

- The number $MP$ of falsely assigned primordia extremal points (corresponding to the case where the primordium rank $k$ has been assigned to an extremal region in the automatic method whereas it has been assigned to a different one in the manual labelling)
- The number $FP$ of falsely detected primordia extremal points (corresponding to the case where the primordium rank $k$ has been assigned to an extremal region in the automatic method whereas it has not been assigned to any region in the manual labelling)
- The number $FN$ of falsely undetected primordia extremal points (corresponding to the case where the primordium rank $k$ has not been assigned to any region in the automatic method whereas it has been assigned to an extremal region in the manual labelling)

The measure we used is the Jaccard index $j = \frac{VP}{VP+MP+FP+FN}$ and our evaluation lead to an overall average of 75.4% correct detection, rising to 87.3% when considering only primordia ranks between $\mathbb{P}_{-2}$ and $\mathbb{P}_3$ (**Appendix 3—figure 4G**). More precisely, both the auxin accumulation and depletion zones of primordia $\mathbb{P}_0$, $\mathbb{P}_2$ and $\mathbb{P}_3$ are remarkably well detected (nearly 95% correct detection).

The loss of performance at $\mathbb{P}_1$ comes from the common missed detection of the depletion zone that only starts forming at this stage, and is only visible as a saddle point in the $\widehat{qDII}$ map. For the rest, most of the errors are at early stages where auxin accumulation is not very marked or at later stages where the organs get larger, the auxin landscape more complicated (leading to a large number of false assignments) and curvature more important (making the 2D projection less relevant, especially at the tip of the organ where auxin accumulation takes place).

Data presented in **Figure 1G**, **Figure 2D–E**, **Figure 3J**, **Figure 1—figure supplement 2D** and **Figure 1—figure supplement 3** were obtained using the manually corrected positions of organ primordia.

## L1 dynamic signal maps

### Developmental state estimation

We assume that all the observed meristems develop at a comparable rate of 2 new organs per day, corresponding to a plastochron of 12 hr, and that they can, at the first order, be considered synchronous (similarly labeled organ primordia being at comparable developmental stages. Consequently we defined a developmental state indexation as:

$$\tau_i = \frac{t_i}{12},$$

(66)

where $i \in [\![0, K_t[\![$ and $t_0 = 0h$.

### Dynamic signal maps

At the scale of one time-lapse sequence, for which we have computed the 2D aligned signal maps $\left\{ \widehat{S}_i \mid i \in [\![0, K_t[\![ \right\}$ we approximate the continuous 2D+t signal using a density function of developmental state $\eta$ defined by:

$$\eta(\delta\tau) = \frac{1}{2} - \frac{1}{2} tanh(k_\tau \cdot (\delta\tau - R_\tau)).$$

(67)

At any developmental state position $\tau$, the estimated map is a weighted average of single time maps, where the weights are time-distance-based density coefficients computed using the $\eta_\tau$ density function computed with a time radius $R_\tau$ and a time slope $k_\tau$:

$$\widehat{S}(r, \theta, \tau) = \frac{1}{\sum_{i=0}^{K_t - 1} \eta_\tau(|\tau_i - \tau|)} \sum_{i=0}^{K_t - 1} \eta_\tau(|\tau_i - \tau|) \cdot \widehat{S}_i(r, \theta)$$

(68)

### Temporal extrapolation of dynamic maps

We evidenced that looking $p$ plastochrons further in time is equivalent to rotating the system of the angle $\theta_p$ corresponding to the direction of the primordium $\mathbb{P}_p$ (see **Figure 1—figure supplement 2G-H**). We use this the spatio-temporal periodicity of the system to extrapolate the evolution of signals beyond the duration of acquisitions.

Considering a range of primordia stages $[\![p_{min}, p_{max}]\!] \subset \mathbb{Z}$, we use the angles $\theta_p^-$ estimated with 2D primordium detection to apply rotations on the computed maps and derive a dynamic map that covers a larger temporal range as:

$$\widehat{S}(r, \theta, \tau) = \frac{\sum_{i=0}^{K_t - 1} \sum_{p=p_{min}}^{p_{max}} \eta_\tau(|(\tau_i + p) - \tau|) \cdot \widehat{S}_i\left(r, \theta + \theta_p^-\right)}{\sum_{i=0}^{K_t - 1} \sum_{p=p_{min}}^{p_{max}} \eta_\tau(|(\tau_i + p) - \tau|)}.$$

(69)

Ultimately, if instead of using all time points of a given sequence, we use the maps of all time points from all the sequences in the previous formula, we reconstruct the average map $\widehat{S}$ over a population of meristems. Such a map combines all the quantitative spatio-temporal information into one dynamic map reflecting the canonical behavior of the system. The result is presented for the Auxin signal in **Video 1**, where it was obtained with the parameter values $R_\tau = 0.1$, $k_\tau = 2$, $p_{min} = -3$ and $p_{max} = 5$.

### Implementation details

- Microscopy image preparation
The confocal images saved as CZI files through the ZEN software (**ZEISS International, 2012**) of the LSM-710 microscope are opened using a Python script (**Gohlke, 2012**) and split into independent channels that are saved separately as INR image files. This operation preserves all the

information contained in the raw image format. In the specific case of acquisitions for which both *pPIN1:PIN1-GFP* (PIN1) and *pRPS5a:DII-VENUS* (DII) are imaged, the close emission wavelengths causes the PIN1-stained cell membranes to appear in the DII nuclei images. In that only case, the PIN1 signal intensity is subtracted from the DII image channel before saving the file.

We use the polygonal selection tool of the ImageJ software to manually define a region of interest in all the slices of every image, and doing so, digitally dissect the outermost organs to get meristem images with at most six visible organs. This binary mask is then applied on all the channels, and masked channels are saved in separate INR files.

- Nuclei detection and quantification

The nuclei detection algorithm has been implemented in the tissue_nukem_3d Python library using the Gaussian filtering functions provided by the SciPy library (*Virtanen et al., 2020*) for the construction of the scale space.

The surface mesh used for the extraction of L1 nuclei was computed using the implementations provided in the VTK (Visualization ToolKit) library (*Schroeder et al., 2006*) for the Marching Cubes algorithm, the mesh smoothing and the quadric decimation (vtkImageMarchingCubes, vtkWindowedSincPolyDataFilter, vtkQuadricDecimation). A version of the isotropic remeshing algorithm has been implemented in the cellcomplex library.

- Image registration

The image registration operations are performed using the blockmatching computing library that comes embedded in the timagetk (Tissue Image ToolKit) image processing library dedicated to 3D microscopy images of multicellular tissues. Registered images are generally saved channel by channel in separate INR files, as well as the vector fields computed in the case of non-linear registration.

- Continuous map generation

The methods for generating continuous maps of epidermal signal, and to detect extremal regions on such maps were implemented in the tissue_nukem_3d Python library using the Gaussian derivative functions provided by the SciPy library (*Virtanen et al., 2020*) for the computation of gradients.

## Quantitative analysis of PIN1 polarity

To study the cellular polarities of the auxin efflux carrier *PIN1*, we quantify information at the interface between cells, and the resulting data is best expressed as vectors. Therefore, it requires a whole different set of computational steps to analyze PIN1 polarities, and integrate polarity information at increasing levels:

- Cell interface level: quantification of interface polarity from image intensities:

To quantify PIN1 polarity, we need to consider pairs of neighboring cells, and it is therefore necessary to reconstruct cell adjacency information from the images. The use of images with a cell wall staining allows partitioning the image grid into contiguous regions of voxels representing cells, which will give us neighborhood information as well as a way to reconstruct the geometry of cell interfaces. Based on these geometries, we need to compute a local PIN1 polarity at the scale of cell interfaces using the local signal intensity of the image.

- Cell level: integration of cell interface polarities into cell polarity vectors:

To match the rest of the data that is defined at cell level, we need to convert the scalar information of polarization for each cell interface into a cellular information. The directional contributions from the different interfaces surrounding a cell necessarily result into a vectorial data at the level of the cell. Then, by applying the geometrical transform estimated as in Appendix B, we have to align not only the spatial positions of cellular objects but also the directions of cell polarity vectors.

- Tissue level: computation of a continuous vector map of polarities that allows population averaging:

Finally, in order to identify global trends in the PIN1 polarity patterns across individuals, we need to use local averaging to build a continuous representation of signal that allows point-wise comparison, using the 2D map formalism. The only difference is that, in the case of polarities, the data we average consists of vectors. The result is therefore a continuous vector map on which we can compute other scalar properties, such as norm or divergence.

This alternative pipeline is illustrated in *Appendix 4—figure 1*, and it results in the continuous polarity maps shown in *Figure 3I* and *Figure 3—figure supplement 2*. Its development led to the introduction of an original method for the quantification of cell interface transporter polarity from images in 3D, which has been evaluated using super-resolution (using radial fluctuation [*Gustafsson et al., 2016*]) acquisitions of the same tissues (see Evaluation of cell interface polarity estimation). The different computational steps involved in the pipeline are detailed in the following sections:

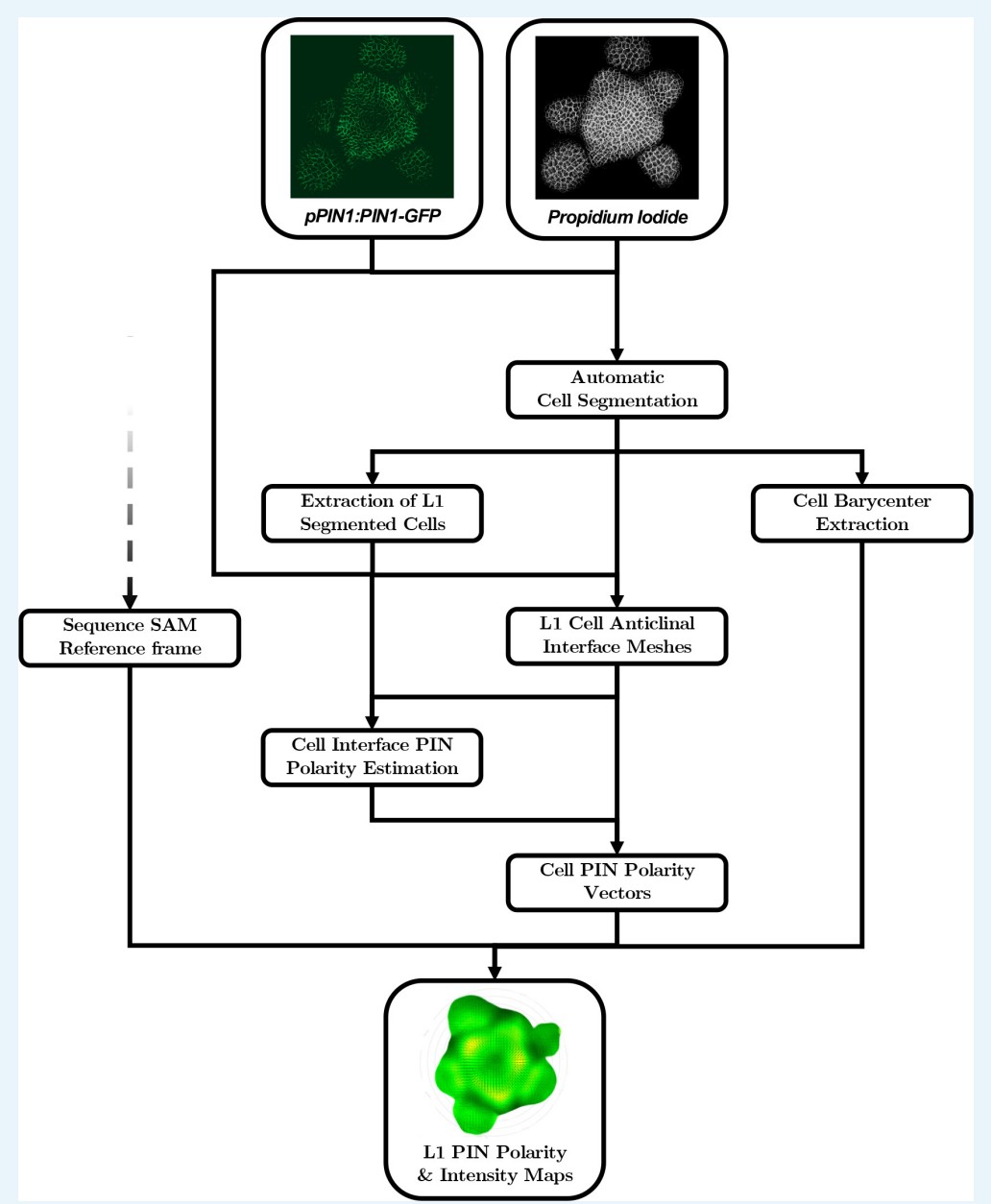

**Appendix 4—figure 1.** Automatic quantification pipeline for the time-lapse microscopy images of PIN1 auxin efflux carrier. The quantitative estimation of PIN polarity relies on the analysis of cell wall and membrane-marker images that need to be processed in a different way than nuclei marker images. An alternative automatic pipeline performs the necessary steps, from the segmentation of the cells and the extraction of L1 cell anticlinal interfaces to the quantification of signal distribution at each cell interface and the reconstruction of the polarized cell network.

## Automatic segmentation of membrane images

In order to be able to quantify membrane-localized signal, we need a segmentation of the tissue at the cellular level. Such automatic cell segmentation procedures are often limited by their capacity to detect the right number of 'seeds' prior to the segmentation of the image according to membrane localized signal.

In our case, the presence of a constitutive nuclei targeted signal (*pRPS5a:TagBFP*), allows to compute the nuclei coordinates in the image. It is thus possible to use these coordinates as

seeds to initialize a watershed segmentation algorithm. The quality of the obtained segmentation, in terms of 'correct number of cells detected', is then directly linked to the nuclei detection quality. Compared to parametric seed detection by methods such as local minima detection (by h-transform algorithm) followed by connected component labeling, the use of detected nuclei signal coordinates allows to reduce the over and under segmentation problems (data not shown). To summarize the pipeline used for this automatic cell wall segmentation step, we performed:

- An adaptative histogram equalization for all z-slices of the membrane stacks to improve and normalize the contrast (*Pizer et al., 1987*);
- Isometric resampling to a voxelsize of (0.2, 0.2, 0.2μ m), when original images are (0.2, 0.2, 0.5μ m), to perform Gaussian smoothing and obtain smoother segmentation along the $z$ axis (*Fernandez et al., 2010*);
- Gaussian smoothing of the cell wall intensity image, with $\sigma = 0.2 \mu m$ to reduce noise in the image when performing watershed segmentation (*Fernandez et al., 2010*);
- Create a *seed image* from the nuclei coordinates to initialize the watershed algorithm (*Fernandez et al., 2010*);
- Run the seeded-watershed algorithm with isometric smoothed intensity image and seed image (*Fernandez et al., 2010*).

No post-segmentation corrections where performed, no cell-fusion (in case of over-segmentation) or morphological corrections (median filters to smooth the walls). In the end we obtain a segmented image $I_\mathrm{seg}$ that assigns an integer label to every voxel of the image grid $\Omega_I$ on which the image $I_\mathrm{PI}$ to segment is defined (see Appendix B). The cells of the tissue are represented by independent connected regions of voxels (so that the same label can not be assigned to voxels that are not part of the same connected component of $I_\mathrm{seg}$). The background corresponds to a specific label, that is systematically set to one to ensure consistency between images. Each cell labeled $c \in [\![1, K_c]\!]$ is the then represented by a connected region $\Gamma_c$ so that:

$$\Gamma_c = \left\{ \mathbf{x} \in \Omega_I \mid I_\mathrm{seg}(\mathbf{x}) = c \right\} \tag{70}$$

## Cell barycenter extraction

To obtain the cells barycenter, we estimate the position $P_c$ of the center of the cell labeled $c$ in the segmented image $I_\mathrm{seg}$ as the average of the voxel coordinates of the cell region $\Gamma_c$, by $\forall c \in [\![1, K_c]\!] \setminus \{1\}$,

$$P_c = \frac{1}{|\Gamma_c|} \sum_{\mathbf{x} \in \Gamma_c} \mathbf{x}. \tag{71}$$

## Extraction of L1 segmented cells

### Definition of cell adjacency

Adjacency relationship is defined through the notion of surface of contact between two regions in the segmented image. In a first time, we consider the 6-connectivity to define neighborhoods at voxel-scale. However, to be robust to potential segmentation errors, we estimate the area of all surfaces of contact representing walls between two cells $c$ and $c'$, and consider as neighbors only those for which the area is greater than a given threshold $A_\mathrm{min}$.

We compute the area of contact $A(\Gamma_c, \Gamma_{c'})$ as the sum of areas of the contact rectangles between pairs of 6-adjacent voxels so that one belongs to $\Gamma_c$ and the other to $\Gamma_{c'}$. An analysis of the distribution of wall areas (data not shown) led us to consider that a value of $A_\mathrm{min} = 5 \mu m^2$ would be suitable. We note $\mathcal{A}_c$ the cells that are adjacent to the cell $c$, namely the labels that verify the surface of contact condition:

$$\mathcal{A}_c = \{c' \in [\![1, K_c]\!] \setminus \{c\} \mid A(\Gamma_c, \Gamma_{c'}) > A_{\min}\} \tag{72}$$

## Estimation of $L_1$ layer

To be able to automatically determine to which layer a given cell belongs to, we use the topology of the tissue, notably the background region $\Gamma_1$. It is the definition of the epidermis to be in contact with the outside world, and all the regions of cells belonging to the $\mathcal{L}_1$ set should therefore be adjacent to $\Gamma_1$. In some problematic cases, the segmentation algorithm can produce a background region that reaches the inside of the tissue, but we assumed that it was not the case in our images and that we could consider that $\mathcal{L}_1 = \mathcal{A}_1$.

## Meshing L1 anticlinal cell interfaces

In order to quantify the PIN signal intensity at the level of each individual cell-cell surface of contact, we need to have a precise identification of the cell interfaces and a faithful 3D representation to describe their position and orientation. If the boundary between two cells can be extracted as a set of voxels in the segmented image $I_{\text{seg}}$, it will generally be too sensitive to noise and to image resolution to be used as such. We chose therefore to use a triangular mesh representation with a high resolution to represent accurately the cell interfaces.

To obtain such meshes, we apply the Marching Cubes algorithm (**Lorensen and Cline, 1987**) to each cell labelled $c$ and represented by its connected region $\Gamma_c$ of identically labeled voxels in the segmented membrane image $I_{\text{seg}}$. This produces a triangular mesh $\mathcal{M}_c = (\mathcal{V}_c, \mathcal{T}_c)$ where:

- $\mathcal{V}_c$ is a set of vertices
- Every vertex $v \in \mathcal{V}_c$ is associated with a 3D coordinate $M_v \in \mathbb{R}^3$
- $\mathcal{T}_c$ is a set of triangular faces $t = (v_1, v_2, v_3) \in \mathcal{V}_c^3$ linking those vertices.

The cell meshes are generally closed (except on image borders) and have a voxel-like resolution.

Using Marching Cubes ensures us that, in an 8-voxel cube with only two labels $c$ and $c'$ (which typically occurs at the interface between two cells) the algorithm will create the same vertices whichever label is considered as 1 or 0. In other words, we know that two cells that are neighbors in the image (with a large enough surface of contact) will have common vertices in their mesh reconstructions $\mathcal{M}_c$ and $\mathcal{M}_{c'}$. We use this property to construct the cell interface mesh $\mathcal{M}_{c,c'} = (\mathcal{V}_{c,c'}, \mathcal{T}_{c,c'})$ of the interface between $\Gamma_c$ and $\Gamma_{c'}$ as the restriction of one of the two meshes to the set of common vertex points:

$$\left\{ \mathcal{V}_{c,c'} = \{v \in \mathcal{V}_c \mid \exists v' \in \mathcal{V}_{c'}, \|M_v - M_{v'}\| < \epsilon\} \quad \mathcal{T}_{c,c'} = \{t \in \mathcal{T}_c \mid \forall v \in t, v \in \mathcal{V}_{c,c'}\} \right. \tag{73}$$

Each interface mesh undergoes then a phase of triangle decimation (**Garland and Heckbert, 1997b**) and isotropic remeshing (**Botsch and Kobbelt, 2004b**) to obtain a regular surface so that the typical length of a triangle edge is close to 0.5μm, which is about the voxel characteristic dimension. On the triangular mesh, we estimate the normal vectors $\mathbf{n}_{c \to c'}(v)$ at each vertex $v$ of $\mathcal{M}_{c,c'}$, ensuring they all point from $c$ to $c'$, and the area of each triangle that allows us to estimate the total area $A_{c,c'} = A_{c',c}$ of the interface between cells $c$ and $c'$. Note that the Marching Cubes intersection, along with the decimation and smoothing, will produce a mesh that is smaller that the actual cell interface (the intersecting part does not extend to cell edges) and the interface area will be underestimated. On the other hand, the voxel-based area estimation $A(\Gamma_c, \Gamma_{c'})$ is known to be largely overestimating, which ultimately provides a way to have both lower and upper bound estimates of the interface areas.

## Quantification of PIN polarity at cell interfaces

PIN proteins are polarly distributed in cells. This potentially induces that on both sides of a given interface between two cells $c$ and $c'$, different concentrations on PIN can be observed. This suggests that a flow of auxin could be oriented from $c$ to $c'$ if more PIN transporters are located at this interface in the cell membrane of $c$ than in $c'$.

This cell interface polarity orientation is denoted $p_{c \to c'} = -p_{c' \to c} \in [-1, 1]$ and is equal to 1 (respectively $-1$) when there exists a marked positive (respectively negative) difference between the PIN membrane concentrations of $c$ and $c'$ at their interface. If the case where there is no difference, $p_{c \to c'} = 0$, and a value between 0 and 1 reflects intermediate difference levels.

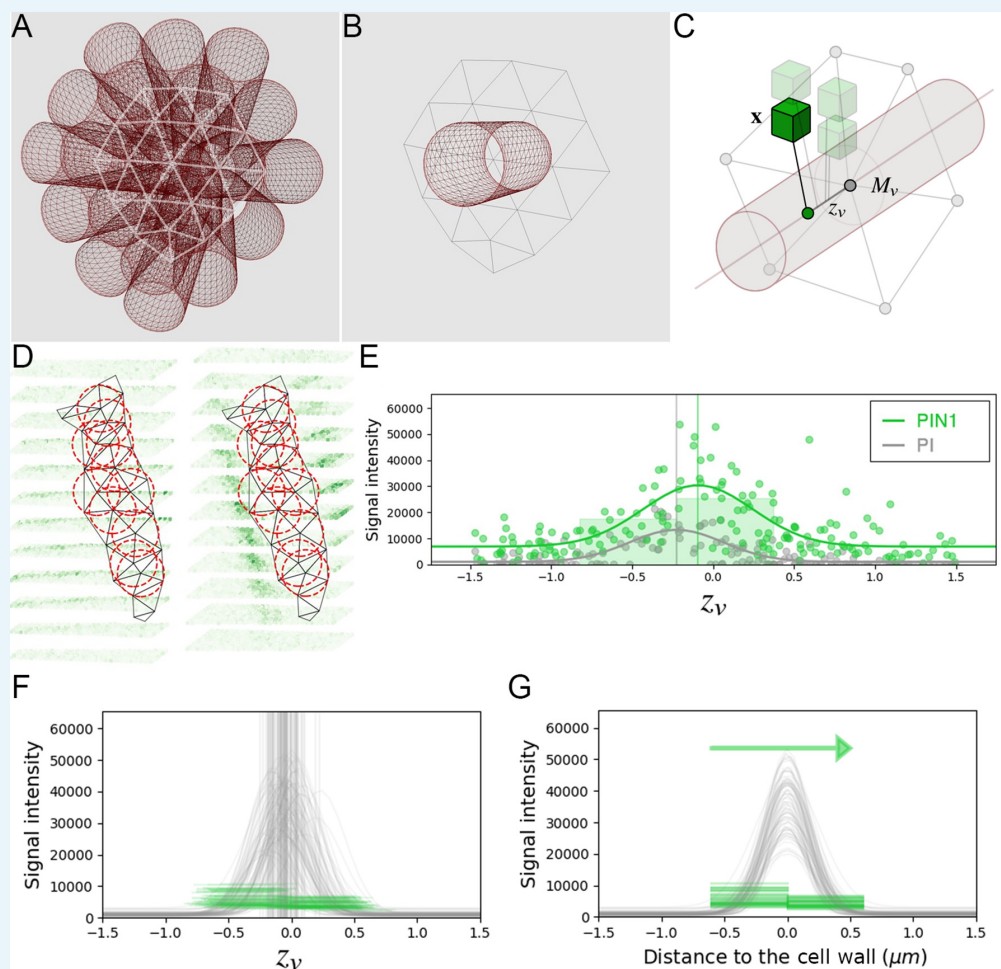

**Appendix 4—figure 2.** Estimation of PIN polarity at cell interface level. (**A**) The triangular mesh representing the cell interface is used to generate a set of 3D cylinders locally orthogonal to the interface and placed at each vertex of the inside of the interface. (**B**) The radius of the cylinders is close to the typical resolution of the mesh. (**C**) For each cylinder, all the neighboring image voxels are projected onto its main axis, and kept if within the distance defined by the cylinder radius. Distance of the projected voxel to the interface mesh vertex is used as abscissa for the evaluation of 1d polarity (**D**) The set of all interface cylinders allow sampling the PIN image signal on either side of the cell-wall at different locations. (**E**) The 1-dimensional distributions of both PI and PIN image signals along each cylinder are used to locate precisely the cell-wall position and quantify signal levels on either side. (**F**) PIN levels are estimated left and right of the detected cell-wall abscissa up to a fixed distance of 0.6µ m. (**G**). Significant difference between left and right distributions of PIN levels across the interface allows deciding for local PIN polarity at the scale of the cell interface.

To estimate this polarity, we aim to sample the PIN signal on each side of the cell-wall separating $c$ and $c'$. For this, we proceed in several steps:

- On each vertex $v$ of $\mathcal{M}_{c,c'}$, we position a cylinder of radius $r_C$ and height $2d_C$, centered on $M_v$ and whose main axis is defined by the local normal vector $\mathbf{n} = \mathbf{n}_{c \to c'}(v)$ (**Appenix 4—figure 2B-D**). We excluded vertices located on the contour of the mesh where the estimation of $\mathbf{n}$ is less robust.
- With each voxel $\mathbf{x} \in \Omega_I$ in the cylinder, we associate a signed abscissa $z_v(\mathbf{x}) = \langle \mathbf{x} - M_v, \mathbf{n} \rangle$ (**Appenix 4—figure 2C**) that allows positioning it on the 1D axis of the cylinder.
- Due to processing errors, the actual cell-wall (marked by PI) may be shifted from the position $z_v = 0$ where the cylinder intersects the interface mesh. We then consider the spatial distribution of PI intensities along the cylinder axis and model it using a parametric Gaussian function:

$$\mathrm{PI}(z_v) = \mathrm{PI}_0 + \mathrm{PI}_1 e^{-\frac{(z_v - z_{\mathrm{PI}})^2}{\sigma_{\mathrm{PI}}^2}}. \tag{74}$$

We estimate the parameters of this function by a least-squares optimization algorithm and use the value of $z_{\mathrm{PI}}$ as a reference defining the actual position of the cell-wall (**Appenix 4—figure 2E**).

- Finally, we compute the average of PIN intensities of voxels $\mathbf{x}$ such that $z_{\mathrm{PI}} - z_{\max} < z_v(\mathbf{x}) < z_{\mathrm{PI}}$ (respectively $z_{\mathrm{PI}} < z_v(\mathbf{x}) < z_{\mathrm{PI}} + z_{\max}$) to estimate the value $\mathrm{PIN}_{c \to c'}(v)$ (respectively $\mathrm{PIN}_{c' \to c}(v)$) of PIN signal in cell $c$ (respectively $c'$) at the level of vertex $v$. Note that the definition is symmetrical, meaning that the result would be exactly the same if $c$ and $c'$ were permuted.

By performing this two-sided estimation on every cylinder over the triangular mesh representing the cell interface, we end up with two paired signal distributions $\{\mathrm{PIN}_{c \to c'}(v)\}$ and $\{\mathrm{PIN}_{c' \to c}(v)\}$. We test statistically if one side of the interface bears significantly more signal that the other by performing two one-sided Wilcoxon tests, a non-parametric version of the T-test for paired samples. They assess respectively if the means of the distributions (noted $\mathrm{PIN}_{c \to c'}$ and $\mathrm{PIN}_{c' \to c}$) can be ranked, that is if $\mathrm{PIN}_{c \to c'} > \mathrm{PIN}_{c' \to c}$ and if $\mathrm{PIN}_{c' \to c} > \mathrm{PIN}_{c \to c'}$ respectively. The tests provide two p-values $\mathrm{pval}_{c \to c'}$ and $\mathrm{pval}_{c' \to c}$ based on which we make a decision on the existence of a polarity. We do not use a binary polarity value but define a polarity value $p_{c \to c'}$ whose sign corresponds to the direction of polarity and absolute value lies among 1, 0.5, 0.25 and to account for situations where some uncertainty remains from the statistical tests:

$$p_{c \to c'} = \begin{cases} \begin{cases} 1 \text{ if } \mathrm{pval}_{c \to c'} < 0.1 \\ 0.5 \text{ if } \leq \mathrm{pval}_{c \to c'} < 0.15 \\ 0.25 \text{ if } 0.15 \leq \mathrm{pval}_{c \to c'} < 0.25 \end{cases} & \text{if } (\mathrm{pval}_{c \to c'} < 0.25) \text{ and } (\mathrm{pval}_{c' \to c} > 0.25) \\ \begin{cases} -1 \text{ if } \mathrm{pval}_{c' \to c} < 0.1 \\ -0.5 \text{ if } 0.1 \leq \mathrm{pval}_{c' \to c} < 0.15 \\ -0.25 \text{ if } 0.15 \leq \mathrm{pval}_{c' \to c} < 0.25 \end{cases} & \text{if } (\mathrm{pval}_{c' \to c} < 0.25) \text{ and } (\mathrm{pval}_{c \to c'} > 0.25) \\ 0 & \text{otherwise} \end{cases} \tag{75}$$

Note that the definition is symmetrical ensuring that $p_{c \to c'} = -p_{c' \to c}$. We also define the intensity of PIN signal at the level of the cell interface $\mathrm{PIN}_{c,c'} = \mathrm{PIN}_{c',c}$ as the average of the signal medians of either side, the barycenter $P_{c,c'}$ of the interface mesh, and interface normal vector $\mathbf{n}_{c \to c'} = -\mathbf{n}_{c' \to c}$ computed as the average of the non-contour interface vertex normals.

## Evaluation of cell interface polarity estimation

We assessed whether this automatic polarity estimation, performed on confocal microscopy images with a typical resolution of 0.2µm (far beyond cell-wall thickness), manages to retrieve a difference of carrier concentration at the level of plasma membranes. To do so, we used super-resolution imaging to provide a set of ground truth polarities that we use to evaluate the result of our automatic method.

Super-resolution images were obtained using the Super-Resolution Radial Fluctuations (SRRF) technique (**Gustafsson et al., 2016**), which produces a 2D image with a resolution of 0.035µm (**Appendix 4—figure 3A**). The same tissue was immediately (0.5 to 2 hr) imaged under the confocal microscope to make sure that the local polarities will be comparable. The confocal image stack was then processed through the automatic pipeline described above. This provides notably unique identifiers to cells and allows to identify the visible cell interfaces by pairs of cell labels that can be matched to the outcome of the pipeline.

Individual cell interfaces were manually annotated by two independent experts in the ImageJ software using the same protocol. We designed a procedure to determine carrier polarities on 2D slices co-imaging membrane-located fluorescent carrier proteins and cell-wall staining inspired from **Shi et al. (2017)**. The following steps are repeated for each cell interface, identified by a pair of cell labels coming from the automatically segmented PI image:

- Draw a minimal amount of 3 lines using the 'Straight line' tool:

Going always from the same cell (left) towards the other (right)
Keeping them orthogonal to the apparent interface plane (**Appendix 4—figure 3B**)
- For each line, sample the PI signal and the PIN1 signal along the line using the 'Plot profile' tool

With a line width set to 5 to 20 pixels for local averaging
- Determine if the peak of PIN1 signal lies clearly and consistently on one side of the peak of PI signal.

If PIN1 if clearly and consistently to the left of PI, assign the polarity 1.
If PIN1 if consistently to the left of PI, but not clearly, assign the polarity 0.5.
If PIN1 if clearly and consistently to the right of PI, assign the polarity $-1$.
If PIN1 if consistently to the right of PI, but not clearly, assign the polarity $-0.5$.
If PIN1 is not consistently on one side of PI, or not clearly, assign the polarity .

Repeating this manual assessment for every pair of neighbor $L_1$ cells visible in the SRRF image plane, each expert filled a data sheet using the same cell labels as in the segmented PI image and the manually determined polarity values. Among the expertized interfaces, we retained only those that were assessed by both experts and that were found in consensus, that is where either:

- Both experts assigned the same polarity value
- One expert assigned the value 1 (resp. $-1$) and the other 0.5 (resp. $-0.5$).

In the latter case, the ground truth polarity value was corrected to 1 (resp. $-1$), acknowledging the certainty given by one expert when both agree on the direction of polarity. This resulted in a set of consensus interface polarities as displayed in **Appendix 4—figure 3C**. In total, these consensus interfaces made up for more than 85% of the interfaces analyzed by both experts as shown in **Appendix 4—figure 3D**, and less than 4% of the interfaces found the experts in disagreement.

We compared these ground truth polarities with the polarity values $p_{c \to c'}$ predicted by the automatic method for the same pairs of cells. For the sake of evaluation, we considered the two lower levels of certainty in the statistical test (corresponding to values 0.5 and 0.25) as the same uncertain value 0.5 in the ground truth assessment. We considered the cell pairs of the expertized interface to be in the order of their assigned polarity, that is such that all the ground truth polarity values $p^*$ are either , 0.5 or 1, which is possible due to the symmetric nature of the polarity values. The comparison translates into a table $\{C(p^*, p) \mid p^* \in \{0, 0.5, 1\}, p \in \{-1, -0.5, 0, 0.5, 1\}\}$ which counts the proportion of cell interfaces manually labelled as $p^*$ and detected as $p$ in the whole population of interfaces. This table is presented in **Appendix 4—figure 3E**.

To obtain the results presented in **Figure 3—figure supplement 1G**, we chose to consider as 'Correct' predictions not only the cases where $p^* = p$, but also when expert and algorithm agree on direction, that is when $p^*$ and $p$ have the same sign, independently of confidence. If we sum up those categories, we end up with more than 72% of cell interfaces being predicted correctly (green squares in **Appendix 4—figure 3E**). Conversely, we considered predictions to

be 'Opposite' to the ground truth when $p^*$ and $p$ have opposite signs (red squares in *Appendix 4—figure 3E*) which only accounts for less than 10% of the interfaces. These 10% of incorrectly predicted interfaces are a problem that may come from an increased sensitivity to image noise with a coarser resolution, but their effect might be mitigated by the integration of interface polarities at cell level (see below), especially since they tend to concern interfaces of smaller area (*Appendix 4—figure 3F*).

The rest of the interfaces was considered as 'Uncertain' as they either consist of interfaces for which the experts determined a polarity when the automatic method did not lead to statistical significance (almost 13% of the interfaces) or the opposite case of detecting a polarity when the experts were consistently uncertain (the remaining 5%). In the first case, the interfaces will have a mild effect on the resulting cell level vectors as they will contribute as a null vector.

This mitigated effect of interface-level polarity errors when we integrate polarity at cell level was evaluated by computing a less quantitative version of the cell polarity vectors than the one presented in the next section. The polarity vector of the cell labelled $c$ is noted $\mathbf{PIN}_c$ and is computed as an area-weighted average of interface normals multiplied by their polarity value:

$$\mathbf{PIN}_c = \frac{\sum_{c' \in \mathcal{A}_c \cap \mathcal{L}_1} A_{c,c'} p_{c \to c'} \mathbf{n}_{c \to c'}}{\sum_{c' \in \mathcal{A}_c \cap \mathcal{L}_1} A_{c,c'}} \tag{76}$$

This allows to compare the directions of the resulting cell vectors. In *Appendix 4—figure 3G–J*, we can see how cell interface polarities that were not correctly predicted (orange circles in *Appendix 4—figure 3H*) lead to cell polarity vectors that have very close directions to those resulting from the ground truth polarities. This preservation of cell polarity vector was studied more extensively on the 83 cells that shared consensus interfaces (*Figure 3—figure supplement 1H*). *Appendix 4—figure 3K* shows that more than 50% of the considered cells have their polarity vector oriented in the exact same direction than the ground truth while than more than 90% were found in qualitatively the same direction (angle error less than 90° difference).

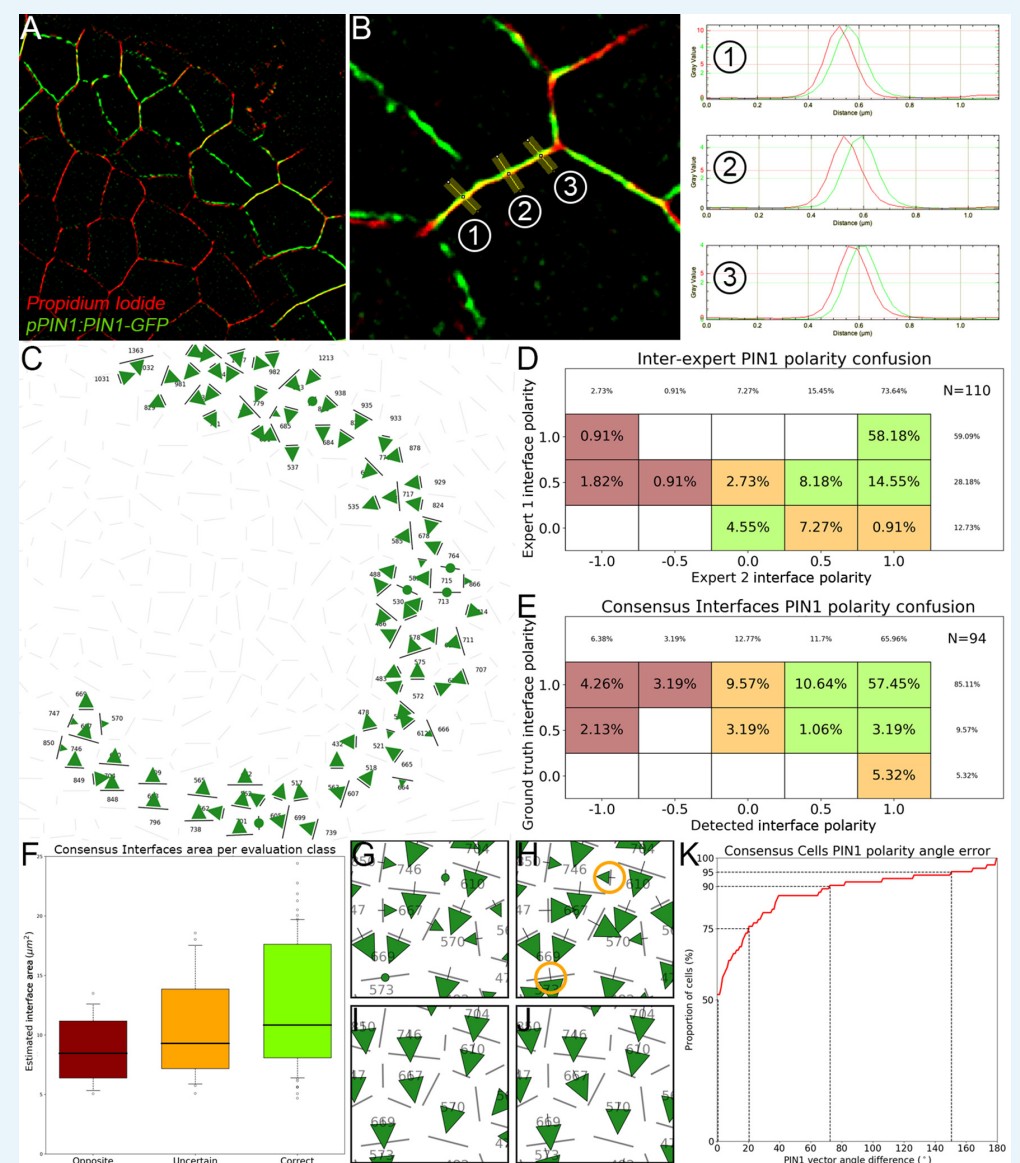

**Appendix 4—figure 3.** Evaluation of cell interface PIN1 polarities using manually annotated SRRF images. (**A**) The super-resolution radial fluctuations method produces a 2D image slice with a greatly improved pixel resolution and a less spread-out signal. (**B**) Using the ImageJ line tool, polarities were manually estimated for each pair of neighbor cells by looking at the relative position of the peaks of PIN1 and PI signals along lines locally orthogonal to the cell interface. In the displayed case, the ground truth polarity value would be -1 as PIN1 lies clearly and consistently to the right of PI. (**C**) The interfaces on which both experts were in consensus form a set of polarity values that we use as a ground truth to evaluate the results of the automatic method. (**D**) Comparison of cell interface polarities assigned by the first expert taken as reference (*y*-axis) with the ones assigned by the second expert (*x*-axis). Each cell corresponds to the percentage of annotated interfaces (N = 110) for which expert one assigned the polarity *y* and expert two the polarity *x*. Agreement between experts is represented in green cells (85%), disagreement in dark red cells (4%) and uncertain cases in orange cells (11%). Empty cells correspond to a percentage 0%. (**E**). Similarly displayed comparison of ground truth polarities taken as reference (*y*-axis) and detected polarities. More than 72% of interfaces have 'Correct' predicted polarities (green cells) and less than 10% have 'Opposite' polarities (dark red cells). 18% of cells have an 'Uncertain' polarity decision (orange cells). (**F**). The interfaces where errors were made appear to be generally smaller ones. (**G–J**).

Local errors between the ground truth interface polarities (**G**) and automatically detected ones (**H**) as those highlighted in orange circles do not necessarily translate into errors between the corresponding polarity vectors integrated at the level of the cells (**I and J**). (**K**). Cumulative histogram of cell-level angle errors, showing a very high angular consistency at cell level despite the proportion of interfaces with opposite polarities.

## Computation of cell polarity vectors

The local cell interface polarity needs to be integrated to the cell level if we want to describe the local directionality of PIN carriers inside the tissue. To do this, we define the polarity of a cell $c$ as a 3D vector $\mathbf{PIN}_c$ that takes into account all the anticlinal cell interfaces surrounding the considered cell. Each interface contributes to the resulting vector proportionally to its area, and if polarized, adds up a directional flux parallel to its normal vector with an intensity equal to the difference of PIN intensities on its sides:

$$\mathbf{PIN}_c = \frac{1}{\sum_{c' \in \mathcal{A}_c \cap \mathcal{L}_1} A_{c,c'}} \sum_{c' \in \mathcal{A}_c \cap \mathcal{L}_1} A_{c,c'} (|\mathrm{PIN}_{c \to c'} - \mathrm{PIN}_{c' \to c}| p_{c \to c'} \mathbf{n}_{c \to c'}) \tag{77}$$

## Polarity vector alignment

To make it possible for the data computed on membrane images to be mapped onto the data from the nuclei channels of the same acquisition, we need to transform the quantified information into the same reference frame. The processes of Rigid time registration and Sequence SAM reference frame determination allow to transform rigidly the images from their original frame to the common reference SAM frame $\mathcal{R}^*$. Therefore, to transform the PIN polarity data into the same referential, we apply this transform to the cell barycenters $P_c$ and to the cell interface centers $P_{c,c'}$ and apply the rotation component of the transformation to the PIN cell polarity vectors $\mathrm{PIN}_{c,c'}$.

## Polarity vector maps

Using the epidermal map formalism, introduced in 2D maps of epidermal signal, and from these transformed coordinates, we can then compute an aligned continuous map of PIN intensity based of the quantification of PIN1 signal at the level of cell interfaces and the estimation of interface areas:

$$\widehat{\mathrm{PIN}}(\mathbf{x}) = \frac{1}{\displaystyle\sum_{\substack{c \in \mathcal{L}_1 \\ c' \in \mathcal{A}_c \cap \mathcal{L}_1}} \eta\left(\|\mathbf{x} - P_{c,c'}^*\|\right) A_{c,c'}} \sum_{\substack{c \in \mathcal{L}_1 \\ c' \in \mathcal{A}_c \cap \mathcal{L}_1}} \eta\left(\|\mathbf{x} - P_{c,c'}^*\|\right) A_{c,c'} \mathrm{PIN}_{c,c'} \tag{78}$$

We also use the same method to compute an aligned vectorial map of PIN polarities, this time based on the estimation of cell polarity vectors:

$$\widehat{\mathbf{PIN}}(\mathbf{x}) = \frac{1}{\sum_{c \in \mathcal{L}_1} \eta\left(\|\mathbf{x} - P_c^*\|\right)} \sum_{c \in \mathcal{L}_1} \eta\left(\|\mathbf{x} - P_c^*\|\right) \mathbf{PIN}_c^* \tag{79}$$

In particular, this is the data we use to quantify the convergence of PIN directions at tissue scale by applying the divergence operator to the vector field resulting from the map computation $\widehat{\mathbf{PIN}}(x,y) = \left(\widehat{\mathrm{PIN}}_x, \widehat{\mathrm{PIN}}_y\right)(x,y)$. This operation gives us a scalar map (as those displayed in *Figure 3—figure supplement 2*, third column) where negative values correspond to areas of local convergence of the PIN directions:

$$\mathrm{div}\,\widehat{\mathbf{PIN}}(x,y) = \frac{\partial \widehat{\mathbf{PIN}}_x}{\partial x}(x,y) + \frac{\partial \widehat{\mathbf{PIN}}_y}{\partial y}(x,y) \tag{80}$$

We use this computed map to approximate the values of $\mathrm{div}\,\widehat{\mathbf{PIN}}$ at punctual locations such as nuclei points $P_n$ or primordia extremal points $\left(r_{\mathbb{P}_i}^-, \theta_{\mathbb{P}_i}^-\right)$ or $\left(r_{\mathbb{P}_i}^+, \theta_{\mathbb{P}_i}^+\right)$ by averaging the values at the four closest grid coordinates (*Figure 3J*). Indeed, it would be very complex to estimate this quantity on a discrete set of points from the $\mathbf{PIN}_c^*$ vectors, on which the divergence operator would be non-trivial.

## Implementation details

### Image processing
Histogram normalization was performed using the implementation provideed in the Scikit-Image library (*van der Walt et al., 2014*), and cell barycenters in the segmented images were computed using the SciPy library (*Virtanen et al., 2020*), made available through the timagetk (Tissue Image ToolKit) image processing library.

### Meshing algorithms
We used the implementations provided in the VTK (Visualization ToolKit) library (*Schroeder et al., 2006*) for the Marching Cubes algorithm, the mesh smoothing and the quadric decimation (vtkImageMarchingCubes, vtkWindowedSincPolyDataFilter, vtkQuadricDecimation). A version of the isotropic remeshing algorithm has been implemented in the cellcomplex library.

### PIN polarity quantification
We have developed specific tools for the quantification of PIN1 polarities, and they are made avaliable in the tissue_paredes library. We used the implementation of Wilcoxon test from the SciPy library (*Virtanen et al., 2020*).

### PIN polarity maps
Continuous maps and vector maps were generated using the functionns implemented in the tissue_nukem_3d Python library. The divergence maps were estimated by applying a 1-D Gaussian derivative filter ($\sigma = 3.75\mu m$) to each component of a vector map in the corresponding dimension, using functions implemented in the SciPy library (*Virtanen et al., 2020*).

## Appendix 5

# Extrapolated cell motion in the developmental continuum

Throughout this work, we strongly relied on the spatio-temporal periodicity of phyllotactic systems, that we demonstrated to be a valid assumption for our considered SAMs. Indeed, the high angular precision and limited plastochron variability make the observed systems close to a steady developing regular phyllotactic system with a divergence angle $\alpha = 137.5° \pm 6.7°$ and a plastochron $T = 12h \pm 2h$ (**Figure 1—figure supplement 2D–G**, Appendix 1).

Notably, we considered that, in the 2D cylindrical reference frame centered on the CZ of the shoot apical meristem, the dynamics of any quantifiable signal $S$ must follow the properties of a spatio-temporally periodic function of spatial period $(0, \alpha, 0)$ and temporal period $T$:

$$\forall \begin{pmatrix} r \\ \theta \\ z \end{pmatrix} \in \mathbb{R}^+ \times [-\pi, \pi] \times \mathbb{R}, \forall t \in \mathcal{T}, \quad S\left( \begin{pmatrix} r \\ \theta \\ z \end{pmatrix}, t + T \right) = S\left( \begin{pmatrix} r \\ \theta \\ z \end{pmatrix} + \begin{pmatrix} 0 \\ \alpha \\ 0 \end{pmatrix}, t \right) \tag{81}$$

This helped us consider signal dynamics on durations that largely overpass the observation range of 10 to 14 hr, by applying successive rotations of the meristems to simulate the passing of time. Notably, an aligned SAM observed at $t = 0h$ rotated of 137.5° clockwise has been shown to be the best next frame to the same aligned SAM observed at $T = 10h$ (**Figure 1G**, **Figure 1—figure supplement 2H**). More generally, any primordium of stage $\mathbb{P}_p$ visible at time $t$ can be used to infer information at time $t + pT$. This means that we were able to reconstruct trajectories of signals at a given 2D projected position $(r, \theta)$ over a duration of up to nine plastochrons (~ 100 hours) from observations spanning only one, but with nine visible primordia $\mathbb{P}_{-3}$ to $\mathbb{P}_5$. This was achieved only by interpolating rotated aligned information (**Video 1**).

Unfortunately, this global reconstruction heuristic could work only while we were looking at the same location in space, where the spatio-temporal property holds. To some extent, it can be generalized to robust primordia landmarks, such as auxin maxima, that we assume to be unique while moving in the course of primordium development. If they can be identified, and associated with a primordium of stage $\mathbb{P}_p$, then they can be positioned on the same developmental axis at a time $t + pT$ to reconstruct a developmental history at the level of this time-tracked landmark.

However, the moment we are interested in cellular processes, such as the dynamics of transcriptional response to auxin for instance, the reconstructed long-term trajectories cannot be used to draw relevant conclusions, as they reflect the dynamics either at fixed coordinates or at non-cell-specific landmark points. It is therefore necessary to find a way to access temporal cell-level information. Individual cells can be tracked in time-lapse sequences, either manually or automatically, which could be use to obtain signal trajectories over 10 to 14 hr (**Figure 2G**). But to achieve the mentioned 100 hr reconstruction, spatio-temporal periodicity has to be used at some point.

# Extrapolated tissue area tracking

We assume the cells in the central and peripheral zones (CZ and PZ respectively) of the SAM to have essentially an outward radial motion that accelerates as cells exit the central zone. This has been confirmed by the cellular motion vectors estimated from vector fields of image deformation (Appendix 2) in which the azimuthal component is in average close to 0, with a limited amplitude compared to the radial component (**Appendix 5—figure 1A**). We note $\mathbf{v}_n$ the speed of the nucleus labelled $n$ in the 2D polar SAM reference frame $\mathcal{R}^*$:

$$\forall n \in \mathcal{L}_1, \mathbf{v}_n = v_{r,n}\mathbf{r}(P_n) + v_{\theta,n}\theta(P_n) \tag{82}$$

where $\mathbf{r}(P_n)$ and $\theta(P_n)$ are the local normal and tangential unitary vectors at $P_n$. In the following, we will then assume a pure radial motion of cells in the L1 of the SAM, that is that:

$$\forall n \in \mathcal{L}_1, v_{\theta,n} \sim 0 \tag{83}$$

We compute the local radial cellular speed as a 2D continuous map on the L1 (**Appendix 5—figure 1B-C**), using the same parameters as for the other cellular signals: $\forall (r,\theta) \in \mathbb{R}^2$,

$$\widehat{v_r}(r,\theta) = 1 \sum_{n \in \mathcal{L}_1} \eta(\|(r,\theta) - P_n\|) \sum_{n \in \mathcal{L}_1} \eta(\|(r,\theta) - P_n\|) v_{r,n}. \tag{84}$$

Defined this way, local radial cellular speed is a tissue-level information, that is not attached to a cell but to a spatial position $(r,\theta)$. Therefore it has a spatio-temporal periodicity property and we can write:

$$\widehat{v_r}(r,\theta,t+T) = \widehat{v_r}(r,\theta + \alpha, t) \tag{85}$$

We use this spatio-temporal periodicity property of local cellular speed to extrapolate cell motion over time from a series of acquisitions of SAMs at discrete times $\{t_0 < \ldots < t_n < T\}$. Let us consider a cell with an initial position $P(t_0) = (r(t_0), \theta_0)$, setting $r(t_0) = r_0$. Using acquisitions at $t_0$, it is possible to estimate $\widehat{v_r}(r_0, \theta_0, t_0)$, which we use to estimate $P(t_1) = (r(t_1), \theta_0)$ assuming a linear motion between $t_0$ and $t_1$:

$$r(t_1) = r_0 + (t_1 - t_0) \cdot \widehat{v_r}(r_0, \theta_0, t_0) \tag{86}$$

More generally, with observations at $t_{i-1}$, we derive $P(t_i)$ with:

$$r(t_i) = r(t_{i-1}) + (t_i - t_{i-1}) \cdot \widehat{v_r}(r(t_{i-1}), \theta_0, t_{i-1}) \tag{87}$$

We perform this progression until we compute the last position $P(t_n)$ for which motion can not be estimated (as there is no next image it compute image deformation) However, we can still extrapolate the lastly computed motion to reach one plastochron, and estimate $P(t_0 + T)$ with:

$$r(t_0 + T) = r(t_{n-1}) + (T - (t_{n-1} - t_0)) \cdot \widehat{v_r}(r(t_{n-1}), \theta_0, t_{n-1}) \tag{88}$$

To proceed further, we would like to progress in time and estimate the cell position at $t_1 + T$. This is where we use the spatio-temporal periodicity property to derive that:

$$\begin{aligned} r(t_1 + T) &= r(t_0 + T) + (t_1 - t_0)) \cdot \widehat{v_r}(r(t_0 + T), \theta_0, t_0 + T) \\ &= r(t_0 + T) + (t_1 - t_0)) \cdot \widehat{v_r}(r(t_0 + T), \theta_0 + \alpha, t_0) \end{aligned} \tag{89}$$

We are therefore able to estimate this next position from observations at $t_0$ simply by rotating the radial speed map (**Appendix 5—figure 1D**). Then iteratively it is possible to go further in time to $t_n + T$, extrapolate motion to $t_0 + 2T$, apply again spatio-temporal periodicity to reach $t_1 + 2T$, and so on. Finally we obtain the two following general equations:

$$\forall i \in [\![1,n]\!], r(t_i + pT) = r(t_{i-1} + pT) + (t_i - t_{i-1})) \cdot \widehat{v_r}(r(t_{i-1} + pT), \theta_0 + p\alpha, t_{i-1}) \tag{90}$$

$$r(t_0 + (p+1)T) = r(t_{n-1} + pT) + (T - (t_{n-1} - t_0)) \cdot \widehat{v_r}(r(t_{n-1} + pT), \theta_0 + p\alpha, t_{n-1}) \tag{91}$$

These are valid for positive integer values of $p$ until reaching the maximal extent $\mathbb{P}$ in the considered data (i.e. when the map used to estimate local radial speed becomes ill-defined), which determines a value $p_{max}$. This defines a radial trajectory in the 2D space that reflects the local motion of cells along several plastochrons (**Appendix 5—figure 1E**). Note that a rigorously identical approach can be used to go backwards in time with negative values of $p$ until $t_0 + p_{min}T$, using motion vectors computed using inverse image deformation (Appendix 2).

In the end, from a single initial position $(r_0, \theta_0)$ we obtain a discrete radial trajectory:

$$\{(r(t_i + pT), \theta_0) \mid i \in [\![0,n]\!], p \in [\![p_{min}, p_{max}]\!]\} \tag{92}$$

To monitor a cellular process over long time courses, the objective would be to estimate the value of the signal $S$ along this spatio-temporal trajectory, that is:

$$\left\{ \widehat{S}(r(t_i + pT), \theta_0, t_i + pT) \mid i \in [\![0, n]\!], p \in [\![p_{\min}, p_{\max}]\!] \right\} \tag{93}$$

Using the spatio-temporal periodicity property of $S$, this translates into:

$$\left\{ \widehat{S}(r(t_i + pT), \theta_0 + p\alpha, t_i) \mid i \in [\![0, n]\!], p \in [\![p_{\min}, p_{\max}]\!] \right\} \tag{94}$$

In other terms, we have defined a series of spatial locations in a time-series of acquisitions such that the sequence of signal values at these locations on the same meristem estimates the cell-level trajectory of the considered signal. If we represent them on a time-series of meristems, we define tissue areas that can be tracked, first in time then in space, to reconstruct the average behavior of a group of cells over time (*Appendix 5—figure 1F-H*). This is the approach we use to reconstruct cell-level auxin trajectories (*Figure 2H*) and to study the relationship between auxin input and transcriptional response in a consistent group of cells (*Figure 4D*, *Figure 4F–H*).

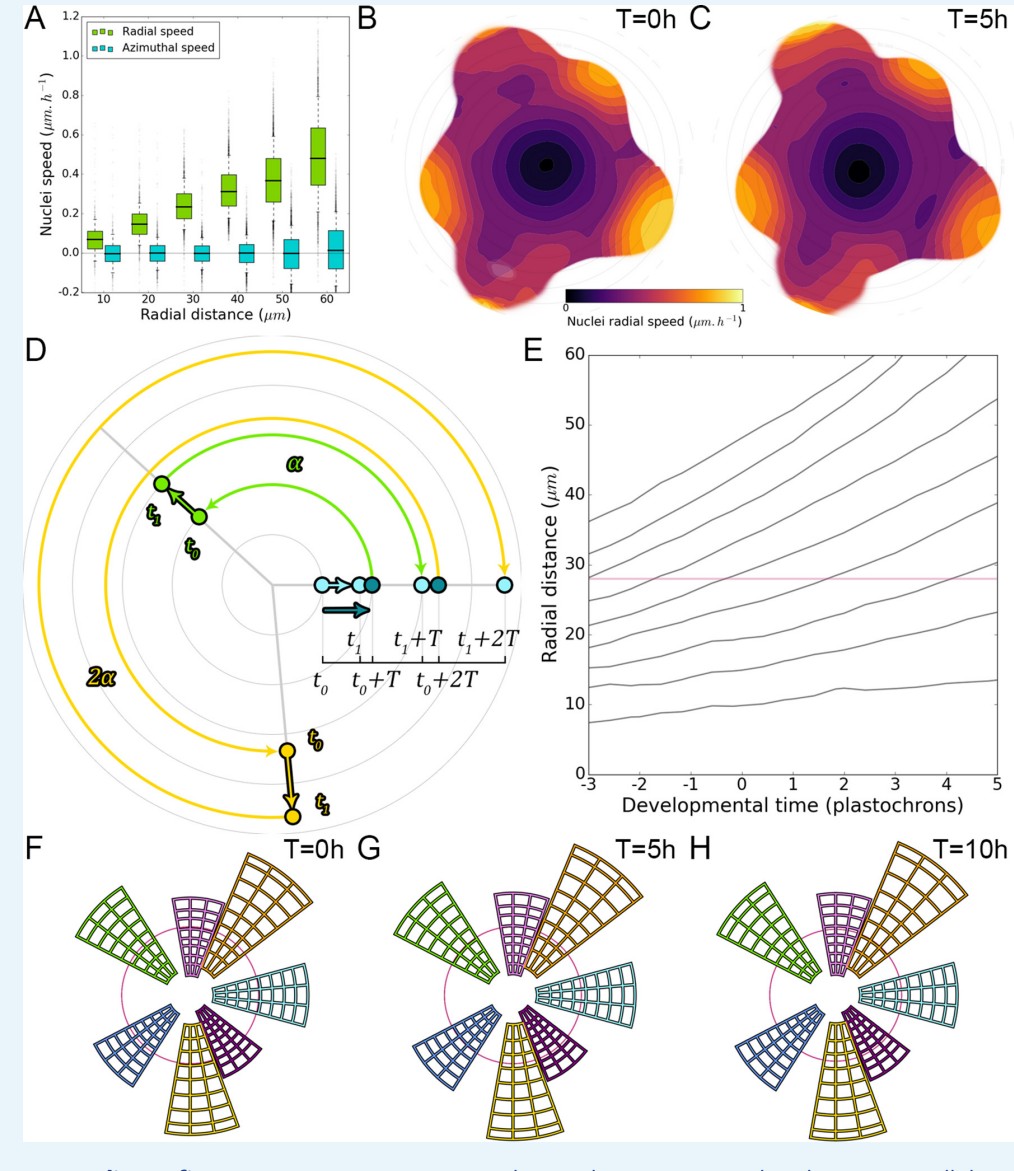

**Appendix 5—figure 1.** Using spatio-temporal periodicity to extrapolate long-term cellular

motions. (**A**). Cellular motion on the L1 is essentially a radial motion towards the periphery of the SAM that accelerates as cells exit the CZ. (**B–C**). Average 2D maps of L1 local radial cellular speed, computed from image deformations between observations at t = 0 hr and t = 5 hr (**B**) and t = 5 hr and t = 10 hr (**C**) ($N = 21$ SAMs). (**D**). Spatio-temporal periodicity allows estimating cellular motion on several plastochrons by successive rotations: motion at $t_0$ can be used to estimate position at $t_1$, which can be extrapolated to $t_0 + T$. By periodicity, radial motion a $t_0 + T$ is equal to radial motion at $t_0$ rotated by one divergence angle $\alpha$, which gives the position at $t_1 + T$, and so on. (**E**). The iteration of this process in time allows the reconstruction of long-term radial cellular trajectories that correspond to the average motion of cells over the population over nearly 100 hr. (**F–H**). These trajectories are used to define spatial domains that reflect cellular motion over time and allow the study of cell-level processes over long durations.

