## [Decision Letter]

**Acceptance summary:**

Your impressive quantitative time lapse approach provides novel insight into the dynamics of auxin distribution in shoot apical meristems. This is an important study for the field of auxin biology and certainly defines a new standard.

**Decision letter after peer review:**

Thank you for submitting your work entitled "Temporal integration of auxin information for the regulation of patterning" for further consideration by *eLife*. Your article has been evaluated by Christian Hardtke (Senior Editor) and Reviewing Editor Jürgen Kleine-Vehn.

The reviewers were highly positive about your work. The reviewers do not request additional experiments, but ask you to revise your manuscript for style and clarity. *eLife* is certainly a forum that allows you to highlight the novelty of your findings, but at the same time also to critically discuss your data and experimental shortcomings. We are very much looking forward to receiving a revised version of your manuscript, which integrates the constructive comments of the reviewers (see below).

Reviewer #1:

Auxin is a central determinant of pattern formation in plants. Here Galvan-Ampudia et al. use a ratiometric DII/mDII reporter to address how spatio-temporal pattern of auxin emerges. It is not very surprising that spatial auxin distribution (as visualized by DII) in the shoot apical meristem follows a precise reiterative pattern, since this correlates with (precise) organ formation. However, the authors here visualize the emergence of this pattern with a high temporal and cellular resolution. However, the manuscript is at places hard to follow and could benefit from better explanations of the methodology and thorough discussion of the related findings. A central claim of this study is that auxin maxima travel through the meristem, which cannot be explained by growth, but depends on intercellular transport. This again confirms previous assumptions, but it would be nice to better describe the underlying data. I am for example wondering how the confinement or enlargement of an auxin maxima would affect the calculations/predicted speed of auxin waves. In the given example (Figure 2A-C), the auxin maxima/DII minima seems to be in time confined to less cells. Do the authors define such a confinement as traveling?

Especially in the central zone, DII dynamics do not correlate with DR5 dynamics confirming the anticipated auxin signaling repression in this zone. The authors experimentally address its nature by exogenous application of auxin. I am a bit worried about the very strong conclusions related to these experiments. Considering that the tissue layout and cell identities differ substantially, the uptake of IAA can be highly diverse in the different regions of the SAM. Hence, I doubt that these results are easy to interpret. It is probably also challenging to obtain uptake information. Moreover, additional components (e.g. metabolism/intracellular transport) may additionally cloud the conclusions on auxin signaling mechanisms. All these aspects also affect the interpretation of the pid mutant, because it has a more regular tissue layout and unified tissue identity. The data on ETTIN is poorly discussed and I feel even misplaced. ETTIN is a non-canonical component of auxin signaling and hence may not be the best choice to discuss general aspects of auxin signaling.

Reviewer #2:

Manuscript by Galvan-Ampudia is an interesting story that demonstrates how auxin concentration is integrated with time and space to deliver new organs in the aerial section of the plant. I was particularly impressed with the combination of imaging techniques and time lapses that were used to gather quantitative data on auxin dynamics. In short, I believe it is an important study that will receive further attention. However, I would like to raise some questions that authors should address before the publication:

1) Results paragraph two:

It is not clear to me what authors mean by minimal information loss when they align SAM images (percentage of error?). This should be clarified and explained better for a general reader.

2) A dynamic range of DII-VENUS is not ideal. In fact, this reporter can only discern between relatively low auxin concentrations – we can actually see some signal. Therefore, it is difficult to distinguish between intermediate or high auxin concentration. Traveling auxin waves would be expected to carry rather medium or high auxin concentrations. Could the authors elaborate more on this matter?

3) Results paragraph four: Has to be clarified – this sentence is difficult to understand

4) I do not understand this sentence "reconstruction of several initiation events from only one?”

5) Auxin is known to travel with the speed of ~10mm/h (Kramer et al., 2011), whereas authors observe waves moving with speeds <1µM/h. Would authors come up with a plausible explanation for this severely reduced auxin waves velocities (~ 4 orders of magnitude difference).

6) Another interesting aspect is that transcriptional responses to auxin are delayed by roughly 12h. Now, is there anything known about the mechanism that could cause such an extensive delay?

7) Auxin treatments presented in the manuscript seem to involve unusually high if not extreme auxin concentration (mM range). Are those concentrations physiologically relevant? Could those effects observed be merely an artifact that is caused by overdosing of auxin?

8) Subsection “Spatio-temporal control of auxin efflux and biosynthesis” paragraph two: Interesting concept presented there. Would then PINs align towards or alternatively from an auxin source (production sites) based on new data presented in the manuscript.

9) Subsection “The role of time in transcriptional responses to auxin”, final sentence in first paragraph: very unclear sentence that needs revising.

Reviewer #3:

This manuscript by Galvan-Ampudia et al. reports an intensive study that investigated spatio-temporal patterning of auxin activity in the shoot meristem. The data reveal an intriguing dynamic over space and time that, to my knowledge, is captured here for the first time. There are various interesting observations here, for instance the observed partial temporal uncoupling between auxin levels and auxin response, which provides important information for judging auxin response in different contexts. Overall, the effort the authors made to monitor auxin dynamics quantitatively in a growing meristem are impressive. There are some limits one could discuss, for example the methodological approach to determine PIN polarity, but in any case, in my opinion the methods are state-of-the-art and thus as good as it gets for now. The manuscript is well written and documented, although I must add that I cannot judge the modeling part competently. It also shows that this paper has already undergone two rounds of revision elsewhere, it is very much honed and as far as I am concerned ready to publish.

---

## [Author Response]

Reviewer #1:Auxin is a central determinant of pattern formation in plants. Here Galvan-Ampudia et al. use a ratiometric DII/mDII reporter to address how spatio-temporal pattern of auxin emerges. It is not very surprising that spatial auxin distribution (as visualized by DII) in the shoot apical meristem follows a precise reiterative pattern, since this correlates with (precise) organ formation. However, the authors here visualize the emergence of this pattern with a high temporal and cellular resolution. However, the manuscript is at places hard to follow and could benefit from better explanations of the methodology and thorough discussion of the related findings.

We apologize for the lack of clarity in some parts of the manuscript. This is largely due to the original choice of a short format. We have followed the comment of the reviewer to address this (see below).

A central claim of this study is that auxin maxima travel through the meristem, which cannot be explained by growth, but depends on intercellular transport. This again confirms previous assumptions, but it would be nice to better describe the underlying data. I am for example wondering how the confinement or enlargement of an auxin maxima would affect the calculations/predicted speed of auxin waves. In the given example (Figure 2A-C), the auxin maxima/DII minima seems to be in time confined to less cells. Do the authors define such a confinement as traveling?

We have measured the motion of the centers of auxin local maxima, independently of the spatial extent of these zones. We have not analyzed how the extent of auxin maxima varies over time, as auxin distribution show dynamic and complex changes over time. In the case of Figure 2A-C, we highlighted cells (green circle) that were around the center of the accumulation zone at time 0 (and not the extent of the auxin maximum zone) and end up depleted in auxin at time 10 (white circles). It can also be seen that the cells on the distal edge of the accumulation zone at time 0 (red circles) at the same time undergo a progressive increase of their auxin level. In other words they follow opposite trends in auxin. Our data thus indicate that these changes are not simply due to a confinement of high auxin concentrations to fewer cells but rather to a spatial shift in the auxin distribution pattern relatively to the cellular canvas of the tissue. We have clarified this in the main text, modified Figure 2A-C and amended the legend of Figure 2.

But it is true also that the maximum is confined to fewer cells with the progressive establishment of the auxin minimum at the boundary of the CZ. We have thus added this in the main text.

Especially in the central zone, DII dynamics do not correlate with DR5 dynamics confirming the anticipated auxin signaling repression in this zone. The authors experimentally address its nature by exogenous application of auxin. I am a bit worried about the very strong conclusions related to these experiments. Considering that the tissue layout and cell identities differ substantially, the uptake of IAA can be highly diverse in the different regions of the SAM. Hence, I doubt that these results are easy to interpret. It is probably also challenging to obtain uptake information. Moreover, additional components (e.g. metabolism/intracellular transport) may additionally cloud the conclusions on auxin signaling mechanisms. All these aspects also affect the interpretation of the pid mutant, because it has a more regular tissue layout and unified tissue identity.

We agree with the reviewer that differential uptake could be an important issue. Although it is effectively challenging to obtain a precise information, having qDII in the treated meristems allows is to show that even after 30’ DII-VENUS is degraded throughout the meristem. We were showing these data only in the form of plots in Figure 5—figure supplement 1 and have now included images (Figure 5—figure supplement 1A-H) in addition to the quantifications (Figure 5—figure supplement 1I). These data suggest that the uptake could be homogeneous. For *pid* mutants, the fact that all cells react with the same dynamics to the treatments suggests also an uptake throughout the peripheral zone. And this is also consistent with previous publications that we have cited (10,12,20) where activation of DR5 throughout the PZ can be observed in different genetic backgrounds.

Taking this into consideration, we have modified the text to better highlight how our results support our conclusions but at the same time to point to the fact that differential auxin uptake could still bias our results.

Concerning additional components such as auxin metabolism and intracellular transport, this experiment does not demonstrate in any way that the delay that we observe in the activation of DR5 is due to mechanisms involving only auxin signaling effectors or chromatin (as supported in part by our results with *ett* and TSA). We completely agree that auxin metabolism and intracellular transport could be implicated in creating the delay and in integrating the auxin concentration. What is in between auxin and DR5 remains a black box in this experiment, however, it allowed us to identify the emergent properties of the different mechanisms acting in this black box. Taking into account the comment of the reviewer, we have now better discussed this and mention auxin metabolism and intracellular transport as possible mechanisms acting in the temporal integration of auxin concentration.

The data on ETTIN is poorly discussed and I feel even misplaced. ETTIN is a non-canonical component of auxin signaling and hence may not be the best choice to discuss general aspects of auxin signaling.

While it is true that ETTIN is a non-canonical ARF, the genetic data recently published by the group of Doris Wagner (Chung et al., 2019) demonstrate clearly that ETTIN is acting redundantly with ARF4 and MONOPTEROS in promoting organogenesis. In addition, our results showing a reduced DR5 expression in the *ett* background support this conclusion. Following the advice of the reviewer, we have now highlighted this in the text.

It is true that confirming the ETTIN results with other genetic backgrounds with perturbation of auxin signaling will be necessary in the future (we had tried this without success due to difficulties with genetic material). We have thus now indicated the need for such an analysis in the discussion at the end of the paragraph on the mechanisms that could act in time integration of the auxin signal (Discussion paragraph two).

Reviewer #2:Manuscript by Galvan-Ampudia is an interesting story that demonstrates how auxin concentration is integrated with time and space to deliver new organs in the aerial section of the plant. I was particularly impressed with the combination of imaging techniques and time lapses that were used to gather quantitative data on auxin dynamics. In short, I believe it is an important study that will receive further attention. However, I would like to raise some questions that authors should address before the publication:1) Results paragraph two:It is not clear to me what authors mean by minimal information loss when they align SAM images (percentage of error?). This should be clarified and explained better for a general reader.

We have rephrased this in the following way:

“All images could be superimposed preserving the spatial distribution of auxin maxima and minima “

2) A dynamic range of DII-VENUS is not ideal. In fact, this reporter can only discern between relatively low auxin concentrations – we can actually see some signal. Therefore, it is difficult to distinguish between intermediate or high auxin concentration. Traveling auxin waves would be expected to carry rather medium or high auxin concentrations. Could the authors elaborate more on this matter?

We are not entirely sure that we follow the reasoning of the reviewer here. Our quantification demonstrates that we are able to detect blue and yellow signal in most cells of the PZ in the SAM. In the PZ, we only get close to saturation at P0 i.e. the absolute maximum of auxin in the meristem (absolute minimum of VENUS fluorescence). 1-DII-VENUS/TagBFP is at 1 when it saturates and one can see for example Figure 4G that most of the values are below 1. qDII has thus a dynamic range that detects the concentration present in the PZ and thus to do the analysis that we report in this manuscript. To make that sure to readers we have amended the text and state clearly that the dynamic range of qDII provide information without saturation in the vast majority of cells in the PZ.

3) Results paragraph four: Has to be clarified – this sentence is difficult to understand

We have clarified the sentence in the following way: “We then considered the temporal changes in auxin distribution by using time-lapse images over one plastochron, which corresponds to the period of this rhythmic system”

4) I do not understand this sentence "reconstruction of several initiation events from only one?”

We apologize but we could not find this phrase in the main text of the manuscript and were thus not able to provide an answer to this concern.

5) Auxin is known to travel with the speed of ~10mm/h (Kramer et al., 2011), whereas authors observe waves moving with speeds <1µM/h. Would authors come up with a plausible explanation for this severely reduced auxin waves velocities (~ 4 orders of magnitude difference).

The speed compiled and reported in Kramer et al., 2011 come from experiments done on tissue segments (root, stem) and very likely involve the vascular tissue. So they give an idea of the speed of transport on long distances going at least in part through specialized tissues, which could be part of the explanation.

More importantly, the speed that we estimate is not the speed at which auxin itself travels between cells but a motion of the distribution pattern. The spatial pattern of the auxin distribution can be seen as the current “steady state” of a system where, among other processes, cells are constantly exchanging auxin through transport at a much higher rate. The dynamic reconfiguration of this system (notably through the changes in orientations of the PIN1 network) leads to changes in its steady state, which takes place over much longer time frames. The motion of auxin accumulation zones corresponds to the spatial dynamics of this slowly varying steady state, hence the smaller speeds.

For these different reasons, we simply think that these are two type of data that cannot be compared. It seemed thus difficult to include a specific text on this point in the manuscript. We have simply stated on several occasion that velocity/movement/wave are only “apparent”.

6) Another interesting aspect is that transcriptional responses to auxin are delayed by roughly 12h. Now, is there anything known about the mechanism that could cause such an extensive delay?

It is the first time that such a delay is demonstrated in response to auxin or to our knowledge to any other developmental signals in plant. It is beyond the scope of the manuscript to elucidate the mechanism beyond this delay and at this stage we can only discuss/speculate about the mechanisms at play. We had and are still discussing a possible mechanism that could be based on the regulation of the acetylation state of auxin target genes. Regulation of chromatin state is a mechanism involved in time integration in animal and plants (Nahmad et al., 2011; we slightly amended the text to make that clearer). This is also pertinent in light of a recent publication by some of the authors (Ma et al., 2019) that we discuss and that show that WUS represses auxin signaling genes and auxin target genes in the CZ through histone deacetylation. The delay could then be due to the time taken to acetylate the auxin target loci. To address a comment of reviewer #1, we are also now suggesting that mechanisms regulating intracellular auxin concentrations could also regulate the delay. This can be found in the Discussion paragraph two.

7) Auxin treatments presented in the manuscript seem to involve unusually high if not extreme auxin concentration (mM range). Are those concentrations physiologically relevant? Could those effects observed be merely an artifact that is caused by overdosing of auxin?

We have used concentrations of exogenous auxin that have already been shown to be physiologically relevant for the SAM. This concentration would be unusually high for roots but they are not for the SAM where concentrations up 30 mM have been used to induce organogenesis or gene expression at the SAM (Reinhardt et al., 2000; Heisler et al., 2005, also Sassi et al. Curr Biol 2014). So the effects that we see and for which we use a concentration as low as 0.1 mM are very unlikely to be caused by overdosing of auxin.

We have thus simply amended the text in the following way: “To test these scenarios, we treated SAMs with auxin for different periods using physiologically relevant concentrations (11,12)”

8) Subsection “Spatio-temporal control of auxin efflux and biosynthesis” paragraph two: Interesting concept presented there. Would then PINs align towards or alternatively from an auxin source (production sites) based on new data presented in the manuscript.

Effectively our data suggest a situation where PIN1 proteins allow for pumping auxin away from the auxin production sites. We have added extra text stating this.

We would like to point at the same time that PIN1 polarities allows pumping auxin toward auxin maxim at the center and organ initia. So clearly further work is needed to understand what drives the organization of the polarities at the SAM scale.

9) Subsection “The role of time in transcriptional responses to auxin”, final sentence in first paragraph: very unclear sentence that needs revising.

We apologize. This was simply a typo: “at” changed for “is” to clarify the sentence.

Reviewer #3:This manuscript by Galvan-Ampudia et al. reports an intensive study that investigated spatio-temporal patterning of auxin activity in the shoot meristem. The data reveal an intriguing dynamic over space and time that, to my knowledge, is captured here for the first time. There are various interesting observations here, for instance the observed partial temporal uncoupling between auxin levels and auxin response, which provides important information for judging auxin response in different contexts. Overall, the effort the authors made to monitor auxin dynamics quantitatively in a growing meristem are impressive. There are some limits one could discuss, for example the methodological approach to determine PIN polarity, but in any case, in my opinion the methods are state-of-the-art and thus as good as it gets for now. The manuscript is well written and documented, although I must add that I cannot judge the modeling part competently. It also shows that this paper has already undergone two rounds of revision elsewhere, it is very much honed and as far as I am concerned ready to publish.

We thank the reviewer for the very positive evaluation of the manuscript. We have taken into account the comment on PIN1 polarity. Concerning our approach and to make clear that there some limits to it, we now state the following:

“This quantitative evaluation (Figure 3D-G, Figure 3—figure supplement 1 and Appendix 4) validates the robustness of our method, showing that, in spite of a coarse image resolution, a vast majority of cellular polarity directions are consistent with super resolution imaging techniques. Our approach is therefore particularly suitable for monitoring global trends at the scale of a tissue”.

[Editors' note: we include below the reviews that the authors received from another journal, along with the authors’ responses.]

First round of Review

Reviewer 1:In plants, the hormone Auxin serves as a major regulator of development, growth and physiological response. Here, looking at organ primordium formation in the Arabidopsis shoot meristem (SAM), the authors address the question of whether the helical arrangement of organs reflects positional information provided by variations of auxin levels in time in addition to (as was previously described) in space. This is an exciting and provocative story for several reasons: (1) it challenges ideas about the primary role of the PINs in auxin-mediated development; (2) it describes a new time-dependent patterning phenomenon and (3) suggests a mechanism for how “memory” of prior auxin can lead to a delayed response. The imaging data are of high quality and these and the modeling data are, in general, presented in a logical and aesthetically pleasing way.

We appreciate this very positive comment on our work.

My comments related to the 3 major points of this story:1) This is clearly challenging PIN-based SAM models and I expect the experts in those fields/proteins can evaluate the subtle differences in behaviors better than I, and also say whether this paper fairly characterizes previous data and interpretations. I will restrict myself to evaluating the data presented here and the authors conclusions. As an outsider, whether they disprove specific PIN hypotheses isn’t all that important to me-if that’s all this paper did, I wouldn’t find that alone to be sufficiently compelling, and it they failed to disprove an old hypothesis I could see ways that this would be suitable for publication.

We thank the reviewer for his appreciation of this part of our work. We agree that the point of our manuscript is not to challenge the PIN1-based SAM models. The reason for exploring the distribution of PIN1 polarity is rather linked to the unexpected dynamics of auxin distribution we measured and that cannot be easily explained by the existing models. We would like to point out here that we have now further validated our approach for quantifying PIN1 polarities using super resolution microscopy. A comparison between the polarities computed from a 3D confocal image and the one determined using single plane super resolution image is now shown on Figure 3B-G. We have also added a new Figure 3—figure supplement 1 with further images and a quantification of the differences between polarities determined with the two approaches. The quantification shows that more than 80% of PIN1 cellular polarities match almost perfectly (deviation less than 30°) when comparing the two approaches (Figure 3—figure supplement 1 and corresponding text).

Note also that we now provide genetic data that, in addition to published work, suggest that auxin biosynthesis from YUC1 and YUC4 in flowers is important for phyllotaxis. Taken all together, our findings provide a tissue-scale vision of the organization of PIN1 polarities and auxin biosynthesis that is compatible with the auxin distribution we observe.

2) This is the strongest portion of the manuscript. It is based on the generation of new reporters (a ratiometric DII-FP) and imaging/computational innovations that showed that auxin distribution (as measured by this reporter) was much more stereotyped than one might have naively expected given that plant cell divisions in the SAM are not predetermined (i.e., the patterning is not via a *C. elegans-*like invariant lineage). Still, following 21 SAMs over a 10 hr interval, it was possible to see that the same reporter behavior almost to cell level resolution. This consistency allowed the authors to explore a time dimension of auxin-mediated patterning, and to test previous models of how auxin maxima are associated with cellular growth. In general, I found the experimental data reported here (Figures 1-4) convincing.

We thank the reviewer for this positive evaluation and agree that this is a crucial part of our analysis.

3) It is convincing that DR5 and DII-Venus have complex, non-linear relationships. It is a really exciting possibility that there is something happening in the SAM that enforces a delay in response. However, non-linear relationships could also be due to less direct or interesting mechanisms than the authors suggest, and of course “inputs” and “responses” are based on imperfect sensors of auxin levels and auxin response. Still, a stronger argument for auxin priming responses in cells could be obtained by modifying the auxin treatments. Of the two possibilities presented, the first one is never really tested. For the second, it’s not really clear that its time and not overall amount of auxin (amount integrated over time). I compare this to experiments with light responses where people are careful to include controls the test the total # of photons received to ensure that a particular response is due to duration of light exposure and not just total amount. Could you take advantage of the regularity of the SAM to compare short pulses of high auxin vs. longer pulses of lower amounts to test the differential effects on different cell populations?

We agree with the reviewer that analyzing the effect of auxin treatments with different concentrations and durations on the induction of transcription was an excellent way to consolidate this essential part of our manuscript. We have now included the results of new treatments with different auxin concentrations and different durations of treatments on Figure 5I. This figure now shows the effect on DR5 expression of treatments with 0.2mM, 1mM and 5 mM auxin for different durations. The effect of each treatment is shown throughout the PZ (x axis with indication of the position of primordia) as the log of the difference between DR5 expression after and before the treatment. This allows us now to show that both the time of exposure and the concentration of auxin control the activation of transcription monitored by DR5, thus supporting the conclusion that cells integrate the concentration of auxin over time and not only time.

Regarding the “first possibility” mentioned by the reviewer i.e. the possibility that the delay in activating transcription is due to intrinsic auxin-independent differences in cells, we would like to mention that we had already stated in the text that the exogenous auxin treatments on wild-type meristems addressed at least in part this question. Indeed in this scenario, the capacity to respond is linked to an intrinsic auxin-independent regulation of the competence that would create an asymmetry between different positions at the periphery of the central zone. It is very unlikely that even a 5h treatment would allow to induce transcription throughout the peripheral zone. One would expect that asymmetries in the response would still exist. Our data from Figure 5 then suggest that such asymmetries do not exist. Also, this conclusion is supported by published data (Reinhardt et al., 2003) demonstrating that in pin-shaped mutants all cells at the periphery of the meristem can respond to auxin by producing outgrowth. This suggests again that cells exiting the stem cell niche are equally competent to respond to auxin. We have now complemented these data by analyzing transcription activation by auxin (using DR5) in pinshaped inflorescence of the pinoid mutant. Organ initiation is inhibited in pinoid meristems and our results indicate that there is no pre-established auxin distribution. We also show that all cells at the periphery of the meristem can respond similarly to an exogenous auxin treatment but again activation of transcription by auxin is dependent on the duration of the treatment. We have included these new results in Figure 5J-M and Figure 5—figure supplement 2A-C.

We explain the effect of auxin treatments in wild-type and pinoid meristems in the text.

The way this section is written gives the impression that the DR5 vs. DII expression in the SAM is a unique feature of this developmental situation, Please cite the studies that show this is not the case elsewhere in the plant.

We have never claimed that a dependence on the time of exposure to auxin for DR5 activation was unique to the SAM and we are unsure why the reviewer had this feeling. Published data (such as the one published by some of the authors in Brunoud et al., 2012 and Band et al., PNAS 2012) show that DR5 might respond with no delay to endogenous or exogenous modifications of auxin concentrations. However, demonstrating that DR5 responds to auxin with no delay in the root and in other parts of the plant would require an extensive quantitative analysis that goes beyond the scope of the manuscript. We have thus highlighted this in the Discussion in the following way:

“Whether integration of auxin temporal information exists in other tissues remains to be established.”

The causal relationships between auxin, HDACs and DR5 response needs to be tested more clearly. How do you know that chromatin modifications aren’t auxin-independent events happening as cells leave the CZ? I would be more convinced by a genetic experiment that should it was an ARF-mediated effect on chromatin needed for response, not just a very broad drug treatment that potentially affects most of the chromatin.

There are few auxin signaling proteins for which a clear function has been established in the SAM. The one with the clearest implication is ARF5/MP but the mutant was shown to be insensitive to auxin in the meristem (Reinhardt et al., 2003) and thus cannot be used here. ARF3/ETTIN and ARF4 are the two other ARFs that show high expression during organ initiation (Vernoux et al., 2011). A key role for ARF3/ETTIN in organ initiation at the SAM has recently been demonstrated by a publication from the laboratory of Doris Wagner. Following the advice of this reviewer (and of another one), we have now included data demonstrating the ARF3/ETTIN gene is necessary for the integration of the auxin signal. These new data are included in Figure 5 (Figure 5N-Q) and the new Figure 5—figure supplement 2 (Figure 5—figure supplement 2D,E, and described in the text in the following way:

“The Auxin Response Factor (ARF) ETTIN (ETT/ARF3) plays an important role in promoting organogenesis in the SAM (Wu et al., 2015; Chung et al., 2019). We found that in a loss-of function ett3 mutant the expression of DR5 was restricted to only 2-3 cells at sites of organogenesis, and that a 300’ 1mM auxin treatment did not induce DR5 in the SAM (Figure 5N-Q and Figure 5—figure supplement 2D-E).”

Concerning the link to histone acetylation, we believe that it is beyond the scope of the revision of this manuscript to explore in depth whether histone acetylation levels provide the mechanism that explains time integration of auxin information. While the functional link between auxin signaling and histone acetylation is well established, understanding it further is a full research question that certainly requires years of research.

Bearing these considerations in mind, we now show the results of the pharmacological inhibition of HDACs on DR5 expression in a supplementary figure (Figure 5—figure supplement 1) and discuss them in the following after the description of the result with ARF3:

“Auxin signaling and ARF3 have been shown to act by modifying acetylation of histones (Wu et al., 2015; Chung et al., 2019 and Long et al., 2006). Pharmacological inhibition of histone deacetylases (HDACs) alone was able to trigger concomitant activation of DR5 at P_0_ and P_-1_ sites in the SAM (Figure 5—figure supplement 1I-K). These results suggest that auxin signal integration depends on a functional ARF-dependent auxin nuclear pathway.”

To further comment on this point, it is indeed likely that chromatin modifications occur both in an auxin-independent and auxin-dependent fashion. Auxin-independent changes when cells exit the stem cell niche are actually suggested by a recent work (Ma et al., 2019) showing that WUSCHEL is required to regulate the histone acetylation status at loci for a large number of auxin signaling genes and auxin targets. It is thus possible that cells exiting the stem cell niche can already experience changes in histone acetylation status due to a lower influence of WUSCHEL. We believe that, given the existing literature, this remains an important discussion point. We have thus kept the corresponding paragraph in the Discussion.

Minor notes:Define azimuth to make more readable to non-expert audience

We added the following explanation in the text: “azimuth (angular distance)”.

Plastrocrone = plastocron?

We apologize for this. It has been corrected throughout the text.

How is this ratiometric sensor different from the other ratiometric DII sensor made by the Weijers group? Have the two been compared in the same tissue?

R2D2 from the Weijers group (Liao et al., 2015) consists in DII-VENUS and a reference fluorescent protein expressed each under their own promoter. For qDII, DII-VENUS and the TagBFP reference protein are expressed from the same promoter, to ensure an equimolar production of DII-VENUS independent of promoter activity variations.

This has been clarified in the text in the following way:

“qDII differs from previously used tools (22) in producing DII-VENUS and a non-degradable TagBFP reference stoichiometrically from a single RPS5A promoter (23, 24)(Figure S1A-H).” However, we believe that a quantitative comparison of the two type of sensors will not add much to the manuscript.

Figure 1—figure supplement 3 Croissant-shaped. Maybe crescent-shaped? Though I rather like the idea of PIN Croissants.

We apologize for this typo. During the revision of the manuscript, we have removed the corresponding term (see below also) and this has thus been corrected in this way.

Some grammar issues throughout, could be resolved through copy-editing

This has been done with the help of a native English-speaking colleague.

“Reconstruction of PIN1 polarities demonstrated that the crescent-shape often thought to indicate polarities in cells does not always correlate with polarities and can thus be sometimes misleading”. This is potentially an important technical point (how do you accurately measure cellular polarity) and it gets a bit lost here—what does “polarities in cells does not always correlate with polarities” mean? Is “tissue” missing? Given the number of people who report “PIN polarities”, I would re-emphasize the importance of accurately measuring this (and the development of such a tool to do so as reported in the supplemental material).

We agree with the reviewer that this phrase was unclear. It should have been “reconstruction of PIN1 polarities demonstrated that the crescent-shape often thought to indicate polarities in cells does not always correlate with actual polarities of the auxin flux”. In the course of the revision of the manuscript, we have removed this phrase to rather put forward the validation of the method we used to quantify PIN polarities from confocal images. As discussed earlier, we have compared the results of our method with PIN1 polarities observed using super-resolution microscopy. We agree that this was an important technical point. In the revised text, we thus highlight the fact that we have used a quantitative approach and explain how we have validated our approach.

Reviewer 2:Unlike animals, plants reiteratively produce organs de novo from stem cells reservoirs at their growing apex, the shoot apical meristem (SAM), and much research has gone into demonstrating that auxin plays a key role in positioning the newly arising organ primordia in an orderly spatial arrangement. However, whether auxin acting as a spatial morphogen can also direct the timing of organ initiation events is unclear. Here the authors address this question by using quantitative imaging to analyze the dynamics of auxin distribution and response during organ initiation in the SAM.This is a novel area of research that the authors tackle with invention and rigor. They for the first time accurately measure the time interval between organ initiation events at the SAM, demonstrate convincingly that auxin maxima move in waves through the SAM to regulate organ patterning, and show that auxin-induced transcription lags significantly behind auxin accumulation. In general the data are robust and sufficiently quantified.

We thank the reviewer for this very positive evaluation of our work.

However, I would also argue that in some cases the authors’ observations are not always clear and do not provide enough support for their conclusions.It is unclear from the vaguely written Abstract what the paper is about – the authors need to be more concrete about what they show auxin does.

We have now rewritten the Abstract focusing on the main discoveries of the work i.e. the fact that auxin carries both spatial and temporal information and the demonstration that temporal integration of auxin information is necessary for activation of transcription.

The introductory text is highly technical and hard for the reader to digest. In particular, the SAM zone terminology and how the PIN1 network functions to regulate auxin distribution, promote auxin accumulation and yet also deplete auxin are briefly summarized without sufficient context.

We have modified the Introduction in order to explain better the SAM terminology. We have also added as a new Figure 1A a schematic representation of the SAM zones in order to help the reader. Concerning the role of PIN1 in regulating auxin distribution, we have better explained the role of PIN1 in controlling the orientation of the auxin fluxes and explained how the spatial organization of PIN1 polarities is thought to control auxin distribution. We believe that we now provide a sufficient context despite the constraint of the format.

In addition, because so much information is presented in such a compressed fashion in the main text, it is quite difficult for the reader to follow the data analysis and interpretation. For instance, entire complex figures, such as Figure 1—figure supplement 3 and Figure 3—figure supplement 1, are described in only a single sentence.

In the revised version of the manuscript, we have extensively edited the text (including the supplementary text and all the figure legends) to make it more accessible.

Concerning Figure 1—figure supplement 3, we effectively only summarize the main findings illustrated in this supplementary figure because of the space limitation and because our focus is primarily on the spatio-temporal dynamics of auxin maxima. However, we have now expanded the legend of Figure 1—figure supplement 3 to provide more information to the reader. We have also added longitudinal sections at primordia positions (Figure 1—figure supplement 3F-L) to illustrate better the dynamics of auxin minima. Concerning Figure 3—figure supplement 1, it has now been largely modified and simplified. To address concerns from several reviewers, we have validated the approach we used to quantify PIN polarities from confocal images. We have compared the results of our method with PIN1 polarities observed using super-resolution microscopy. A comparison between the polarities computed from a 3D confocal image and the one determined using single plane super resolution image is now shown on Figure 3B-G. We have also added a new Figure 3—figure supplement 1 with further images and a quantification of the differences between polarities determined with the two approaches. The quantification shows that more than 80% of PIN1 cellular polarities match almost perfectly (deviation less than 30°) when comparing the two approaches (Figure 3—figure supplement 1 and corresponding text). Here also, the legend provides more information about the data shown in Figure 3—figure supplement 1.

With respect to the experimentation, some of the PIN1 experiments were unclear. Figure 3A-C is described as showing PIN1 concentration over time, but only T0 data are shown.

We thank the reviewer for pointing out this error. Indeed the changes of PIN1 over time are shown in Figure 3—figure supplement 2 and not in Figure 3. We have also corrected the text accordingly notably: “Over the course of a plastochron, only limited changes of the PIN1 polarities are observed (Figure 3—figure supplement 2), …”

Also the vector field mapping experiments were poorly described and thus difficult to interpret.

We apologize for this lack of clarity. We have modified the text to describe better what we do. We do not use the term “vector fields” anymore but only talk about “vector maps” and explain in the text how we go from cell wall polarity vectors to cell polarity vectors by integrating cell wall vectors. We then explain how we generate vector maps in the following way:

“Local averaging of the cellular vectors obtained from confocal images was then used to calculate continuous PIN1 polarity vector maps in order to identify the dominant trends in auxin flux directions in the SAM (Figure 3I, and Appendix 4).”

Note that we have also reorganized the supplementary material to include an Appendix 4 that is dedicated specifically to the analysis of PIN1 polarity and provide extensive details on the methods we have used.

Figure 3 and Figure 3—figure supplement 3 do not persuasively show any YUC6 promoter activity in the SAM proper, much less in the CZ, and no evidence is provided to show that YUC- generated auxin in the primordia is important for auxin flux or organogenesis in the SAM.

We agree with the reviewer that YUC6 promoter activity was difficult to see on both figures. In the revised manuscript, we now show only YUC4 promoter activity on Figure 3 as it has the strongest activity and to limit the size of the figure. We have also modified Figure 3—figure supplement 3 to ease the visualization of the expression of YUC6. To do so, we present for YUC1,4 and 6 both the GFP signal alone (Figure 3—figure supplement 3A-C) and an overlay of the GFP signal and of the PI signal (Figure 3—figure supplement 3D-F). Concerning YUC6, a weak signal can be seen at the center of the meristem.

Moreover, we have added to the manuscript the results of an analysis of yuc1yuc4 double mutants, which showed organ positioning and morphological defects. We present these results on Figure 3L and Figure 3—figure supplement 3O-U. These genetic data, in addition to published work, support the idea that auxin biosynthesis from YUC1 and YUC4 in flowers is important for phyllotaxis. This is explained in the text in the following way: “In addition, yuc1yuc4 loss-offunction mutants show severe defects in SAM organ positioning and size (37, 38) (Figure 3L and Figure 3—figure supplement 3O-U). Taken with the organization of PIN1 polarities, these results suggest that P3-P5 are auxin production centers for the SAM that regulate phyllotaxis.”

I am also unconvinced that the data in Figure 5 demonstrate that auxin addition to the SAM for 30-120 minutes leads to a measureable increase in DR5 expression in the P_0_, P_1_ and P_2_ primordia because of the noise in the experimental output.

Here we made a serious mistake in the text and apologize for this. It should have been P_-2_, P_-1_ and P_0_ and we thank the reviewer for spotting this. In any case, we had chosen on Figure 5 a representation that was as close as possible to the original data but the noise effectively made difficult the visualization of the response for the 30 and 120’ treatments. As the noise in the logtransformed data has a symmetric distribution, we could use a statistical model to calculate the mean of the difference between DR5 expression after and before the treatment, and the 95% confidence interval of the response curves (we explain this in the Material and Method section, Quantification and statistical analysis paragraph). These data are shown on Figure 5 I. An absence of overlap of the confidence interval at a given angular position then indicates that the differences are statistically significant between treatments/control. In the case of the 30’ and 120’ treatments with 1mM auxin, significant induction can be detected at the P_-1_ and P_-2_ sites and even at the predicted angular position of P_-3_. A small induction is visible at P_2_ but there are no significant changes at P_0_ and P_1_. We explain this in the text in the following way: “In the shorter auxin treatments (30’ and 120’), the auxin transcriptional response was mainly enhanced at P_-1_ and P_-2_ and to a lesser extent at the position of the predicted P_-3_ i.e. where cells are already being exposed to auxin (Figure 5I).”

Note also that we have now analyzed the effect of auxin treatments on the induction of transcription using different concentrations and durations of treatment. The results are included in Figure 5I that shows the effect on DR5 expression of treatments with 0.2mM, 1mM and 5 mM auxin for different durations. This allows us now to demonstrate that both the time of exposure and the concentration of auxin control the activation of transcription monitored by DR5, thus supporting the conclusion that cells integrate the concentration of auxin over time and not simply time. In addition, we have analyzed transcription activation by auxin (using DR5) in pin-shaped inflorescence of the pinoid mutant. Organ initiation is inhibited in pinoid meristems and our results indicate that there is no pre-established auxin distribution. We also show that all cells at the periphery of the meristem can respond similarly to an exogenous auxin treatment, which provides further argument that there are no intrinsic differences in the capacity of the cells to respond to auxin. We have included these new results in Figure 5J-M and Figure 5—figure supplement 2A-C. We explain the effect of auxin treatments in wild-type and pinoid meristems in the text.

Similarly, the effect of histone deacetylase inhibition is mild at best and without additional experimental manipulation (such as the use of HDAC mutants) it seems unjustified to draw the broad conclusion that the timing of auxin signaling at a primordium depends on the chromatin acetylation status.

We agree with this reviewer and others that our analysis of the implication of histone acetylation was too preliminary to support this conclusion. We also believe that properly understanding how histone acetylation is involved in time integration would take us largely beyond what is reasonable to include in this manuscript. In the revised manuscript, we only include in the discussion the possibility that histone acetylation could provide a mechanism for time integration based on the literature as we believe that this is an important discussion point.

In link to this comment, note however that we have kept the results of the TSA treatments but rather as a way to support an implication of auxin signaling in time integration of the auxin signal. Indeed other reviewers suggested that genetics could be used to show an involvement of the auxin signaling pathway in time integration of the auxin signal. There are few auxin signaling proteins for which a clear function has been established in the SAM. The one with the clearest implication is ARF5/MP but the mutant was shown to be insensitive to auxin in the meristem (Reinhardt et al., 2003) and thus cannot be used here. ARF3/ETTIN and ARF4 are the two other ARFs that show high expression during organ initiation (Vernoux et al., 2011). A key role for ARF3/ETTIN in organ initiation at the SAM has recently been demonstrated by a publication from the laboratory of Doris Wagner (Chung et al., 2019). We have included in the revision data demonstrating the ARF3/ETTIN gene is necessary for the integration of the auxin signal. These new data are included in Figure 5 (Figure 5N-Q) and a new Figure 5—figure supplement 2 (Figure 5—figure supplement 2D,E) and described in the text in the following way:

“The Auxin Response Factor (ARF) ETTIN (ETT/ARF3) plays an important role in promoting organogenesis in the SAM (Wu et al., 2015; Chung et al., 2019). We found that in a loss-of function ett3 mutant the expression of DR5 was restricted to only 2-3 cells at sites of organogenesis, and that a 300’ 1mM auxin treatment did not induce DR5 in the SAM (Figure 5N-Q and Figure 5—figure supplement 2D-E).”

As the functional link between auxin signaling and histone acetylation is well established, we now show the results of TSA treatments on DR5 expression in a supplementary figure (Figure 5—figure supplement 1) and discuss them in the following way after the description of the result with ARF3:

“Auxin signaling and ARF3 have been shown to act by modifying acetylation of histones (Wu et al., 2015; Chung et al., 2019 and Long et al., 2006). Pharmacological inhibition of histone deacetylases (HDACs) alone was able to trigger concomitant activation of DR5 at P_0_ and P_-1_ sites in the SAM (Figure 5—figure supplement 1I-K). These results suggest that auxin signal integration depends on a functional ARF-dependent auxin nuclear pathway.”

Minor comment:The manuscript would benefit from English language polishing, particularly to clarify word choices such as “exposition of cells to auxin” which I take to mean “exposure of cells” (Abstract and Discussion) and “diverge fluxes away from areas” where I suspect “divert fluxes away” (Discussion) is meant.

We apologize for this. We effectively meant “exposure” and “divert”. We have polished the English language as requested with the help of a native English-speaking colleague.

Reviewer 3:This manuscript uses quantitative imaging to investigate the role of auxin in the rhythmic initiation of organs on the flanks of the shoot apical meristem. Using a ratiometric auxin sensor, the authors establish a precise spatial and temporal map of auxin accumulation in the SAM. They show that this stereotypical map is highly conserved from plant to plant, and rotates by 137 degrees (the divergence angle between consecutive organs) every 12h (the timing between the initiation of 2 consecutive organs). The authors also analyze the polarity of PIN1 polarity in the SAM and suggest a pattern of auxin fluxes different from what was previously proposed. The manuscript also describes the expression of auxin biosynthesis genes YUCCA1-11, which suggest that young primordia act as sources of auxin for the rest of the SAM, rather than auxin sinks as previously thought. Finally, the authors show that auxin response, as indicated by a DR5 reporter, is delayed compared to auxin accumulation, due to the need for cells to be exposed to auxin for a prolonged time before auxin response can take place; they suggest and propose this delay is linked to chromatin modification.This study represents a tremendous amount of work and provides invaluable quantitative information on the role of auxin in patterning the SAM. I am particularly appreciative of the fact that the authors not only quantify auxin accumulation and response in a single representative shoot apical meristem, but do so in a large number of plants to show that the mechanisms they describe is highly conserved from plant to plant.An important part of the data presented here is very convincing and conclusions are well supported. I believe this paper will be an important advance in our understanding of plant development.

We thank the reviewer for this very positive evaluation of our manuscript.

The link between histone deacetylation and auxin response in the SAM deserves further attention, but again, the amount of data presented in this study is impressive, and sufficient for publication.

We agree with this reviewer and others that our analysis of the implication of histone acetylation was too preliminary. We also believe that properly demonstrating how histone acetylation is involved in time integration would take us beyond what is reasonable to include in this manuscript. In this revised version, we only kept in the discussion the possibility that histone acetylation is involved in time integration based on the literature, as we believe that this is an important discussion point.

In link to this comment, note however that we have kept the results of the TSA treatments but rather as a way to support an implication of auxin signaling in time integration of the auxin signal. Indeed other reviewers suggested that genetics could be used to show an involvement of the auxin signaling pathway in time integration of the auxin signal. A key role for ARF3/ETTIN in organ initiation at the SAM has recently been demonstrated by a publication from the laboratory of Doris Wagner (Chung et al., 2019). We have included in the revision data demonstrating the ARF3/ETTIN gene is necessary for the integration of the auxin signal. These new data are included in Figure 5 (Figure 5N-Q) and a new Figure 5—figure supplement 2 (Figure 5—figure supplement 2D,E) and described in the text in the following way:

“The Auxin Response Factor (ARF) ETTIN (ETT/ARF3) plays an important role in promoting organogenesis in the SAM (Wu et al., 2015; Chung et al., 2019). We found that in a loss-of function ett3 mutant the expression of DR5 was restricted to only 2-3 cells at sites of organogenesis, and that a 300’ 1mM auxin treatment did not induce DR5 in the SAM (Figure 5N-Q and Figure 5—figure supplement 2D-E).”

As the functional link between auxin signaling and histone acetylation is well established, we now show the results TSA treatments on DR5 expression in a supplementary figure (Figure 5—figure supplement 1) and discuss them in the following after the description of the result with ARF3:

“Auxin signaling and ARF3 have been shown to act by modifying acetylation of histones (Wu et al., 2015; Chung et al., 2019 and Long et al. 2006). Pharmacological inhibition of histone deacetylases (HDACs) alone was able to trigger concomitant activation of DR5 at P_0_ and P_-1_ sites in the SAM (Figure 5—figure supplement 1I-K). These results suggest that auxin signal integration depends on a functional ARF-dependent auxin nuclear pathway.”

My main reservation is about PIN1 polarity. I do not think that confocal microscopy has a sufficient resolution to determine which side of the cell wall the PIN1 signal is located. In optimal conditions, the maximum resolution attainable with a confocal is 200nm, and the cell wall in the shoot apical meristem is barely thicker than that. I find the images and quantifications in Figure S5 B-O unconvincing at that level of resolution (also why represent the levels of PI signal as curves yet the PIN1 signal as an average value over more than a half micron?). To be fair to the authors, my reservations here are not specific to this manuscript: there are other articles (e.g. Heisler et al., 2005) presenting vector fields of PIN1 orientation/auxin flux based on images on which one cannot discern which side of the cell wall PIN1 really is, so it may very well be that earlier publications were wrong and this study is right. If anything, this study looks at PIN1 with more detail than previous studies. However, I think it still fails to solve the question of PIN1 polarity. To the best of my knowledge, the only published images clearly showing PIN1 polarity used immunolocalization (e.g. Reinhardt et al., 2003), but these are performed on sections and only provide 2D information… I think the way to solve this would be to use superresolution. Single localization microscopy (max resolution 20nm) might be hard to implement on these samples, but structured Illumination microscopy (max resolution 100nm) might allow the authors to see more clearly which side of the cell wall PIN1 is located in the L1.

We agree with the reviewer that the maximum resolution of confocal microscopes is insufficient to visibly separate the PI signal from the GFP signal on images. However, our approach uses the properties of the point spread function of light from each channel in 3D, to detect spatial shifts between the two signals throughout cell walls that would not be visible on images. This is thus a computational approach to increase the resolution and to detect polarities. We have followed the suggestion of this reviewer (and others) and used super-resolution radial fluctuations (SRRF) microscopy to validate our computational approach.

A comparison between the polarities computed from a 3D confocal image and the one determined using single plane super resolution image is now shown on Figure 3B-G. We have also added a new Figure 3—figure supplement 1 with further images and a quantification of the differences between polarities determined with the two approaches. The quantification shows that more than 80% of PIN1 cellular polarities match almost perfectly (deviation less than 30°) when comparing the two approaches (Figure 3—figure supplement 1 and corresponding text).

Note that we have also reorganized the supplementary material to include Appendix 4 that is dedicated specifically to the analysis of PIN1 polarity and provide extensive details on the methods we have used.

I am also a bit confused at the differences between PIN1 vector norm (Figure 3F), divergence map (Figure 3D) and vector shield (Figure 3B). Figure 3B shows some divergence at the center of the SAM, while Figure 3D shows lower convergence than in P_-2_, P_-1_ and P_0_, but Figure 3F indicates a maximum convergence at the center of the SAM. Also, Figures 3B and 3F show pattern of vector field/vector norm at P-1 pointing from the periphery to the center of the SAM, patterns that seem to correspond to divergence values between -500 and 500 in Figure 3E, yet the divergence map in Figure 3D indicates lower divergence values for P_-2_, P_-1_ and P_0_ than in the center of the SAM. I might not fully grasp what these panels exactly represent (the equations in supplemental data are admittedly beyond me), but this doesn’t seem fully consistent to me.

As explained above, adding the comparison between 3D quantification from confocal images and super-resolution led us to significantly modify Figure 3. We do not show anymore the data from former Figure 3B to focus our analysis on the tissue-scale organization of the polarities. Also local divergence maps (as shown in former Figure 3D) are now only shown in Figure S5.

We have also clarified how we analyze PIN1 polarities organization in the text. We do not use the term “vector fields” anymore but only talk about “vector maps” and explain in the text how we go from cell wall polarity vectors to cell polarity vectors by integrating cell wall vectors. We then explain how we generate vector maps in the following way: “Local averaging of the cellular vectors obtained from confocal images was then used to calculate continuous PIN1 polarity vector maps in order to identify the dominant trends in auxin flux directions in the SAM (Figure 3I, and Appendix 4).”

Concerning the comment of the reviewer, the PIN1 vector norm is the local length of vectors in the PIN1 vector map, which may be interpreted as the local intensity of the flux as it takes into account both PIN1 polarities and the level of the protein. The divergence map is then computed from the PIN1 vector map and thus takes into account both the directionality and the intensity information: the PIN1 polarity divergence index measures not only the organization of the PIN1 vectors but also their intensity. The higher divergence (lower convergence) in the CZ despite the visual impression on PIN1 vector maps is due to the lower PIN1 vector norm in the CZ resulting from the low expression of PIN1. We agreed that the examples displayed next to the axis of the former Figure 3E were confusing (by suggesting that divergence results only from directionality) and therefore removed them from the new Figure 3J.

Below are some more minor comments:“Fingers-like protrusions (visible at P_-2_, P_-1_ and P_0_)”. P_-2_ is not labeled in any of the figures; please label it.

Thank you for spotting this. We have added the label to the Figures whenever we refer to this position.

Labelling the different primordia in Video S1 (and their changing stages) would also be useful.

As the stages of the primordia are changing all the time on the video, it would impossible to label them accurately. We have thus kept the video unchanged.

Line 165-166/Figure 3—figure supplement 3F: the expression of YUC6 in the CZ doesn’t show in the figure. Please show an image with only the YUC6-GFP channel and no PI.

We have modified Figure 3—figure supplement 3. We now present for YUC1, 4 and 6 both the GFP signal alone (Figure 3—figure supplement 3A-C) and an overlay of the GFP signal and of the PI signal (Figure 3—figure supplement 3D-F). Concerning YUC6, a weak signal can be seen at the center of the meristem as stated in the text.

It is not always clear whether panels in the different figures show pRPS5a::DII-Venus expression, or ratiometric values for qDII. This should be clarified.

We apologize for this lack of precision. We have now been careful to indicate in figure legends when DII-Venus, qDII ratio or Auxin (1-qDII) is presented.

Figure 1—figure supplement 2: I have trouble understanding Figure 1—figure supplement 2H. Legend indicates more/different colors than shown in the panel. Also, what is the orange dot?

We thank the reviewer for highlighting this mistake. We have corrected the legend of the figure but we only use one color now in the panel (see below).

In the previous version, Figure 1—figure supplement 2H showed blue curves for individual meristems that represented the error between the qDII map for this meristem at 10h and the map at 0h that is rotated from 180° to +180°. The red/orange curves corresponded to the same analysis but when the error was measured between qDII maps at 10h and the same 10h map that was rotated, in order to provide a reference. This analysis allows demonstrating that the error is minimized when the map at 0h was rotated from an angle close to 137°, and thus that a 137° rotation is a suitable way of approximating the temporal evolution of the system after one plastochrone. The blue dots indicated for each meristem (blue curves) the position of the minimal error. The orange dot was a mislabeling and we apologize for this.

With this comment of the reviewer, we realized that this figure panel was too complex. In the revised Figure 1—figure supplement 2H, we have thus only kept the blue curves i.e. the error between the qDII map for this meristem at 10h and the map at 0h that is rotated from -180° to +180°. The blue dots point to the error minimum when the rotation of the qDII map at 0h is close to 137°. We also provide in the figure legend a clearer explanation of what is shown in this figure panel.

Figure 1—figure supplement 3:– The legend is insufficient and even though one can figure out the information that is not specified in the legend, it would be much easier with a more detailed legend. Please specify that the circled areas in panel A are auxin minima. More precision about what exactly the boxes in panel A would also be appreciated. I assume it represents the extent of auxin maxima?

We have expanded the legend of this figure to explain better the panel A. Concerning the boxes, they are positioned at auxin minima and maxima. They are double boxplots that represent the radial and angular variability of the values. This is also stated in the figure legend.

– I don’t understand the difference between panels B and C – they seem to show the same data with slightly different values.

We thank the reviewer for spotting this mistake. It was indeed the same data. We have corrected the figure showing the right plot in C that shows the changes in the radial position of maxima and minima.

– I find it confusing that auxin levels between auxin maxima and minima in the last P5 (last column on the right of panel 2B) seem identical: it looks like the filled box and outline box are nearly completely superimposed. It is not clear whether this is the case in panel 2C, I don’t see an outline box.

We are following the values of the maximum and minimum in P5 over the time course in Figure 1—figure supplement 3B. As stated above the corrected Figure 1—figure supplement 3C shows their radial position. The overlap for P5 is due the fact that at this stage of flower development the value of the minimum is increasing and the value of the maximum is decreasing (as can be seen in the figure), leading to a more homogeneous radial auxin distribution in the flower.

Reviewer 4:The manuscript by Galvan-Ampudia provides convincing evidence that, in the epidermis of the inflorescence apex of Arabidopsis, cells with peak expression of a proxy of auxin concentration do not move apart because of intercalating cell growth or proliferation; rather, maxima themselves move across cells over time. The authors go on suggesting that it's the exposure of cells to auxin for a certain amount of time that defines how those cells will respond to auxin. I think this is the key and most interesting conclusion of this study

We thank the reviewer for this positive evaluation of our work and agree that these finding constitutes the key message in our manuscript.

– One, however, which I believe is not as well supported by evidence: how can the authors be sure that it's the exposure to auxin for a certain amount of time and not the total amount of auxin to which cells are exposed to over time that makes the difference? If time only were important, exposure to lower amounts of auxin over longer periods would have more dramatic effects than higher amounts over shorter periods; unfortunately, such evidence is missing from the manuscript.

We agree that consolidating this key part of our manuscript was necessary. As suggested by this reviewer as well as another one, we analyzed the effect of auxin treatments with different concentrations and durations on the induction of transcription. The new Figure 5I now shows the effect on DR5 expression of treatments with 0.2mM, 1mM and 5 mM auxin for different durations. The effect of each treatment is shown throughout the PZ (x axis with indication of the position of primordia) as the log of the difference between DR5 expression after and before the treatment. This allows us now to show that both the time of exposure and the concentration of auxin control the activation of transcription monitored by DR5, thus supporting the conclusion that cells integrate the concentration of auxin over time and not simply time. In addition, we have analyzed transcription activation by auxin (using DR5) in pin-shaped inflorescence of the pinoid mutant. Organ initiation is inhibited in pinoid meristems and our results indicate that there is no pre-established auxin distribution. We also show that all cells in the PZ of pinoid meristems can respond similarly to an exogenous auxin treatment, which provides further argument that there are no intrinsic differences in the capacity of the cells to respond to auxin. We have included these new results in Figure 5J-M and Figure 5—figure supplement 2A-C. We explain the effect of auxin treatments in wildtype and pinoid meristems in the text.

Most important, I am left wondering whether all that is biologically relevant or simply an interesting circumstance. For example, what are the effects on the development of primordia of exposure to lower amounts of auxin over longer periods and what are those of exposure to higher amounts over shorter periods? What is the biologically relevant output of those treatments?

To address this concern, we have performed daily auxin treatments of specific durations and concentrations on inflorescences from plant grown on soil in order to test how these treatments affect flower distribution on the stem and phyllotaxis. The results of these experiments support the idea that time integration of the auxin signal is biologically relevant and that its perturbation affects phyllotaxis. The corresponding data are presented on Figure 5—figure supplement 1B-D and explained in the text in the following way: “Supporting this idea, we also found that daily exogenous auxin treatments at the SAM affected phyllotaxis and that the efficiency of the treatment increased with both auxin concentration and treatment length. This was particularly evident for 30’ and 120’ treatments (Figure 5—figure supplement 1B-D). 300’ treatments were less efficient at higher auxin concentrations, possibly due to compensation mechanisms. These results suggest that temporal integration of auxin information at the SAM is essential for phyllotaxis.”

Our conclusion here is also supported by genetic data that we have added upon suggestion of the reviewer (see next point).

Moreover, one of the advantages of Arabidopsis is the outstanding genetic resources, including mutants in all the steps between perception of auxin concentrations (reported by qDII) and transcriptional responses (reported by DR5): are any of those mutants defective in their response to exposure to different amounts of auxin over different periods? And what are the biological defects associated with such defective responses?

There are few auxin signaling proteins for which a clear function has been established in the SAM. The one with the clearest implication is ARF5/MP but the mutant was shown to be insensitive to auxin in the meristem (Reinhardt et al., 2003) and thus cannot be used here. ARF3/ETTIN and ARF4 are the two other ARFs that show high expression during organ initiation (Vernoux et al., 2011). A key role for ARF3/ETTIN (together with ARF4) in organ initiation at the SAM has recently been demonstrated by a publication from the laboratory of Doris Wagner (Chung et al., 2019). In the revised manuscript, we have included data demonstrating the ARF3/ETTIN gene is necessary for the integration of the auxin signal. These new data are included in Figure 5 (Figure 5N-Q) and a new Figure 5—figure supplement 2 (Figure 5—figure supplement 2D,E) and described in the in the following way:

“The Auxin Response Factor (ARF) ETTIN (ETT/ARF3) plays an important role in promoting organogenesis in the SAM (Wu et al., 2015; Chung et al., 2019). We found that in a loss-of function ett3 mutant the expression of DR5 was restricted to only 2-3 cells at sites of organogenesis, and that a 300’ 1mM auxin treatment did not induce DR5 in the SAM (Figure 5N-Q and Figure 5—figure supplement 2D-E).”

Moreover we also added data that, together with published work (Simonini et al., 2017), show that ett-22/arf3 mutants have strong defects on phyllotaxis. These results have been included on Figure 5—figure supplement 1E-H and are explained in the text together with the results from the daily auxin treatments of specific durations and concentrations on inflorescence (see previous point). Together, these two sets of results support the conclusion that time integration of the auxin signal during organ initiation is important for phyllotaxis.

Related to this part of the manuscript, the difference between Figure 5K and J is small but visible; however, I no longer see it in the quantification (Figure 5L), which is the real data.

The difference was clearly visible and statistically significant in the quantification we presented Figure 5—figure supplement 1C in the previous version of the manuscript (now Figure 5—figure supplement 1K; see below). On Figure 5K in the previous version of the manuscript, there was effectively a script problem and a number of points were removed during the generation of the panel.

As other reviewers suggested, the link to histone acetylation was too preliminary. As properly demonstrating how histone acetylation is involved in time integration would take us beyond what is reasonable to include in this manuscript, we are not presenting the TSA treatments data in a main figure anymore. We have however kept the results of the TSA treatments but rather as a way to support the implication of auxin signaling in time integration of the auxin signal. As the functional link between auxin signaling and histone acetylation is well established, we now show the results of the TSA treatments on DR5 expression in a supplementary figure (Figure 5—figure supplement 1) and discuss them in the following way after the description of the result with ARF3:

“Auxin signaling and ARF3 have been shown to act by modifying acetylation of histones (Wu et al., 2015; Chung et al., 2019 and Long et al., 2006). Pharmacological inhibition of histone deacetylases (HDACs) alone was able to trigger concomitant activation of DR5 at P_0_ and P_-1_ sites in the SAM (Figure 5—figure supplement 1I-K). These results suggest that auxin signal integration depends on a functional ARF-dependent auxin nuclear pathway.” Note also that, in the revised manuscript, we have kept only in the discussion the possibility, based on the literature, that histone acetylation could provide a mechanism for time integration.

Still related to this part of the manuscript is the blatant absence of reference to the R2D2 sensor reported by Liao et al., 2015, to which — for all I can see — qDII is identical, except for the fluorescence protein used. This should of course be corrected.

R2D2 from Liao et al., 2015 consists in DII-VENUS and a reference fluorescent protein expressed each under their own promoter. For qDII, DII-VENUS and the TagBFP reference protein are expressed from the same promoter, to ensure an equimolar production of DII-VENUS independent of promoter activity variations.

This has been clarified in the text in the following way:

“qDII differs from previously used tools (Liao et al., 2015) in producing DII-VENUS and a non-degradable TagBFP reference stoichiometrically from a single RPS5A promoter (Wend et al., 2013; Goedhart et al., 2011)(Figure S1A-H).”

I also find interesting the part of the manuscript in which the authors suggest that claims based on qualitative visual inspection of PIN1 asymmetric distribution at the plasma membrane in the epidermis of the inflorescence apex may not always be accurate. I agree with the authors, but I don’t think their method is convincingly better: the lateral resolution of conventional light microscopy, including confocal microscopy, does not allow resolving plasma membranes of adjacent cells separated by such a thin cell wall. It would instead be necessary to compare the results of the authors' method with those of super-resolution or electron microscopy to understand the advantages and limitations of the authors' method. As mentioned, this is an interesting and meritorious part of the manuscript, but also one that is foreign to the main point of the manuscript — whether auxin concentration is only the result of transport or also that of production, for example, is not the main message of this study. And frankly I find naive that one would think that production, degradation, conjugation, etc. are irrelevant to auxin concentrations, given the strong genetic evidence that in fact they are.

We agree that a vision where only transport control the distribution of auxin is naive but the strong emphasis on the role of PIN1 in the SAM in the literature has somehow obscured the contributions of the metabolic regulation. We have attempted here to fill this gap because the spatio-temporal distribution of auxin cannot be easily explained by the dominant hypothesis present in the literature (proposing that points of convergence of PIN1 pumps lead to local accumulation of auxin). Our goal here is to provide a quantitative framework linking auxin transport and biosynthesis to auxin distribution. While we agree that it is not the main point of the manuscript, we still believe that it adds an interesting dimension to our work by providing a unique high-resolution tissue scale spatio-temporal reference dataset.

We agree with the reviewer that the resolution of confocal microscopes is insufficient to visibly separate the PI signal from the GFP signal on images. However, our approach uses the properties of the point spread function of light from each channel in 3D, to detect spatial shifts between the two signals throughout cell walls that would not be visible on images. This is thus a computational approach to increase the resolution and to detect polarities. We have followed the suggestion of this reviewer (and others) and used super-resolution radial fluctuations (SRRF) microscopy to validate our computational approach.

A comparison between the polarities computed from a 3D confocal image and the one determined using single plane super resolution image is now shown on Figure 3B-G. We have also added a new Figure 3—figure supplement 1 with further images and a quantification of the differences between polarities determined with the two approaches. The quantification shows that more than 80% of PIN1 cellular polarities match almost perfectly (deviation less than 30°) when comparing the two approaches (Figure 3—figure supplement 1 and corresponding text).

Note that we have also reorganized the supplementary material to include an Appendix 4 that is dedicated specifically to the analysis of PIN1 polarity and provide extensive details on the methods we have used.

A few minor points related to this part:– The authors seem to use interchangeably PIN and PIN1: PIN1 is one of the 8 genes in the PIN family, so there are no PIN1 genes/proteins as the authors instead write:

It should have been PIN1 throughout the manuscript and we have corrected this.

A network of PIN-FORMED1 (PIN1) efflux carriers– And what do the authors mean when they write that those PIN1 [sic] efflux carriers form a network? A genetic interaction network? A protein interaction network?

The organization of the PIN1 fluxes can be represented as an oriented graph whose vertices are the cell centroids and whose oriented edges between two vertices reflects the cell adjacency and the direction of the PIN1 polarity. This is the reason why we used the term network here but to avoid confusion we have removed the reference to network when talking about PIN1.

Reviewer 5:The self-organizing properties of plant shoot meristems have fascinated researchers for decades. In the past twenty years genetic and substance-application studies have demonstrated the role plant hormones and other factors. These findings have led to complementing experimental and theoretical approaches, which link measurable parameters, like the concentrations and transport routes of the plant hormone auxin, to algorithms explaining self-organization. It is amazing that through all the thirteen years since the publication of the first of such integrated studies, many fundamental difficulties in measuring critical parameters (e.g. auxin distribution) have persisted.It is within this context, that the submitted study fills a void as it demonstrates the successful use of new tools for a better quantification of critical parameters. The authors are introducing new markers, use them in intelligent combinations with each other and with previously characterized markers and make some stunning observations on the reproducibility of auxin distribution patterns. In particular, the use of a ratiometric auxin sensor is a huge achievement, when combined with live imaging able to show fluctuating auxin waves in the meristem at high resolution in both space and time. Moreover, by relating those patterns to auxin-response patterns at the transcriptional level, the authors raise the possibility that a cellular memory of auxin exposure has a critical role in directing a cells’ behavior.

We thank the reviewer for this very positive evaluation of our work and agree that these finding constitutes the main message in our manuscript. Note that we have consolidated our demonstration of the role of time integration of the auxin signal in organogenesis. We provide details in our response to the more specific concerns of the reviewer below.

Beyond this, the manuscript also re-assesses and re-evaluates directions of PIN1-based auxin transport routes and identifies expression sites for YUCCA genes at critical stages.Unfortunately, these two last aspects do not add much to the overall impact of the study. The data on PIN1 polarity lead to a new picture, but are based on primary acquisition technology which is as problematic as in previous studies.The YUCCA expression sites in developing organs, though presented as concept-changing revelations in the Abstract, may be reproducible, but could be gratuitous as there are no genetic tests in the entire paper.

Those were fair criticisms. However, we have now provided in the revised manuscript (in response of the concerns of this reviewer and others) new data showing that 1- our acquisition technology performs well and 2- that YUC expression sites are unlikely gratuitous. We provide details in our response to the more specific concerns of the reviewer below. Note also that we are providing genetic data in this revised version of the manuscript (see below).

For the four reasons listed below, I believe that the claim of the study (though surprisingly vaguely worded in title and Abstract) is too high. The authors have made great progress in measuring the dynamics of auxin distributions, and detected delay phases in transcriptional auxin responses. With a bit more cell biological work on the “hysteresis” effect, these two aspects alone would constitute a tremendous contribution.

We agree that our Abstract was too vague. We have now rewritten the Abstract focusing on the main discoveries of the work i.e., as pointed out by the reviewer, the fact that auxin carries both spatial and temporal information and the demonstration that temporal integration of auxin information is necessary for activation of transcription.

By contrast, the other, more premature parts just damage the long-term reputation of the study as a whole.

Following the reviewer advice, the part of the results that aimed at showing the role of histone acetylation in time integration of the auxin signal has been modified (see concern 3 below) and we do not claim anymore that we provide evidence for this in the manuscript.

Concerning the part on the auxin transport, while we agree that it is not the main point of the manuscript, we still believe that it adds an interesting dimension to our work by providing a high resolution tissue scale spatio-temporal reference dataset with information on auxin, distribution, its transport and its biosynthesis. It is also justified by the fact that we describe a dynamics of auxin distribution that cannot be easily explained by our current vision.

For these reasons, we have kept the analysis of auxin transport and biosynthesis in this extensively revised version of the manuscript, while at the same time focusing the logic of manuscript more clearly on the central part of the manuscript: auxin distribution dynamic characterization and the temporal integration of the auxin signal during organ initiation. As explained in more details below, all the parts of the manuscript have been consolidated.

1) Unlike for auxin concentration, the authors have no fundamentally new solution for overcoming the uncertainties in determining location, polarity and intensity of auxin transport routes and for localizing auxin sources. They end this paragraph with the revealing sentence “PIN1 in auxin distribution could also have been overestimated” by referring to not further specified “other carriers”.

We now demonstrate that we do describe a solution for a quantitative analysis of PIN1 polarity distribution. Our approach uses confocal images and the properties of the point spread function of light from each channel in 3D, to detect spatial shifts between the two signals throughout cell walls that would not be visible on images. This is thus a computational approach to increase the resolution and to detect polarities. Following suggestions from several reviewers, we used super resolution radial fluctuations (SRRF) microscopy to validate our computational approach. A comparison between the polarities computed from a 3D confocal image and the one determined using single plane super resolution image is now shown on Figure 3B-G. We have also added a new Figure 3—figure supplement 1 with further images and a quantification of the differences between polarities determined with the two approaches. The quantification shows that more than 80% of PIN1 cellular polarities match almost perfectly (deviation less than 30°) when comparing the two approaches (Figure 3—figure supplement 1 and corresponding text). We have also reorganized the supplementary material to include a Supplementary Methods 3 that is dedicated specifically to the analysis of PIN1 polarity and provide extensive details on the methods we have used. Note that the approach we describe in the manuscript allows to obtain not only quantitative information on the direction of the PIN1-mediated transport but also on the intensity of the protein on each wall of the meristem, which is a likely determinant parameter of the transport intensity.

Also we have removed the sentence pointed out by the reviewer, as it was unnecessarily vague and, in doing so, we keep the focus on the parameters that we have analyzed in order to keep the claim of the manuscript in line with the data we present. For the same reason, we also insist on the fact that we have aimed at providing a quantitative tissue-scale vision of the organization of PIN1 polarities and we have been careful to avoid premature conclusions.

For the localization of the auxin sources, we provide details in our response to the point 3 of the reviewer below.

2) Even if one could measure all necessary parameters at the desired resolution, to make a point that the new findings lead to a consistent new picture, extremely detailed mathematical modelling would have to accompany each proposed novel mechanism.

It is fair to say that the consistency of the new picture we obtain would ultimately require mathematical modeling. While we are currently working on such models, adding those to the manuscript effectively goes beyond what can be reasonably included in the manuscript. Again our goal on the part concerning auxin transport and biosynthesis is to provide the community with a high-resolution tissue scale spatio-temporal reference dataset with information on auxin, distribution, its transport and its biosynthesis.

3) As even the best imaging technology merely establishes reliable correlation, there has always been criticism when studies were entirely based on imaging. There is no genetics and very few application experiments in this study to critically test the modelled hypotheses.

To address this concern, we have now added genetic data to support two conclusions of the manuscript. This first concerns the role of YUC and biosynthesis on SAM function. Genetic data have been previously published (Shi et al., 2018; Pinon et al., 2013; Cheng 2006); our expression data motivated the addition in the revised manuscript of an analysis of the yuc1yuc4 mutants. yuc1yuc4 double mutants show organ positioning and morphological defects as shown on Figure 3L and Figure 3—figure supplement 3O-U. In addition to published work, this support the idea that auxin biosynthesis from YUC1 and YUC4 in flowers is important for phyllotaxis. This is explained in the text in the following way: “In addition, yuc1yuc4 loss-offunction mutants show severe defects in SAM organ positioning and size (Shi et al., 2018; Pinon et al., 2013) (Figure 3L and Figure 3—figure supplement 3O-U). Taken with the organization of PIN1 polarities, these results suggest that P3-P5 are auxin production centers for the SAM that regulate phyllotaxis.”

The second conclusion concerns time integration of auxin information in the regulation of transcription during organogenesis. We have now analyzed the effect of auxin treatments on the induction of transcription using different concentrations and durations of treatment. The results are included in Figure 5I that shows the effect on DR5 expression of treatments with 0.2mM, 1mM and 5 mM auxin for different durations. This allows us now to demonstrate that both the time of exposure and the concentration of auxin control the activation of transcription monitored by DR5, thus supporting the conclusion that cells integrate the concentration of auxin over time and not simply time. In addition, we have analyzed transcription activation by auxin (using DR5) in pinshaped inflorescence of the pinoid mutant. Organ initiation is inhibited in pinoid meristems and our results indicate that there is no pre-established auxin distribution. We also show that all cells at the periphery of the meristem can respond similarly to an exogenous auxin treatment, which provides further argument that there are no intrinsic differences in the capacity of the cells to respond to auxin. We have included these new results in Figure 5J-M and Figure 5—figure supplement 2A-C. We explain the effect of auxin treatments in wild-type and pinoid meristems in the text.

Other reviewers then suggested that genetics could be used to show an involvement of the auxin signaling pathway in time integration of the auxin signal. There are few auxin signaling proteins for which a clear function has been established in the SAM. The one with the clearest implication is ARF5/MP but the mutant was shown to be insensitive to auxin in the meristem (Reinhardt et al., 2003) and thus cannot be used here. ARF3/ETTIN and ARF4 are the two other ARFs that show high expression during organ initiation (Vernoux et al., 2011). A key role for ARF3/ETTIN in organ initiation at the SAM (alongside ARF4) has recently been demonstrated by a publication from the laboratory of Doris Wagner. We have included in the revision data demonstrating that the ARF3/ETTIN gene is necessary for the integration of the auxin signal. These new data are included in Figure 5 (Figure 5N-Q) and a new Figure 5—figure supplement 2 (Figure 5—figure supplement 2D,E) and described in the text in the following way:

“The Auxin Response Factor (ARF) ETTIN (ETT/ARF3) plays an important role in promoting organogenesis in the SAM (Wu et al., 2015; Chung et al., 2019). We found that in a loss-of function ett3 mutant the expression of DR5 was restricted to only 2-3 cells at sites of organogenesis, and that a 300’ 1mM auxin treatment did not induce DR5 in the SAM (Figure 5N-Q and Figure 5—figure supplement 2D-E).”

As the functional link between auxin signaling and histone acetylation is well established,, we now show the results of TSA treatments on DR5 expression in a supplementary figure (Figure 5—figure supplement 1) and discuss them in the following way after the description of the result with ARF3:

“Auxin signaling and ARF3 have been shown to act by modifying acetylation of histones (Wu et al., 2015; Chung et al., 2019 and Long et al., 2006). Pharmacological inhibition of histone deacetylases (HDACs) alone was able to trigger concomitant activation of DR5 at P_0_ and P_-1_ sites in the SAM (Figure 5—figure supplement 1I-K). These results suggest that auxin signal integration depends on a functional ARF-dependent auxin nuclear pathway.”

Here, the combination of auxin treatments and genetics allows us to provide a deeper understanding of how spatial and temporal auxin information is used to trigger transcription of target genes during organ initiation.

4) While being detailed in experimental documentation, at the conceptual level, the paper is rather poorly written. The following sub-points may (incompletely) illustrate, how this manuscript text would fall short of the expectations, for example in a developmental genetics graduate course.– What are the authors distinguishable definitions for “morphogen” vs. “morphogenetic regulators“?– The selection of references is correspondingly arbitrary. That substances “trigger” certain cell behaviors in some of the references is certainly insufficient to put them in the “morphogen/morphogenetic” category. Likewise, all kinds of gradients are abundant in biology; do the authors claim that distinct concentrations of auxin correspond to distinct cell fates or not?

Auxin has not been formally established as a morphogen according to the criteria used in animal systems. Auxin is clearly a regulator of morphogenesis and we used the term “morphogenetic regulator” in the manuscript for this only reason. But we agree that this introduced an unnecessary confusion. Addressing whether auxin indeed acts as a morphogen goes beyond the scope of the manuscript, although we have established in this work some key tools to do so. We have thus removed the term “morphogenetic regulator” from the revised manuscript and rather focus on the concept of positional information. Corrections have been introduced throughout the manuscript and we believe that we now provide a solid text at the conceptual level.

– Because of the pivotal role the epidermis had in earlier SAM modelling, it is surprising that the main text does not address this issue. To mention in the technical sections that an experiment “focuses” on the epidermis, does not answer what conceptual roles different tissue layers have.

The pivotal role of the epidermis is also established genetically, with the demonstration that the pin1 mutant can be complemented through expression of PIN1 only in the epidermis (Kierzkowski et al., 2013). We have clarified this in the main text: “We used the spatial distribution of 1-DII-VENUS/TagBFP as a proxy for auxin distribution, hereafter named “auxin” (Figure 1C) and focused on the epidermal cell layer (L1) where organ initiation takes place (Jonsson et al., 2006; Kierzkowski et al., 2013; Smith et al., 2006; Reinhardt et al., 2003).”

Limited genetic and functional data make it difficult to conceptualize the function of auxin in the different tissue layers and the respective contribution of these layers. As the manuscript’s focus is on linking auxin information to transcriptional activation, which also occurs first in the L1 layer, analyzing the L1 layer is logical. Addressing the role of the different tissue layers would require to analyze specifically this question, which goes beyond the scope of this manuscript.

– Some of the co-authors have implicated auxin, along with cytokinin in shaping the SAM stem cell zone (Leibfried et al., 2005, Zhao et al., 2010). How are those multiple roles of the same signal substance(s) are integrated conceptually?

We agree that this is an interesting question. There is a paragraph in the discussion (that was already in the previous version of the manuscript) that discusses how the dynamics of cell differentiation when cells exit the stem cell niche might be regulated through opposite effects of auxin and WUS on the histone acetylation status of auxin signaling and target genes. In this paragraph, we refer the reader to a recent paper some of the co-authors published in Nature Com (Ma et al., 2019). This paper further demonstrates the need of a low level of auxin signaling for stem cell activity. It is possible that different levels of auxin signaling activity (and possibly other regulations) lead to activation of different set of genes in the PZ and CZ, explaining different roles. However a significant amount of work is still necessary to understand this. As our work focuses primarily on the action of auxin in organogenesis, we have not added extra discussion on this point in the revised manuscript.

Second round of Review

Reviewer 1:In this revision many of my previous concerns were addressed, notably controls about the timing vs. total amount of auxin application and a de-emphasis of the HDAC results.This new version keeps it strengths in quantitative imaging and reveals a number of fascinating phenomenon: namely the high degree to which auxin (as read out by DIIVenus) accumulates in stereotyped patterns in the SAM and primordia and the fact that these patterns are not merely a consequence of cell and tissue growth. The model of time integration for spatial outputs, while not completely nailed down, is interesting and supported well enough to be a reasonable hypothesis.

We note that the reviewer still finds that we report fascinating phenomenons.

There are still some sections where I do not follow the logic and/or feel that the conclusion is not justified by the data. Specifically the ETT/HDAC section is not particularly well explained or convincing. Inhibition of HDACs by broad chemical is likely to be different than the behavior of ETT toward target histones. To make the case that ETTs works through HDAC I would expect to see WT, ET + HDAC inhibted and ett mutant meristems probed for the specific histone acetylation and a correlation between the affected cells/ regions and auxin behaviors. Line 258-9, is there an alternative to the idea that auxin signal integration requires an auxin dependent nuclear pathway? And how does inhibiting HDACs show this?DR5 and DII-Venus may be the most commonly used markers of auxin levels and transcriptional response, but they are still proxies, and in the case of DR5, almost certainly inaccurate in the specifics. I think the authors should refer to DII-VENUS the same way they do DR5—by name rather than “auxin” or “auxin input”. In my view, the degree to which the data presented in Figure 4 represents TRUE auxin input and output is still up for debate, but I think, given the way the community has used these reporters, the evidence of them not correlating in this assay is useful.

We feel that most of these comments can be easily addressed though adjustment of the text. Although this is stated in the text, we might have not been fully clear on the fact that the demonstration of a role for HDAC in regulation of genes by ETT has already been well established by the team of Doris Wagner in Chung et al., 2019. We thus feel that it is not needed to “make the case that ETT works through HDAC”, in accordance with the fact that the reviewer appreciates that we “de-emphasized the HDAC results”.

Reviewer 4:The manuscript by Galvan-Ampudia et al. is a resubmission of a manuscript I had previously reviewed. In that review, I suggested that the Authors had provided convincing evidence that in the epidermis of the inflorescence apex of Arabidopsis cells with peak expression of a proxy of auxin concentration do not move apart because of intercalating cell growth or proliferation; rather, it's the maxima themselves that move across cells over time. My biggest concern with the previous version of this manuscript was the lack of biological relevance for the claim — derived from the observations above — that it's the exposure of cells to auxin for a certain amount of time that defines how those cells will respond to auxin.I believe Figure 5A–I convincingly shows that treatment with 0.2 mM IAA for 300 minutes leads to – at least – the same level of DR5 expression that treatment with 1 mM IAA for 120 minutes does. However, the former treatment only led to 50% of plants with defective phyllotaxis, whereas the latter treatment gave rise to nearly 90% of plants with defective phyllotaxis (Figure 5—figure supplement 1D). Therefore, shorter treatment with higher auxin concentration led to nearly twice as many abnormal plants than longer treatment with higher auxin concentration, suggesting that the amount of time cells are exposed to auxin is perhaps not as biologically relevant as the Authors wish to claim; this reduces my enthusiasm for this manuscript.I was suprised to see that an important experiment such as the one reported in Figure 5—figure supplement 1D — one that is meant to provide evidence of the biological relevance of the Authors' observations — had been relegated to a single panel in the supplemental material. I was also suprised to find that both the number of plants on which that experiment had been performed and the statistical analysis of those data were missing. And I was surprised to see that the Authors grouped all plant responses to the treatments into a single, uninformative phenotype class vaguely defined as "defective phyllotaxis". This latter deficiency may have precluded the identification of components of plant response to the treatment that are specfically affected by the time cells are exposed to auxin. For example, though the fraction of plants showing "phyllotaxis defects" is lower when plants are treated with 0.2 mM IAA for 300 minutes than when they are treated with 1 mM IAA for 120 minutes, it is possible that the fraction of plants with a specific defect is instead higher. This could suggest what biological parameter, if any, depends on temporal integration of auxin signals; most important, it would also suggest that such temporal integration does matter for plant growth and development.

Adjusting the figure can easily be done if needed. The reviewer is right that the effects of auxin treatments could raise questions when different concentrations are compared. However, we would like to point that a duration-dependent effect is seen for the different concentrations and that the reviewer seems to have overlooked our analysis of the ett mutant that provide evidence at the SAM level of the effect of an alteration of time integration of the auxin signal. For this reason, we disagree with the final conclusion of the reviewer. But it could be that the way we did our analysis of phyllotaxis are the reason for the inconsistencies. We have some more detailed information on this experiment that we are currently analyzing. We have also plants almost ready to repeat this experiment in depth if needed.

On a minor note, the response to the reviewers' comments I had access to mentioned the use of the MP::MPDelta:GR and MP::MPDelta:EAR:GR lines generated by the Jiao lab, but I could not find any sign of them in the resubmitted manuscript.

We had simply problems to have this material working in our hands and the analysis of the ett mutant gave us most of the answers we needed.

Reviewer 5:The submitted study presents novel reporter constructs and groundbreaking imaging work that sheds new light on the distribution of auxin during organ formation in the shoot meristem. In particular the authors show precise temporal dynamics and the impressive reproducibility of auxin distribution patterns, plus they chart those relative to the movements of the participating cells. Not least, they relate those distributions to patterns of transcriptional responses to auxin, which could reflect a cellular memory of auxin exposure and the system is interrogated by external auxin applications of different concentrations and durations. Finally, the manuscript provides data on the dynamics of PIN1 polarities and maps expression sites of YUCCA genes. The strength of the manuscript clearly lies in the novel and superior precision of auxin distribution recordings. Moreover, the authors show highly reproducible dynamic (temporal) patterns with interesting relationships to the dynamics of cell proliferation in the same areas. These data provide solid experimental ground to anchor future and to test current mathematical models that link auxin distribution patterns and cell behaviour in shoot meristems. There is also some progress on quantification of the “hysteresis” effect, although the manipulations are relatively crude. Together, these findings constitute a solid and important contribution and a more relaxed density of presentation would allow for the integration of Figure 1—figure supplement 2 into the body of the manuscript, as it is too central for being left out.

We note that the reviewer praised the solidity of the findings. We would like however to point that we do not understand why our manipulations are “crude”. We did extensive quantifications of transcriptional activation down to a cellular level in treatments with 3 auxin concentrations and 2-3 durations of treatments for each concentration. We report precisely how transcription changes relatively in space, with adequate statistical processing to support our conclusions. We thus believe that we have analyzed in depth the “hysteresis effect”. Concerning Figure 1—figure supplement 2, it could certainly be a main figure if needed.

It is not really clear why those solid data should to be diluted by weaker data just to generate a comprehensive model, which would be as weak as its weakest support. The PIN polarity data are not fundamentally superior to other published records and their functional relevance as well as those of YUCCA expression data is not experimentally challenged. There are only two (as it appears, arbitrarily chosen) genotypes introduced, although there would be numerous genotypes available, including (but by far not limited to) those enabling local targeted manipulation of auxin distribution and response. Such genetic tests could challenge and provide functional support for any model.

Here, the reviewer might have overlooked the fact that we have used super-resolution microscopy to co-visualize PIN1 and PI staining in order to validate our quantitative analysis of PIN1 polarities using confocal microscopy. This is superior to analysis using visual inspection (as found in the literature) that do not provide quantitative data. The reviewer might also have overlooked the fact that we have experimentally challenged YUCCA expression using mutants (in complement of published studies) and that these mutants were not arbitrarily chosen but chosen on expression as we provided an analysis of the expression of all YUCCAs in the SAM. Our results could effectively be further tested using genetics but we believe that it is out of the scope of the manuscript.